# A Unifying View of Variational Generative Wasserstein Flows

Paul Caucheteux [1] Clément Bonet [2] Anna Korba [1]

## Abstract

Many modern generative models can be viewed as minimizing divergences between probability distributions, yet they rely on different algorithmic and geometric principles. Wasserstein gradient flows provide a continuous-time formulation for optimizing over distributions, and can be approximated through their implicit discretization via the Jordan–Kinderlehrer–Otto (JKO) scheme. In this work, we present a unified theoretical framework for generative modeling based on Wasserstein gradient flows, which we refer to as Generative Wasserstein Flows (GWF). We show that a broad class of existing methods can be derived as instances of parametric JKO schemes for $f$-divergence objectives, and we establish equivalences between several recently proposed algorithms. We extend this framework beyond $f$-divergences to Integral Probability Metrics and squared Maximum Mean Discrepancy, deriving new JKO-based generative algorithms, and clarifying their connections with GANs. We study empirically the impact of the JKO regularization for a wide set of objectives. Finally, we analyze parametric Wasserstein flows, where the dynamics are restricted to distributions induced by parametrized maps.

## 1. Introduction

The goal of generative modeling is to generate new samples from an unknown target distribution $\nu$, from which we have access to a finite number of samples. In generative modeling, many paradigms face each other including diffusion models (Song et al., 2021), flow matching (Lipman et al., 2023), optimal transport maps based methods (Makkuva et al., 2020; Rout et al., 2022) and approaches based on divergence minimization such as variational autoencoders

(Kingma & Welling, 2014), GANs (Goodfellow et al., 2014) or normalizing flows (Papamakarios et al., 2021). While the field is currently dominated by diffusion methods, which have demonstrated very good results and scalability, recent advances in normalizing flows (Zhai et al., 2025), GANs (Huang et al., 2024), and Wasserstein gradient flows (Choi et al., 2024) have shown that these alternative paradigms remain highly competitive when well designed, while allowing for faster sampling. These models differ not only in architecture but in the underlying geometric structure induced by the objective they minimize. Understanding these methods from a unified perspective remains a central challenge.

A particularly fruitful viewpoint is provided by Wasserstein gradient flows (WGFs), *i.e.*, continuous-time paths of distributions which describe the steepest descent of a functional over the space of probability measures endowed with the Wasserstein metric, namely the Wasserstein space. The Jordan–Kinderlehrer–Otto (JKO, (Jordan et al., 1998)) scheme offers a principled time-discretization of such flows through a sequence of proximal optimization problems in Wasserstein space. Recently, JKO-based methods have gained attention in generative modeling. These approaches reveal connections between likelihood-based models (Finlay et al., 2020; Vidal et al., 2023; Xu et al., 2023), transport maps (Fan et al., 2022; Choi et al., 2024), and adversarial objectives (Lin et al., 2021). However, existing works often focus on specific divergences, and specific parameterizations or algorithmic instantiations. This raises the question of how a wide range of generative methods can be derived, analyzed, and implemented in a principled and unified manner.

**Contributions.** In this work, we propose a unified theoretical framework and methodology for generative modeling based on Wasserstein gradient flows, which we refer to as Generative Wasserstein Flows (GWFs). This work makes several contributions. First, we show in Section 3 that minimizing a wide class of $f$-divergences through JKO schemes naturally leads to generative algorithms expressed with transport maps, encompassing and clarifying connections between variational Wasserstein gradient flows, unbalanced optimal transport–based methods, and adversarial training objectives. In particular, we establish the equivalence, both theoretically and practically, of several methods introduced in the last few years (Fan et al., 2022; Choi et al., 2024; Bap-

---

[1]ENSAE, CREST, IP Paris [2]CMAP, CNRS, École Polytechnique, Institut Polytechnique de Paris, Palaiseau, France. Correspondence to: Paul Caucheteux <paul.caucheteux@ensae.fr>.

*Proceedings of the 43rd International Conference on Machine Learning*, Seoul, South Korea. PMLR 306, 2026. Copyright 2026 by the author(s).

tista et al., 2025). Second, we investigate the minimization of $f$-divergences using different variational formulations such as the Donsker-Varadhan formula for the KL divergence (Birrell et al., 2022). We also extend in Section 4 the JKO framework beyond classical $f$-divergences to integral probability metrics and squared Maximum Mean Discrepancy (MMD). This leads to a novel generative scheme that reveals new connections between MMD GANs and recently proposed JKO training for GANs (Lin et al., 2021). Third, we provide in Section 5 numerical experiments which investigate the impact of the JKO regularization for training generative models for a wide range of objectives. Finally, we present in Section 6 a theoretical framework for parametric Wasserstein flows, where the evolution is restricted to distributions generated by parametrized maps, and interpret these methods as projected Wasserstein gradient flows.

**Notation.** We write $\mathcal{M}^+(\mathbb{R}^d)$ for the set of finite non-negative measures over $\mathbb{R}^d$. We write $\mathcal{P}(\mathbb{R}^d)$ for the space of probability measures over $\mathbb{R}^d$, $\mathcal{P}_2(\mathbb{R}^d)$ its restriction to measures with bounded second moment, and $\mathcal{P}_{\mathrm{ac},2}(\mathbb{R}^d)$ the restriction of the latter space to absolutely continuous measures *w.r.t.* the Lebesgue measure. Given $f : (0,\infty) \to \mathbb{R}$ convex with $f(1) = 0$, the $f$-divergence is defined as $\mathrm{D}_f(\mu\|\nu) = \int f(\frac{\mathrm{d}\mu}{\mathrm{d}\nu})\mathrm{d}\nu$ for $\mu$ absolutely continuous with respect to $\nu$ ($\mu \ll \nu$, $\frac{\mathrm{d}\mu}{\mathrm{d}\nu}$ denoting the Radon-Nikodym density of $\mu$ relative to $\nu$), and $+\infty$ otherwise. For $\nu$ a given target distribution, important particular cases are the reverse KL divergence $\mathcal{F}(\mu) = \mathrm{KL}(\mu\|\nu) = \int \log(\frac{\mathrm{d}\mu}{\mathrm{d}\nu})\mathrm{d}\mu$ obtained for $f(x) = x\log x$, and the forward KL divergence $\mathcal{F}(\mu) = \mathrm{KL}(\nu\|\mu)$ obtained for $f(x) = -\log x$. For a measurable map $\mathrm{T} : \mathbb{R}^d \to \mathbb{R}^d$, $\mathrm{T}_{\#}\mu$ denotes the pushforward measure, which satisfies for all $A \in \mathcal{B}(\mathbb{R}^d)$, $\mathrm{T}_{\#}\mu(A) = \mu(\mathrm{T}^{-1}(A))$. Id denotes the identity map on $\mathbb{R}^d$. For $\varphi : \mathbb{R}^d \to \mathbb{R}^p$, $L^k(\mu)$ denotes the space of functions such that $\int \|\varphi\|^k\,\mathrm{d}\mu < \infty$ and is equipped with the norm $\|\cdot\|_{L^k(\mu)}$.

## 2. Wasserstein Gradient Flows

In this section, we provide some background on optimal transport (OT) and Wasserstein gradient flows to minimize a functional. We refer to (Ambrosio et al., 2008; Villani et al., 2009; Santambrogio, 2015) for more details.

### 2.1. Wasserstein Space

For two probability distributions $\mu$ and $\nu$ in $\mathcal{P}_2(\mathbb{R}^d)$, the 2-Wasserstein distance is defined by means of an optimal coupling between $\mu$ and $\nu$ in the following way:

$$\mathrm{W}_2^2(\mu,\nu) := \inf_{\pi\in\Pi(\mu,\nu)} \int \|x-y\|_2^2\,\mathrm{d}\pi(x,y), \quad (1)$$

where $\Pi(\mu,\nu)$ denotes the set of possible couplings between $\mu$ and $\nu$. Beyond this primal formulation, the quadratic cost

endows the problem with a particularly rich structure. In particular, when $\mu \in \mathcal{P}_{\mathrm{ac},2}(\mathbb{R}^d)$, the OT plan is induced by a deterministic map, as stated by Brenier's theorem (Brenier, 1991). More precisely, there exists a convex function $\varphi : \mathbb{R}^d \to \mathbb{R}$ such that the optimal coupling $\pi^\star$ is given by $(\mathrm{Id}, \nabla\varphi)_{\#}\mu$, and $\mathrm{W}_2$ can be expressed as

$$\mathrm{W}_2^2(\mu,\nu) = \int \|x - \nabla\varphi(x)\|_2^2\,\mathrm{d}\mu(x) = \|\mathrm{Id} - \nabla\varphi\|_{L^2(\mu)}^2.$$

The map $\mathrm{T}^\star = \nabla\varphi$ is referred to as the OT map and belongs to $L^2(\mu)$. A complementary viewpoint is provided by the Kantorovich dual formulation, which highlights the variational nature of the problem. In this case, $\mathrm{W}_2$ admits the (semi-dual) representation (Villani et al., 2009, Th. 5.10)

$$\tfrac{1}{2}\mathrm{W}_2^2(\mu,\nu) \quad = \quad \sup_{\phi\in L^1(\nu)} \quad \int \phi^c\,\mathrm{d}\mu \quad + \quad \int \phi\,\mathrm{d}\nu \quad (2)$$

where $\phi : \mathbb{R}^d \to \mathbb{R}$ is a Kantorovich potential and $\phi^c$ denotes the $c$-transform of $\phi$, *i.e.*, $\phi^c(x) = \inf_y\{c(x,y) - \phi(y)\}$, where unless stated otherwise, we use $c(x,y) = \frac{1}{2}\|x-y\|_2^2$. Let $\mu \in \mathcal{P}_{\mathrm{ac},2}(\mathbb{R}^d)$ and denote by $\phi_{\mu,\nu} \in L^1(\nu)$ an optimal Kantorovich potential (*i.e.*, a maximizer of the semi-dual OT problem), then the OT map between $\mu$ and $\nu$ satisfies $\mathrm{T}^\star(x) = \mathrm{argmin}_y \frac{1}{2}\|x-y\|_2^2 - \phi_{\mu,\nu}(y)$. This formulation will be useful for the generalization of OT to the unbalanced setting that we will consider later. Applying Danskin's theorem to $\phi_{\mu,\nu}^c$, see *e.g.* (Blondel & Roulet, 2024, Chapter 11), we also get the well known formula $\mathrm{T}^\star(x) = x - \nabla\phi_{\mu,\nu}^c(x)$ which is indeed the gradient of the convex function $x \mapsto \frac{1}{2}\|x\|^2 - \phi_{\mu,\nu}^c(x)$ (Santambrogio, 2015, Th. 1.17).

### 2.2. Gradient Flows in Wasserstein Space

Wasserstein gradient flows (Ambrosio et al., 2008; Santambrogio, 2017) describe the time evolution of a probability measure following the steepest–descent direction of a functional $\mathcal{F} : \mathcal{P}_2(\mathbb{R}^d) \to \mathbb{R}$ with respect to the 2-Wasserstein distance $\mathrm{W}_2$. Consequently, to minimize $\mathcal{F}$, one evolves $(\mu_t)_{t\geq 0}$ via the continuity equation with velocity field the negative Wasserstein gradient:

$$\partial_t\mu_t \;=\; \nabla\cdot\big(\mu_t\,\nabla_{\mathrm{W}_2}\mathcal{F}(\mu_t)\big). \qquad \text{(WGF)}$$

The Wasserstein (sub)-gradient at $\mu$ is defined as $\nabla_{\mathrm{W}_2}\mathcal{F}(\mu) = \nabla\mathcal{F}'(\mu)$, *i.e.*, as the Euclidean gradient of the first variation $\mathcal{F}'(\mu) : \mathbb{R}^d \to \mathbb{R}$ of $\mathcal{F}$ at $\mu$[1]. It is an element of $L^2(\mu)$. There are in general no closed-form solutions to (WGF). To approximate the flows, different schemes based on time discretizations have been proposed, *e.g.* based on explicit (forward) Euler discretizations, implicit (backward)

---

[1]If it exists, it is the function such that for any $\nu \in \mathcal{P}(\mathbb{R}^d)$, $\mathcal{F}(\mu+\varepsilon(\nu-\mu)) = \mathcal{F}(\mu) + \varepsilon\int\mathcal{F}'(\mu)(x)\,\mathrm{d}(\nu-\mu)(x) + o(\varepsilon)$.

ones, or splitting schemes mixing both (Wibisono, 2018; Salim et al., 2020). The explicit Euler scheme, often referred to as the Wasserstein gradient descent (Bonet et al., 2024), is of the form, for a step-size $\gamma > 0$, $\mu_0 \in \mathcal{P}_2(\mathbb{R}^d)$,

$$\forall \ell \geq 0, \ \mu_{\ell+1} = \left(\text{Id} - \gamma \nabla_{W_2} \mathcal{F}(\mu_\ell)\right)_{\#} \mu_\ell. \quad (3)$$

In this work, we mostly focus on the implicit scheme introduced by Jordan et al. (1998), and thus also known as the JKO scheme, given for a step-size $\tau > 0$, $\mu_0 \in \mathcal{P}_2(\mathbb{R}^d)$,

$$\forall \ell \geq 0, \ \mu_{\ell+1} = \underset{\mu \in \mathcal{P}_2(\mathbb{R}^d)}{\text{argmin}} \ \mathcal{F}(\mu) + \frac{1}{2\tau} W_2^2(\mu, \mu_\ell). \quad \text{(JKO)}$$

Under mild assumptions, this scheme converges as $\tau \to 0$ towards (WGF) (Ambrosio et al., 2008, Th. 4.0.4).

## 3. JKO for Generative Modeling based on $D_f$

We now study generative modeling approaches that minimize $f$-divergences, $\mathcal{F}(\cdot) = D_f(\cdot \| \nu)$, using a JKO scheme, where $\nu \in \mathcal{P}_{ac,2}(\mathbb{R}^d)$ is the target probability measure, *e.g.*, the data distribution from which we wish to generate new samples. This section presents a unifying view of methods within this framework, which we call Generative Wasserstein Flows (GWFs).

### 3.1. Unifying JKO-based Schemes for $f$-Divergences

**Primal formulation of JKO.** If $\mu_\ell \in \mathcal{P}_{ac,2}(\mathbb{R}^d)$, we can simplify (JKO) as an optimization problem over maps

$$\begin{cases} T_{\ell+1} = \underset{T \in L^2(\mu_\ell)}{\text{argmin}} \ \mathcal{F}(T_{\#}\mu_\ell) + \frac{1}{2\tau} \|T - \text{Id}\|_{L^2(\mu_\ell)}^2 \\ \mu_{\ell+1} = (T_{\ell+1})_{\#}\mu_\ell, \end{cases} \quad (4)$$

by leveraging Brenier's theorem. In particular, $T_{\ell+1}$ is necessarily the OT map between $\mu_\ell$ and $\mu_{\ell+1}$ (Lemma F.1), and thus the gradient of a convex function. Yet if $\mu_\ell \notin \mathcal{P}_{ac,2}(\mathbb{R}^d)$, the squared norm term in the objective for T is an upper bound on the $W_2$ distance, (as a map may not be able to reach any measure). Note that for $\mathcal{F}$ an $f$-divergence, $\mu_\ell$ necessarily belongs to $\mathcal{P}_{ac,2}(\mathbb{R}^d)$ as $\nu \in \mathcal{P}_{ac,2}(\mathbb{R}^d)$. We always assume that a minimum $T_{\ell+1}$ exists, which holds *e.g.* if $\mathcal{F}$ is lower-semi continuous *w.r.t.* the narrow topology and bounded below, see *e.g.* (Ambrosio et al., 2008, Corollary 2.2.2). Mokrov et al. (2021); Alvarez-Melis et al. (2022) used this formulation and Brenier's theorem by parameterizing maps $T_{\ell+1}$ as gradient of convex functions with Input Convex Neural Networks (ICNNs) (Amos et al., 2017), but did not use it for generative modeling. The formulation (4) suffers from some limitations to minimize $f$-divergences $\mathcal{F}(\cdot) = D_f(\cdot \| \nu)$, as it requires to evaluate the density of the current iterate $\mu_\ell$. This can be solved by using invertible neural networks such as Normalizing Flows

(Papamakarios et al., 2021), and this was used *e.g.* in (Bonet et al., 2022; Vidal et al., 2023) in the JKO setting. However, such architectures are known to have limited expressiveness (Kong & Chaudhuri, 2020), although recent works have proposed more expressive architectures (Zhai et al., 2025; Gu et al., 2025). Moreover, some $f$-divergences, such as the reverse KL, also require approximating the density of the target $\nu$, which is unknown in generative modeling.

To circumvent these problems, Fan et al. (2022) proposed a method called Variational Wasserstein Gradient Flow (VWGF), by leveraging the variational formulation of $f$-divergences (Nguyen et al., 2010)

$$D_f(\mu \| \nu) = \sup_{h \in C_b(\mathbb{R}^d)} \int h \, d\mu - \int f^* \circ h \, d\nu := \mathcal{L}(\mu, h), \quad (5)$$

where the supremum is over continuous bounded maps from $\mathbb{R}^d$ to $\mathbb{R}$ (Birrell et al., 2022, Remark 1), and $f^*$ is the convex conjugate of $f$, *i.e.*, $f^*(s) = \sup_t \ st - f(t)$. In this case, the minimization problem over T in (4) is equivalent to

$$\inf_T \sup_h \int \left(\frac{1}{2\tau} \|T - \text{Id}\|_2^2 + h \circ T\right) d\mu_\ell - \int f^* \circ h \, d\nu. \quad (6)$$

This formulation can be evaluated given only samples of $\mu_\ell$ and $\nu$, making it well suited to generative modeling. It can be approximated by parameterizing T and $h$ with any neural networks through an adversarial training procedure. In that setting, the map T (or its compositions through iterations) plays the role of a generator, pushing source samples to target ones, and $h$ of a discriminator between generated and real data samples. Note that it can also be seen as a bilevel optimization problem $\inf_T \mathcal{L}(T_{\#}\mu_\ell, h^*(T_{\#}\mu_\ell))$ with $h^*(T)$ the maximum in (5), where the outer loss is $D_f(\cdot \| \nu)$ and outer loops correspond to JKO steps.

**Dual formulation of JKO.** When $\mathcal{F}(\mu) = D_f(\mu \| \nu)$, a step of the JKO scheme (JKO) can alternatively be interpreted as the source-fixed Unbalanced Optimal Transport (sUOT) problem (Liero et al., 2018; Choi et al., 2024)

$$\text{sUOT}(\mu, \nu) = \inf_{\pi_2 \in \mathcal{P}_2(\mathbb{R}^d)} \lambda_2 D_f(\pi_2 \| \nu) + W_2^2(\mu, \pi_2), \quad (7)$$

with $\lambda_2 = 2\tau$, $\mu = \mu_\ell$. The (semi-)dual of (7) is given, for $c_\tau(x, y) = \frac{1}{2\tau} \|x - y\|_2^2$, by (Vacher & Vialard, 2023)

$$\text{sUOT}(\mu_\ell, \nu) = 2\tau \cdot \sup_{h \in C_b(\mathbb{R}^d)} \int h^{c_\tau} d\mu_\ell - \int f^* \circ -h \, d\nu. \quad (8)$$

Choi et al. (2024) proposed a method called Scalable JKO (S-JKO) to simulate (JKO) for $f$-divergence objectives, by seeing it as a sequence of sUOT problems, and where each of these sUOT problems is solved using the method introduced in (Choi et al., 2023). In their setting, denoting $h$ a

maximizer of (8), the map $\mathrm{T}(x) = \operatorname{argmin}_y c_\tau(x, y) - h(y)$ corresponds both to the unbalanced transport between $\mu_\ell$ and $\nu$, and the OT map between $\mu_\ell$ and $\mu_{\ell+1}$ (the rebalanced measure) (Eyring et al., 2024; Gallouët et al., 2025). Hence, parametrizing $h^{c_\tau}$ (the $c_\tau$-transform of $h$), by a measurable map T as $h^{c_\tau}(x) = \inf_{\mathrm{T}} \frac{1}{2\tau} \|x - \mathrm{T}(x)\|_2^2 - h(\mathrm{T}(x))$ and swapping the infimum and the integral[2], (8) is equivalent to

$$\sup_h \inf_{\mathrm{T}} \int \left( \frac{1}{2\tau} \|\mathrm{T} - \mathrm{Id}\|_2^2 - h \circ \mathrm{T} \right) \mathrm{d}\mu_\ell - \int f^* \circ -h \, \mathrm{d}\nu. \quad (9)$$

This problem can again be solved by parameterizing T and $h$ with neural networks through an adversarial training scheme. While (9) and (6) differ in the order of the supremum and infimum (and in the sign convention for $h$), we show below that they are in fact equivalent.

**Proposition 3.1.** *Let $\mu_\ell, \nu \in \mathcal{P}_{\mathrm{ac},2}(\mathbb{R}^d)$. Then, the objectives (6) (VWGF) and (9) (S-JKO) are equal.*

Hence, although not stated by Choi et al. (2024), S-JKO and VWGF constitute the exact same generative scheme. We refer to Appendix F.1 for the proof. Formulas of the objectives for common $f$-divergences are detailed in Appendix C.1.

**Explicit scheme as a JKO step.** $f$-divergences are only well suited to compare probability measures sharing the same support. Hence, several works regularized them with Moreau's envelope, allowing to compare any probability measures. The MMD's Moreau's envelope was *e.g.* used in (Glaser et al., 2021; Chen et al., 2025; Stein et al., 2026). Baptista et al. (2025) recently proposed a regularized $f$-divergence based on the $W_2$-Moreau's envelope:

$$\mathcal{F}_\varepsilon(\mu) = \inf_{\eta \in \mathcal{P}_2(\mathbb{R}^d)} W_2^2(\mu, \eta) + \varepsilon \mathrm{D}_f(\eta \| \nu). \quad (10)$$

Baptista et al. (2025) proposed to minimize $\mathcal{F}_\varepsilon$ in the particular case of the reverse KL divergence using an explicit discretization, *i.e.*, the Wasserstein gradient descent scheme (3). We show in the next proposition that this scheme coincides for any $f$-divergence with the JKO scheme of $\mathcal{F} = \mathrm{D}_f(\cdot \| \nu)$ for a particular choice of $\varepsilon$.

**Proposition 3.2.** *Let $\nu \in \mathcal{P}_{\mathrm{ac},2}(\mathbb{R}^d)$. A step of Wasserstein gradient descent of (10) with step size $\gamma = \frac{1}{2}$ and $\varepsilon = 2\tau$ is equivalent to a step of (JKO) for $\mathcal{F}(\mu) = \mathrm{D}_f(\mu \| \nu)$.*

The proof is given in Appendix F.2. Baptista et al. (2025) proposed a scheme with ICNNs based on (Makkuva et al., 2020) to perform the Wasserstein gradient descent of $\mathcal{F}_\varepsilon$. We show in Appendix C.3 that this scheme is equivalent to solving the JKO scheme using S-JKO or VWGF with the Donsker-Varadhan formulation that we introduce next.

---

[2]Using (Rockafellar & Wets, 1998, Theorem 14.60 and Example 14.29) as $(x, y) \mapsto c_\tau(x, y) - h(y)$ is a normal integrand since $x \mapsto c_\tau(x, y) - h(y)$ is measurable and $y \mapsto c_\tau(x, y) - h(y)$ is continuous.

**Donsker-Varadhan.** Besides the usual variational formulation of $f$-divergences (5), the KL divergence admits another dual variational representation, known as the Donsker-Varadhan formula (Donsker & Varadhan, 1983)

$$\mathrm{KL}(\mu \| \nu) = \sup_{h \in C_b(\mathbb{R}^d)} \int h \, \mathrm{d}\mu - \log \int e^h \, \mathrm{d}\nu. \quad (11)$$

Note that for any function $h$, this objective is pointwise greater or equal to the classical variational bound (5), which corresponds to $\int h \, \mathrm{d}\mu - \int e^{h-1} \mathrm{d}\nu$ for the reverse KL. In particular, both representations coincide at optimality but Donsker-Varadhan yields a tighter lower bound. This was noticed by Fan et al. (2022); Baptista et al. (2025), but it has not yet been used in a generative modeling scheme to the best of our knowledge. We provide numerical comparisons in Section 5.1, and refer to Appendix C.2 for more details.

### 3.2. Reparametrization Trick and GANs

**Reparametrization trick.** At each JKO step, since the update is given by $\mu_{\ell+1} = \mathrm{T}_{\ell+1 \#} \mu_\ell$, we need access to samples from the current distribution $\mu_\ell$. Hence, obtaining such samples requires recursively composing all previous transport maps, which leads to $\mathcal{O}(\ell^2)$ evaluations of neural networks just to get samples during the training (assuming we generate new source samples at each iteration $\ell$). To alleviate this complexity issue, it is possible instead to always train a neural network which pushes samples from the source $\mu_0$ to get samples from $\mu_{\ell+1}$, *e.g.* using the reparametrization trick of (Choi et al., 2024). We define $\bar{\mathrm{T}}_\ell = \mathrm{T}_\ell \circ \cdots \circ \mathrm{T}_1$, so that, when $\mu_0 = \mu$, we get $\bar{\mathrm{T}}_{\ell \#} \mu = \mu_\ell$, and $\bar{\mathrm{T}}_\ell = \mathrm{T}_\ell \circ \bar{\mathrm{T}}_{\ell-1}$. This enables to write (6) as

$$\inf_{\bar{\mathrm{T}}} \sup_h \int \left( \frac{1}{2\tau} \|\bar{\mathrm{T}} - \bar{\mathrm{T}}_\ell\|_2^2 + h \circ \bar{\mathrm{T}} \right) \mathrm{d}\mu_0 - \int f^* \circ h \, \mathrm{d}\nu. \quad (12)$$

Up to a change of variable, both problems are equivalent since $\mu_\ell \in \mathcal{P}_{\mathrm{ac},2}(\mathbb{R}^d)$. This trick has two advantages: first, it reduces the cost at inference time, when generating new samples from $\mu$ as it only requires to evaluate once the final map obtained by the algorithm; second, during training, it allows to sample new instances from the source $\mu$ at each iteration $\ell$, without composing $\ell$ maps.

**GANs and their link to JKO.** GANs are a class of generative models based on minimizing objectives defined over the parameters of two neural networks (the generator and the discriminator) with a min-max formulation. Depending on the choice of the discrepancy, several families of GANs exist, such as the ones based on $f$-divergences (Nowozin et al., 2016), Wasserstein distances (Arjovsky et al., 2017; Genevay et al., 2017; Korotin et al., 2021) or Integral Probability Metrics (Mroueh et al., 2017; Li et al., 2017).

The class of $f$-GANs (Nowozin et al., 2016) trains the discriminator $h$ by maximizing the lower bound derived

through the dual representation of $\mathcal{F}(\mu) = \mathrm{D}_f(\nu\|\mu)$, and the generator T by minimizing the lower bound of $\mathcal{F}$ *w.r.t* $\mu = \mathrm{T}_\#\mu_0$ for some $\mu_0 \in \mathcal{P}_2(\mathbb{R}^d)$, *i.e.*, it solves

$$\inf_{\mathrm{T}} \sup_{h} \int h \, \mathrm{d}\nu - \int f^* \circ h \circ \mathrm{T} \, \mathrm{d}\mu_0. \quad (13)$$

In contrast with JKO methods presented earlier, GANs solve one *global* optimization problem to bring $\mu$ close to $\nu$. In practice, the discriminator is often reparametrized as $h = g_f \circ \tilde{h}$ with $g_f : \mathbb{R}^d \to \mathbb{R}$ valued in the domain of $f^*$, and the optimization is done over $\tilde{h}$. Given the choice of $g_f$, it allows to recover many formulations of GANs including vanilla GAN, see (Nowozin et al., 2016). For simplicity, we will only write $h$. Lin et al. (2021) proposed the Relaxed Wasserstein Proximal GAN (RWP-GAN) method, adding an outer optimization loop and a regularization term over the generator, *i.e.*, solving a sequence of regularized problems, where at each step $\ell$ of the sequence the objective writes

$$\inf_{\mathrm{T}} \sup_{h} \int h \, \mathrm{d}\nu - \int f^* \circ h \circ \mathrm{T} \, \mathrm{d}\mu_0 + \tfrac{1}{2\tau}\|\mathrm{T} - \mathrm{T}_\ell\|_{L^2(\mu_0)}^2. \quad (14)$$

**Proposition 3.3.** (6) *(VWGF) and* (14) *(RWP-$f$-GAN) correspond to the same scheme applied to opposite orientations of the $f$-divergence. When instantiated with the same orientation, the two methods coincide.*

Consequently, in their original formulations, VWGF applied to $\mathrm{D}_f(\mu\|\nu)$ corresponds to RWP-$f$-GAN applied to the reverse divergence $\mathrm{D}_f(\nu\|\mu)$ (and vice versa), we refer to Appendix F.3 for the proof. For example, RWP-GAN for the forward KL corresponds to VWGF with the reverse KL.

To summarize, we have proved in this section the equivalence between several schemes, among which VWGF (Fan et al., 2022) and SJKO (Choi et al., 2024) as well as with the regularized GANs (RWP-GAN) methods. These equivalent formulations can all be interpreted as instances of the Generative Wasserstein Flow (GWF) framework introduced in this work. A schematic summary of these correspondences and of the overall GWF framework is provided in Figure 3.

## 4. JKO for Generative Modeling based on $\mathrm{MMD}^2$ and IPMs

In this section, we define a novel family of generative modeling methods, based on JKO schemes applied to functionals $\mathcal{F} = \mathrm{D}(\cdot, \nu)$ for D divergences which can be obtained through an alternative variational formulation other than $f$-divergences, hence extending the GWF framework. In particular, we consider Integral Probability Metrics (IPMs) (Müller, 1997) such as the Maximum Mean Discrepancy (MMD) and the Wasserstein-1 distance. While not stricto sensu an IPM, we also focus on the squared MMD, and make connections with MMD GANs.

---

**Algorithm 1** Primal–Dual Training Procedure of $\mathrm{MMD}_k^2$

**Inputs**: $K$ outer (JKO) steps, $N$ inner steps, $k$ a p.s.d. kernel, $\tau$ JKO step size, $\eta_\mathrm{T}$ network step size
Initialize $\mathrm{T}_{\theta_0^N} = I_d$
**for** $\ell = 1, \ldots, K$ **do**
  **for** $i = 1, \ldots, N$ **do**
    Sample $(z_j)_{j=1}^n \sim \mu_0$, $(y_j)_{j=1}^m \sim \nu$
    Compute $x_j(\theta_\ell^i) = \mathrm{T}_{\theta_\ell^i}(z_j)$ for $j = 1, \ldots, n$
    Define $g = \frac{1}{n}\sum_{j=1}^n k(x_j(\theta_\ell^i), \cdot) - \frac{1}{m}\sum_{j=1}^m k(y_j, \cdot)$
    $J(\theta_\ell^i) = \frac{1}{n}\sum_{j=1}^n \left[ \frac{1}{2\tau}\|\mathrm{T}_{\theta_{\ell-1}^N}(z_j) - x_j(\theta_\ell^i)\|_2^2 - g(x_j(\theta_\ell^i)) \right]$
    Update $\theta_\ell^{i+1} = \theta_\ell^i - \eta_\mathrm{T}\nabla_\theta J(\theta_\ell^i)$.
  **end for**
  $\theta_{\ell+1}^1 = \theta_\ell^N$
**end for**

---

**Dual formulation of JKO.** Beyond $f$-divergences, several popular metrics are defined through a variational formulation. For instance, IPMs are distances between probability distributions defined as $\mathrm{IPM}_\mathcal{G}(\mu, \nu) = \sup_{f \in \mathcal{G}} \left| \mathbb{E}_\mu[f(X)] - \mathbb{E}_\nu[f(X)] \right|$ for a chosen class of test functions $\mathcal{G}$. When $\mathcal{G}$ is the unit ball of a reproducing kernel Hilbert space $\mathcal{H}_k$ with positive semidefinite kernel $k$, we obtain the MMD, and for $\mathcal{G}$ the set of 1-Lipschitz functions, we obtain the Wasserstein-1 distance, *i.e.* the OT problem with cost $c(x, y) = \|x-y\|_2$ (Sriperumbudur et al., 2009). While not an IPM, the squared MMD also admits a variational form as (see Lemma D.1 or (Mroueh & Nguyen, 2021))

$$\tfrac{1}{2}\mathrm{MMD}_k^2(\mu, \nu) = \sup_{g \in \mathcal{H}_k} \int g \, \mathrm{d}(\mu - \nu) - \tfrac{1}{2}\|g\|_{\mathcal{H}_k}^2. \quad (15)$$

In particular, the supremum for the squared MMD can be found in closed-form as $g(\cdot) = \int k(x, \cdot) \, \mathrm{d}(\mu - \nu)(x)$.

Following the method of (Choi et al., 2024) for $f$-divergences, we derive the dual formulation of the JKO scheme seen as a source-fixed UOT problem with $\mathcal{F}$ chosen either as an IPM or the squared MMD. Full derivations are in Appendix D. In particular, the dual for IPMs follows from (Manupriya et al., 2024). We now present the method for the squared MMD which is new to the best of our knowledge. In this case, the dual of the source-fixed UOT problem is

$$\sup_{g \in \mathcal{H}_k} \int g^c \, \mathrm{d}\mu + \int g \, \mathrm{d}\nu - \tfrac{1}{4\tau}\|g\|_{\mathcal{H}_k}^2. \quad (16)$$

Hence, for the $\ell$-th step of JKO for the squared MMD, we can then parametrize $g^c$ by a measurable map T as $g^c(x) = \inf_\mathrm{T} c(x, \mathrm{T}(x)) - g(\mathrm{T}(x))$ and obtain for $c(x, y) = \|x - y\|_2^2$ and with the change of variable $g = 2\tau h$ the problem

$$\sup_{h \in \mathcal{H}_k} \inf_{\mathrm{T}} \int \left( \tfrac{1}{2\tau}\|\mathrm{Id} - \mathrm{T}\|_2^2 - h \circ \mathrm{T} \right) \mathrm{d}\mu_\ell + \int h \, \mathrm{d}\nu - \tfrac{1}{2}\|h\|_{\mathcal{H}_k}^2. \quad (17)$$

Hence, we define a generative modeling scheme by solving this problem. Straightforward computations (detailed in Appendix D.3) yield the scheme presented in Algorithm 1, where we additionally use the reparametrization trick as described in Section 3.2. Note that compared to the case of $f$-divergences, we do not have an adversarial scheme because the witness function (the discriminator) is available in closed form in the case of squared MMD. This scheme can be seen as a JKO regularized version of the Generative Moment Matching scheme (Li et al., 2015; Dziugaite et al., 2015).

**JKO MMD GAN.** We observe in practice that the previous algorithm does not perform well in high dimensions (see Appendix J.2 for further details). We hypothesize that one might need a discriminator function that is adapted to the current iteration. To overcome this, we take inspiration from the MMD GAN literature (Li et al., 2017; Bińkowski et al., 2018), and learn simultaneously an embedding through which we compare the measures (hence a discriminator). The problem becomes

$$\max_{\phi} \min_{\mathrm{T}} \tfrac{1}{2}\|\mathrm{Id} - \mathrm{T}\|_{L^2(\mu_\ell)}^2 + \mathrm{MMD}_{k_\phi}^2(\mathrm{T}_{\#}\mu_\ell, \nu), \quad (18)$$

where $k_\phi(x,y) = k\big(\phi(x), \phi(y)\big)$. The resulting optimization problem is solved using an adversarial scheme, see Appendix D.4 and Algorithm 2. This new algorithm can be seen as a specification of the RWP-GAN to MMD GANs (*i.e.*, a JKO regularization of MMD GANs), which was not considered by Lin et al. (2021).

Note that a similar derivation can be carried out for IPMs, such as the Wasserstein-1 distance; we detail the corresponding UOT dual derivation in Appendix D.1 and omit it here for brevity. The overall GWF framework is illustrated in Figure 3, and its practical implementation is detailed in Appendix H.

# 5. Experiments

In this section, we empirically support and extend the results developed throughout the paper. Our experiments are designed to illustrate the behavior of Generative Wasserstein Flow (GWF) under different divergence choices and regularization regimes.

- In Section 5.1, we compare two variational formulations of the Kullback–Leibler divergence: the classical $f$-divergence representation and the Donsker-Varadhan formulation introduced in Section 3.1.

- In Section 5.2, we study GWF with the squared MMD. We highlight the necessity of adversarial training in these models, and compare our approach with the unregularized counterpart, namely MMD-GAN.

- In Section 5.3, we investigate the role of regularization across a broad class of divergences, including

$f$-divergences, IPMs and squared MMD.

- In Appendix J.3, we report complementary experimental results, including an analysis of inner optimization accuracy, a comparison between forward and reverse KL formulations, and an evaluation of the computational cost of JKO, together with additional quantitative tables and qualitative sample visualizations.

We focus on image generation tasks, and conduct experiments on the MNIST ($1 \times 28 \times 28$) and the CIFAR-10 datasets ($3 \times 32 \times 32$) (Krizhevsky & Hinton, 2009). We use four different architectures throughout our experiments: *U-Net*, *Large-Net*, *Small-Net*, and *ResNetMMD*, for both generators and critics. Across all experiments, model performance is evaluated using the Fréchet Inception Distance (FID) (Heusel et al., 2017). Unless stated otherwise, we use the dimension-normalized quadratic transport cost $c(x,y) = \|x - y\|_2^2/d$. All experiments are conducted on a single NVIDIA H100 SXM5 GPU. Architectural details, as well as additional implementation details and hyperparameters, are provided in Appendix I[3].

## 5.1. Donsker-Varadhan Formulation

We study whether the Donsker–Varadhan (DV) formulation of the KL divergence (11) improves the empirical performance of GWF. Although its Monte Carlo estimator is biased due to the logarithm, it optimizes a larger variational lower bound. We therefore evaluate whether this bias nonetheless leads to improved sample quality in practice.

We conduct the experiments with the *U-Net*, *Large-Net*, and *Small-Net*. We fix the JKO step size to $\tau = 0.5$ for the *Small-Net*, and $\tau = 0.2$ for the other architectures. At each JKO step, we perform $N = 2000$ inner optimization steps to train the *Large-Net*, and $N = 1000$ for the other architectures. Figure 1 reports the evolution of the FID as a function of the JKO iteration for each architecture and for both the DV and classical variational formulation of the KL (5). In addition, the final FID values are summarized in Table 3. Overall, the DV formulation leads to improved or comparable FID values across architectures throughout the JKO trajectory. These results suggest that the DV formulation constitutes a competitive alternative to the classical variational KL formulation for generative modeling.

## 5.2. Squared MMD JKO GAN

We now investigate GWFs with the squared MMD, introduced at the end of Section 4. In all experiments, we use a weighted sum of kernels whose weights are learned during training, following the approach of Zhang et al. (2025), see Appendix I for more details. As discussed in Section 4,

---

[3]Code available at https://github.com/Paulcauch/Generative_Wasserstein_Flows

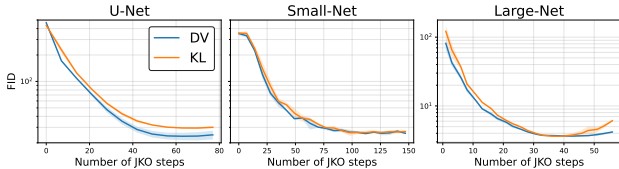

*Figure 1.* FID versus JKO iteration on CIFAR-10, comparing the classical KL formulation with the Donsker–Varadhan (DV) formulation across three network architectures.

training with Algorithm 1 leads to poor sample quality; additional experiments and details are provided in Appendix J.2. As a consequence, all subsequent experiments rely on a specific instantiation of our framework, namely Algorithm 2, based on (18) and derived in Appendix D.4. This algorithm can be interpreted as a JKO-regularized version of the classical MMD-GAN.

We consider several variants of MMD-based adversarial losses commonly studied in the literature: *ckMMDGAN* (Zhang et al., 2025) and *sMMDGAN* (Arbel et al., 2018). These variants modify the standard MMD objective to improve performance, see Appendix H for further details. Our goal is to assess whether the proposed JKO-regularized formulation improves over the corresponding unregularized MMD-GAN baselines. We conduct experiments on MNIST and CIFAR-10, fixing the JKO step size to $\tau = 0.5$ and running the algorithm for $K = 140$ JKO steps on CIFAR-10 and $K = 100$ on MNIST, with $N = 1000$ optimization steps at each JKO iteration. In the unregularized setting which corresponds to the original underlying GANs, where no JKO regularization is used, we use the same total number of parameter updates, namely $K \times N$. We use the *ResNet-MMD* architecture for MNIST and *Small-Net* for CIFAR-10. Quantitative results are summarized in terms of FID in Table 1. On MNIST, JKO consistently leads to lower FID across all losses. On CIFAR-10, JKO yields clear gains for MMD and sMMD, while having a more limited effect for the ckMMD loss. Overall, these results suggest that JKO regularization generally improves or stabilizes MMD-based adversarial training.

### 5.3. GWF for Various Divergences

We now extend our experimental study to a broader class of divergences, including $f$-divergences, IPMs and squared MMD. For $f$-divergences, we consider the variational formulation of KL, Jensen-Shannon and $\chi^2$ divergences, as well as the DV formulation of KL, following the framework discussed in Section 3. Following Section 4, we consider the Wasserstein-1 ($W_1$) distance as an IPM, and the $\mathrm{MMD}^2$. The corresponding losses, as well as a generic training algorithm, are detailed in Appendix H. Our objective is to assess the need for JKO regularization for each divergence, extending experiments done in (Fan et al., 2022; Choi et al., 2024) for $f$-divergences, and in (Lin et al., 2021) for GANs.

*Table 1.* FID for different MMD losses with or without JKO regularization across MNIST and CIFAR10 (averaged over 3 runs).

| Loss | MNIST | | CIFAR-10 | |
| --- | --- | --- | --- | --- |
| | no JKO | JKO | no JKO | JKO |
| ckMMD | $2.173 \pm 0.145$ | $\mathbf{1.971 \pm 0.079}$ | $13.96 \pm 0.56$ | $\mathbf{12.78 \pm 0.88}$ |
| sMMD | $1.464 \pm 0.092$ | $\mathbf{1.324 \pm 0.070}$ | $59.79 \pm 7.79$ | $\mathbf{26.79 \pm 0.23}$ |
| MMD | $1.028 \pm 0.110$ | $\mathbf{0.983 \pm 0.026}$ | $49.39 \pm 13.48$ | $\mathbf{15.76 \pm 0.08}$ |

Note that these experiments were not designed to achieve the best possible performance, but rather to isolate and study the effect of JKO regularization under a unified experimental setup. To ensure a fair comparison across divergences, we use the same architecture (*Small-Net*) and comparable hyperparameters for all methods, which allows us to attribute performance differences to the optimization scheme rather than to model capacity.

We compare several JKO step sizes, namely $\tau \in \{0.01, 1, 100\}$, as well as a baseline without JKO regularization. The latter corresponds to an infinite-step-size limit and recovers standard $f$-GAN training for the reverse $f$-divergence (*e.g.*, KL yields the reverse-KL $f$-GAN), while for symmetric divergences such as $W_1$ and $\mathrm{MMD}^2$, it coincides with WGAN and MMDGAN.

The resulting training dynamics on CIFAR-10, measured in terms of FID as a function of the JKO iteration, are reported in Figure 2, and summarized quantitatively in Table 4. We recall that for all methods, we use the same number of generator updates (see end of Section 5.2). Overall, JKO regularization improves training across most divergences, provided that the step size is chosen appropriately. Very small step sizes ($\tau = 0.01$) often lead to slow or unstable convergence, while excessively large step sizes ($\tau = 100$) tend to degrade performance, as the objective approaches the unregularized regime. Compared to the results of Section 5.2, which relied on a different architecture, the benefits of JKO regularization for MMD-based objectives (MMD, ckMMD, sMMD) are much more clearly visible in this unified experimental setting, with improvements of up to 20 FID points in the MMD case. For $f$-divergences, JKO yields moderate yet consistent improvements. In the case of $W_1$, small JKO step sizes significantly stabilize training compared to the standard WGAN objective. These results highlight the importance of tuning $\tau$, and show that the effectiveness of JKO regularization depends on both the divergence and the neural network architecture.

### 5.4. Practical Insights and Limitations

We summarize here several practical insights and limitations observed during our experiments with the GWF framework.

**Effect of the JKO step size.** The step size $\tau$ controls the trade-off between stability and convergence speed and therefore requires careful tuning in practice. Its optimal value depends on the divergence, architecture, and dataset, making the choice of $\tau$ problem-specific rather than universal

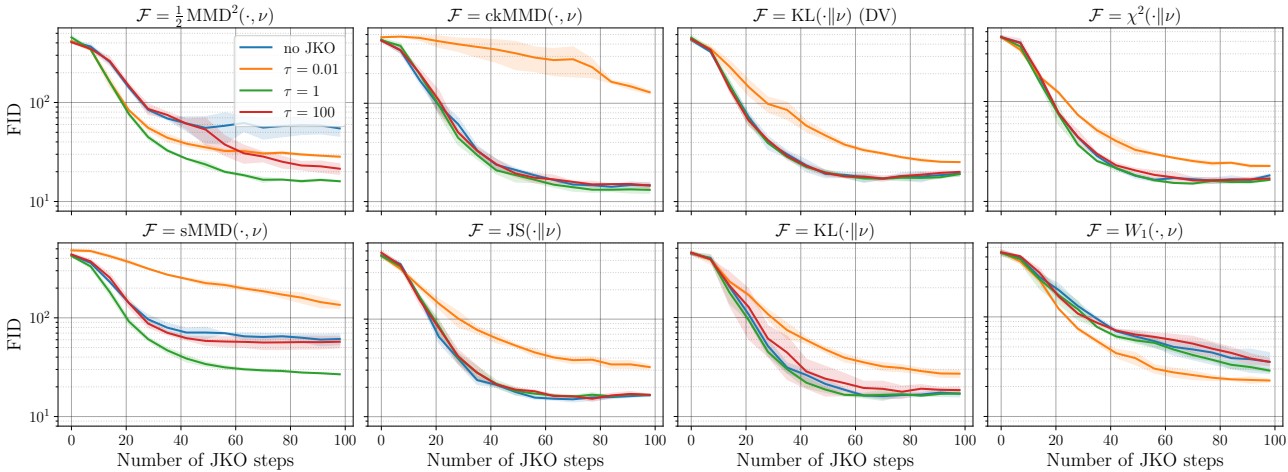

*Figure 2.* Impact of JKO regularization on training dynamics for various divergences and step sizes $\tau$ on CIFAR-10. All experiments use the same architecture (*Small-Net*) and comparable hyperparameters, and are averaged over 3 runs.

(see Figure 2).

**Inner optimization steps.** The number of generator and critic updates per JKO step affects both performance and computational cost. A moderate number of inner iterations is typically sufficient: too many increase cost without clear gains, while too few degrade the quality of the subproblem solution (see Figure 6).

**Reverse divergences.** We did not observe major convergence differences between opposite orientations of a divergence. For example, in the case of forward and reverse KL, the usual mode-seeking behavior associated with $\mathrm{KL}(\mu\|\nu)$ appears mitigated in practice (see Figure 8), likely because the variational formulations depend only on expectations and are therefore less sensitive to support mismatch.

**Adversarial optimization.** As in standard GAN training, insufficient critic regularization leads to unstable training across divergences. More generally, the reliance on adversarial optimization makes the method sensitive to hyperparameter tuning and initialization compared to alternative paradigms such as score-based models.

**Cost–benefit trade-off and scope of the method.** Introducing JKO regularization increases computational cost due to the additional proximal term and inner optimization steps. In our experiments, this overhead remained moderate and does not depend on the architecture (approximately $8.6\%$ on average; see Tables 6 and 7), but its benefit is not uniform across divergences. However, when $\tau$ is properly chosen, it does not degrade the original performance.

## 6. Theoretical View of Parametric Flows

In this section, we study the dynamics induced by the parametric schemes considered throughout the paper when the optimization is restricted to a family of pushforward mea-

sures. Our goal is to clarify how these dynamics relate to Wasserstein gradient flows, and to show that the parametric JKO scheme used in practice can be interpreted, in the small-step regime, as a projected Wasserstein gradient flow.

Consider the optimization problem

$$\min_{\theta \in \mathbb{R}^p} \mathcal{F}(\mu_\theta), \qquad \mu_\theta = (F_\theta)_\#\mu, \qquad (19)$$

where $\mu \in \mathcal{P}_2(\mathbb{R}^d)$ is a given source distribution and $F_\theta : \mathbb{R}^d \to \mathbb{R}^d$ is a measurable map parametrized by $\theta \in \mathbb{R}^p$. This formulation encompasses generative methods based on direct divergence minimization, such as normalizing flows, variational autoencoders, or GANs assuming a perfect discriminator, as well as the JKO-based schemes parametrized by neural networks introduced in the previous sections. Furthermore, throughout this section, we assume that $F_\theta$ is invertible for all $\theta$, and for all $x \in \mathbb{R}^d$, $\theta \mapsto F_\theta(x)$ is differentiable.

**Parametric Flows.** Let $(\theta_t)_{t\geq 0}$ be a parameter trajectory. We are interested in studying the induced dynamic on the space of probability distributions, through the lens of a continuity equation

$$\partial_t \mu_{\theta_t} + \nabla \cdot (\mu_{\theta_t} v_{\theta_t}) = 0, \qquad (20)$$

with $v_{\theta_t} : \mathbb{R}^d \to \mathbb{R}^d$ a parametric vector field, describing the evolution of $\mu_{\theta_t}$ through time. Let $x_t = F_{\theta_t}(z)$ for $z \sim \mu$, so that $x_t \sim \mu_{\theta_t}$. Since $F_{\theta_t}$ is invertible, particles evolve according to $\dot{x}_t = \partial_\theta F_{\theta_t}(F_{\theta_t}^{-1}(x_t))\dot{\theta}_t$ where $\partial_\theta F_\theta(\cdot) \in \mathbb{R}^{d\times p}$ denotes the Jacobian of $\theta \mapsto F_\theta(\cdot)$. The induced velocity field is therefore $v_{\theta_t}(x) = \partial_\theta F_{\theta_t}(F_{\theta_t}^{-1}(x))\dot{\theta}_t$. Define the operator $\mathcal{G}_\theta : L^2(\mu_\theta) \to \mathbb{R}^p$ as, for $f \in L^2(\mu_\theta)$,

$$\mathcal{G}_\theta(f) = \int \partial_\theta F_\theta(z)^\top f(F_\theta(z)) \, \mathrm{d}\mu(z) \in \mathbb{R}^p. \qquad (21)$$

**Proposition 6.1.** *Assume* $\theta \mapsto \mathcal{F}(\mu_\theta)$ *differentiable and consider* $\dot{\theta}_t = -\nabla_\theta \mathcal{F}(\mu_{\theta_t})$. *Define the operator* $\mathcal{H}_\theta : L^2(\mu_\theta) \to L^2(\mu_\theta)$ *for any* $f \in L^2(\mu_\theta)$, $x \in \mathbb{R}^d$ *by*

$$\left[\mathcal{H}_\theta f\right](x) := \partial_\theta F_\theta\big(F_\theta^{-1}(x)\big) \mathcal{G}_\theta(f). \quad (22)$$

*Let* $v_{\theta_t} := -\mathcal{H}_{\theta_t} \nabla_{W_2} \mathcal{F}(\mu_{\theta_t})$, *then* $v_{\theta_t}$ *satisfies* (20).

Hence, when the parameter dynamics follow the Euclidean flow, the induced evolution at the distribution level can be interpreted as a WGF preconditioned by the operator $\mathcal{H}_\theta$.

**Projected flows.** The operator $\mathcal{H}_\theta$ is in general not a projection operator, since typically $\mathcal{H}_\theta \circ \mathcal{H}_\theta \neq \mathcal{H}_\theta$ (see Proposition G.2). We now show that a projection structure can be recovered by endowing the parameter space with the metric induced by the parametrization. More precisely, for a smooth curve $(\theta_t)_t$, the squared $L^2(\mu_{\theta_t})$ norm of the induced velocity field is given by $\|v_{\theta_t}\|_{L^2(\mu_{\theta_t})}^2 = \dot{\theta}_t^\top G(\theta_t) \dot{\theta}_t$, where $G(\theta) = \int \partial_\theta F_\theta(z)^\top \partial_\theta F_\theta(z) \, d\mu(z) \in \mathbb{R}^{p \times p}$. This defines a Riemannian metric on $\mathbb{R}^p$, provided that $G(\theta)$ is invertible. We refer to Appendix G.1 for the precise geometric derivations. In the following, we state results under this invertibility assumption, as in (Zuo et al., 2025; Dumont et al., 2026). In practice, however, $G(\theta)$ may fail to be invertible, especially for overparameterized neural networks, in which case one may replace $G(\theta)^{-1}$ by the Moore–Penrose pseudoinverse, as in (Jin et al., 2025).

**Proposition 6.2.** *Assume* $\theta \mapsto \mathcal{F}(\mu_\theta)$ *differentiable and consider* $\dot{\theta}_t = -G(\theta_t)^{-1} \nabla_\theta \mathcal{F}(\mu_{\theta_t})$. *Define the operator* $\tilde{\mathcal{H}}_\theta : L^2(\mu_\theta) \to L^2(\mu_\theta)$ *for any* $f \in L^2(\mu_\theta)$, $x \in \mathbb{R}^d$ *by*

$$[\tilde{\mathcal{H}}_\theta f](x) = \partial_\theta F_\theta\big(F_\theta^{-1}(x)\big) G(\theta)^{-1} \mathcal{G}_\theta(f). \quad (23)$$

*Then* $\tilde{\mathcal{H}}_\theta$ *defines an orthogonal projector from* $L^2(\mu_\theta)$ *to the subspace* $V_\theta = \{x \mapsto \partial_\theta F_\theta\big(F_\theta^{-1}(x)\big)u, \ u \in \mathbb{R}^p\}$.

*Furthermore, if we define* $v_{\theta_t} := -\tilde{\mathcal{H}}_{\theta_t} \nabla_{W_2} \mathcal{F}(\mu_{\theta_t})$, *then* $v_{\theta_t}$ *satisfies* (20).

Therefore, when the parameter dynamics follow the preconditioned flow

$$\dot{\theta}_t = -G(\theta_t)^{-1} \nabla_\theta \mathcal{F}(\mu_{\theta_t}), \quad (24)$$

the induced evolution at the distribution level can be interpreted as a projected WGF on the subspace of realizable velocity fields.

Note that our preconditioned flow (24) does not correspond in general to the Wasserstein natural gradient flow on distributions as introduced in (Li & Zhao, 2019; Chen & Li, 2020); see Appendix G.2 for further details. Yet, they do coincide in some cases (see Proposition G.3), for example in 1D or for some linear maps and Gaussian source distribution, *i.e.* $F_\theta(x) = Ax + m$, where $A = \sigma I_d$ or $A$

positive diagonal. Then, if the source distribution is Gaussian, *e.g.* $\mu = \mathcal{N}(0, I_d)$ the induced distributions along the flow remain Gaussian with isotropic or diagonal covariance matrices, respectively, and the flow also coincides with the Bures-Wasserstein gradient flow (Lambert et al., 2022), see Appendix G.4.

Next, we show below that the parametrized JKO scheme with reparameterization trick used in previous sections,

$$\theta_{\ell+1} \in \underset{\theta}{\arg\min} \ \mathcal{F}\big((F_\theta)_\# \mu\big) + \frac{1}{2\tau} \|F_\theta - F_{\theta_\ell}\|_{L^2(\mu)}^2, \quad (25)$$

is closely connected to the preconditioned flow (24).

**Proposition 6.3.** *Assume that* $\theta \mapsto F_\theta$ *is sufficiently regular as a map into* $L^2(\mu_\theta)$, *and that* $\theta \mapsto \mathcal{F}(\mu_\theta)$ *is differentiable. Let* $\theta_{\ell+1}$ *be defined by* (25). *Assume that* $\theta_{\ell+1} - \theta_\ell = \mathcal{O}(\tau)$ *as* $\tau \to 0$. *Then, as* $\tau \to 0$, *we have*

$$G(\theta_{\ell+1}) \frac{\theta_{\ell+1} - \theta_\ell}{\tau} = -\nabla_\theta \mathcal{F}(\mu_{\theta_{\ell+1}}) + o(1). \quad (26)$$

*Thus, the parametric JKO update is, at first order, an implicit discretization of the preconditioned gradient flow* (24).

A detailed proof of this result, as well as assumptions on $\theta \mapsto F_\theta$ such that the sequence of iterates $(\theta_\ell)_\ell$ are guaranteed to behave appropriately, are provided in Appendix G.3. As a consequence, solving the reparametrized JKO scheme (25) in practice is not merely performing a Euclidean descent in parameter space. In the small-step regime $\tau \to 0$, the induced dynamics are instead consistent, at first order, with the preconditioned flow (24). In that sense, the proximal structure of the JKO update implicitly introduces a preconditioning, even though the inverse of the matrix $G(\theta)$ is never formed explicitly in practice. This is appealing, since the latter would be intractable for a large neural network. In Appendix J.1, we empirically assess whether reparametrized JKO (25) remains close to the flow (24), as suggested by Proposition 6.3, even when its assumptions are not fully satisfied.

## 7. Conclusion

This work unifies several generative modeling methods based on the JKO scheme and extends this framework to a large class of divergences which include $f$-divergences, IPMs, and squared MMDs. Moreover, we investigated empirically the benefits of the JKO regularization on a wide range of these objectives, showing that it benefits a lot for MMD and $W_1$ objectives, and more moderately for $f$-divergences. We also propose a preliminary theoretical framework for parametric flows, which correspond to the schemes computed in practice. Our assumptions on the map $F$ are rather strong, hence we plan for future works to alleviate them, with the goal to study the convergence of these flows for more realistic neural networks.

## Acknowledgements

We thank the anonymous reviewers for their valuable comments. This work was granted access to the HPC resources of IDRIS under the allocation 2025-AD011016536 made by GENCI. CB was partly funded by the Agence nationale de la recherche, through the PEPR PDE-AI project (ANR-23-PEIA-0004). PC and AK were funded by the European Union (ERC, Optinfinite, 101201229). The views and opinions expressed are, however, of the author(s) only and do not necessarily reflect those of the European Union or the European Research Council Executive Agency. Neither the European Union nor the granting authority can be held responsible for them.

## Impact Statement

This paper presents work whose goal is to advance the field of Machine Learning. The proposed methods are primarily theoretical and do not directly involve human subjects or sensitive data. As with other generative modeling methods, improper use of learned models in downstream applications could lead to unintended consequences. We do not anticipate societal impacts beyond those commonly associated with generative modeling techniques.

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

# Appendix

The appendix is organized as follows. In Appendix A, we review related work. Appendix B provides background on optimal transport. Appendix C reviews variational formulations of several $f$-divergences, introduces the Donsker-Varadhan representation of the KL divergence, and clarifies connection between generative schemes. In Appendix D, we derive the objectives for IPM regularization as well as for squared MMD objective, and provide the full JKO MMD GAN algorithm. In Appendix E, we provide a schematic summary of the unifying results introduced in the paper. Proofs of the main theoretical results are gathered in Appendix F. Appendix G provides a deeper analysis of parametric Wasserstein flows, with explicit computations in the Bures–Wasserstein setting. In Appendices H and I, we detail the general GWF algorithm and practical implementation aspects of the proposed framework. Finally, in Appendix J, we report additional experimental results covering both theoretical and practical aspects.

## Contents

## A. Related Work

**JKO Scheme.**    The JKO scheme has been introduced by Jordan et al. (1998) and has been shown to be a time discretization of the Wasserstein gradient flow. Thus, it motivated several works to try to compute it in practice to be able to approximate Wasserstein gradient flows of complex functionals. However, its implementation in dimension $d \geq 2$ has been limited by the difficulty to compute the Wasserstein distance in high dimension. Several strategies have been proposed. For instance, Benamou et al. (2016b) leveraged Brenier's theorem to reframe the problem as an optimization problem over convex functions and discretize the space of convex functions. Peyré (2015) instead added an entropic regularization to approximate the JKO scheme, and Benamou et al. (2016a) used the dynamic formulation of optimal transport.

**JKO with Neural Networks.**    More recent works propose to approximate the JKO scheme with neural networks, which allowed to use it for generative modeling. More precisely, Mokrov et al. (2021); Alvarez-Melis et al. (2022) simultaneously proposed to learn the OT map at each JKO step with ICNNs. Then, Fan et al. (2022) used the variational formulation of $f$-divergences to avoid the need to evaluate the density of the current distribution and of the data distribution, and to perform generative modeling. Choi et al. (2024) improved the method of Fan et al. (2022) by adding a reparametrization trick which allows to learn a map from the source distribution. Some works also proposed to use neural networks to learn the JKO scheme in other geometries, *e.g.* in the space of probability distributions endowed with the Sliced-Wasserstein distance (Bonet et al., 2022), which is easier to compute, or endowed with the KL divergence (Yao et al., 2024).

Instead of parametrizing the map, Park et al. (2023) parametrized the density with neural networks to solve the JKO scheme. Other works parametrize the velocity field and induce from it a map pushing the distributions. This is the case for instance of Continuous Normalizing Flows regularized with the Wasserstein distance under the dynamic formulation (Finlay et al., 2020; Onken et al., 2021; Vidal et al., 2023; Lee et al., 2024), whose objective is the JKO scheme with KL divergence. Xu et al. (2023); Cheng et al. (2024) also recently proposed to solve the JKO scheme by learning the velocity field of each JKO step. However, they did not use the dynamic formulation of optimal transport, and optimize the KL divergence with respect to the noise, starting from $\mu_0 = \nu$ in the same spirit of diffusion models.

**GANs.**    Originally, GANs do not approximate Wasserstein gradient flows. But Lin et al. (2021) proposed to regularize them with the $L^2$ distance between the last map and the new map, which is equivalent to the JKO scheme up to the reparametrization trick. We also note that Yi et al. (2023) made a connection between the training step of the generator of $f$-GANs and the forward Wasserstein gradient descent of a well chosen objective.

**Wasserstein Gradient Flows of MMD.**    Altekrüger et al. (2023) proposed to solve the JKO scheme with MMD objectives, for which the existence of the OT map at each step is not guaranteed. Hence, they proposed to disintegrate the coupling with respect to the first marginal $\mu_\ell$ and parametrize the resulting kernel with a neural network, hence allowing to learn a plan. Galashov et al. (2025) proposed a method for generative modeling based on the Wasserstein gradient flow of the MMD (Arbel et al., 2019) and leveraging ideas of diffusion models, which does not rely on adversarial training.

**Parametric Wasserstein Flows.** Natural gradient flows are gradient flows preconditioned by a Riemannian metric reflecting the structure of the parametric space. In general, they can be written as $\dot{\theta}_t = -G(\theta_t)^{-1}\nabla_\theta L(\theta_t)$ for $L$ some loss and $G(\theta)$ the metric. Originally, the metric is induced by the KL divergence, and corresponds to the Fisher information metric. However, several works proposed to instead use the Wasserstein information matrix (Li & Zhao, 2019), which leads to the Wasserstein Natural Gradient Flow (Li & Montúfar, 2018; Li & Zhao, 2019; Chen & Li, 2020). Li et al. (2019) studied this scheme for a KL divergence objective and a parametric parametrization in the 1D and Gaussian case. Computing the metric tensor in general is computationally heavy, hence it was proposed to approximate it using kernels (Arbel et al., 2020; Moskovitz et al., 2021) or to use surrogate (Lin et al., 2021). In particular, the surrogate considered in Section 6 with $G(\theta) = \int \partial_\theta F_\theta(z)^\top \partial_\theta F_\theta(z)\,\mathrm{d}\mu(z)$ coincides in one dimension with the Wasserstein metric, and was studied by Zuo et al. (2025) with $F_\theta$ chosen as neural networks in this setting, while Jin et al. (2025) considered higher dimensional spaces also with $F_\theta$ chosen as neural networks. More recently, Dumont et al. (2026) considered the problem of minimization over $L^2(\mu)$ using an implicit scheme similar to (25), and also made connections with natural gradient flows.

# B. Background on Optimal Transport

## B.1. Optimal Transport

Let $c : \mathbb{R}^d \times \mathbb{R}^d \to \mathbb{R}$ be a cost function. The Kantorovich problem between $\mu, \nu \in \mathcal{P}_2(\mathbb{R}^d)$ is defined as

$$\mathrm{OT}_c(\mu,\nu) = \inf_{\pi \in \Pi(\mu,\nu)} \int c(x,y)\,\mathrm{d}\pi(x,y), \tag{27}$$

where $\Pi(\mu,\nu) = \{\gamma \in \mathcal{P}_2(\mathbb{R}^d \times \mathbb{R}^d),\ \pi^1_\# \gamma = \mu,\ \pi^2_\# \gamma = \nu\}$ is the set of couplings between $\mu$ and $\nu$, $\pi^1 : (x,y) \mapsto x$ and $\pi^2 : (x,y) \mapsto y$. This problem admits a dual of the form

$$\mathrm{OT}_c(\mu,\nu) = \sup_{(f,g) \in \Phi_c} \int f\,\mathrm{d}\mu + \int g\,\mathrm{d}\nu, \tag{28}$$

with $\Phi_c = \{(f,g) \in L^1(\mu) \times L^1(\nu),\ f(x) + g(y) \le c(x,y) \text{ for } \mu \otimes \nu\text{-a.e. } (x,y)\}$, $L^1(\mu) = \{f : \mathbb{R}^d \to \mathbb{R},\ \int |f|\,\mathrm{d}\mu < \infty\}$. For $g : \mathbb{R}^d \to \mathbb{R}$, define its $c$-transform as $g^c(x) = \inf_{y \in \mathbb{R}^d} c(x,y) - g(y)$. $f \in L^1(\mu)$ is said to be $c$-concave if there exists $g : \mathbb{R}^d \to \mathbb{R}$ such that $f = g^c$. Note that by definition of $g^c$, the pair $(g^c, g)$ always satisfies the constraint in (28), i.e., for all $x, y$, $g^c(x) + g(y) \le c(x,y)$. In particular, (28) is equivalent to the semi-dual (Villani et al., 2009, Theorem 5.10)

$$\mathrm{OT}_c(\mu,\nu) = \sup_{g \in L^1(\nu)} \int g^c\,\mathrm{d}\mu + \int g\,\mathrm{d}\nu. \tag{29}$$

**Wasserstein distance.** An important particular case is obtained for $c(x,y) = \frac{1}{2}\|x-y\|_2^2$ for which the value of the infimum of the problem defines a distance, referred to as the Wasserstein distance, i.e.

$$\frac{1}{2}\mathrm{W}_2^2(\mu,\nu) = \inf_{\pi \in \Pi(\mu,\nu)} \frac{1}{2}\int \|x-y\|_2^2\,\mathrm{d}\pi(x,y). \tag{30}$$

Note that the Wasserstein distance is equivalent to the OT problem with cost $c(x,y) = -\langle x,y\rangle$ as

$$\frac{1}{2}\mathrm{W}_2^2(\mu,\nu) = \int \frac{\|x\|_2^2}{2}\,\mathrm{d}\mu(x) + \int \frac{\|y\|_2^2}{2}\,\mathrm{d}\nu(y) + \inf_{\pi \in \Pi(\mu,\nu)} \int -\langle x,y\rangle\,\mathrm{d}\pi(x,y). \tag{31}$$

Moreover, for any $f \in L^1(\mu), g \in L^1(\nu)$ satisfying the dual constraint, we have the equivalence

$$f(x) + g(y) \le \frac{1}{2}\|x-y\|_2^2 \iff f(x) - \frac{1}{2}\|x\|_2^2 + g(y) - \frac{1}{2}\|y\|_2^2 \le -\langle x,y\rangle. \tag{32}$$

Hence, up to the reparametrization $\varphi = \frac{1}{2}\|\cdot\|_2^2 - f$, $\psi = \frac{1}{2}\|\cdot\|_2^2 - g$, we also have

$$\frac{1}{2}\mathrm{W}_2^2(\mu,\nu) = \int \frac{\|x\|_2^2}{2}\,\mathrm{d}\mu(x) + \int \frac{\|y\|_2^2}{2}\,\mathrm{d}\nu(y) - \inf_{\varphi(x)+\psi(y)\ge\langle x,y\rangle\ \forall x,y} \int \varphi\,\mathrm{d}\mu + \int \psi\,\mathrm{d}\nu, \tag{33}$$

which admits as semi-dual representation (Villani, 2003, Theorem 2.9)

$$\frac{1}{2}\mathrm{W}_2^2(\mu,\nu) = \int \frac{\|x\|_2^2}{2}\,\mathrm{d}\mu(x) + \int \frac{\|y\|_2^2}{2}\,\mathrm{d}\nu(y) - \inf_{\varphi \text{ convex}} \int \varphi\,\mathrm{d}\mu + \int \varphi^*\,\mathrm{d}\nu, \tag{34}$$

with $f^*(y) = \sup_x \langle x,y\rangle - f(x)$ the Legendre transform.

**OT map.** Let $\pi \in \Pi_o(\mu, \nu)$ be an optimal coupling between $\mu$ and $\nu$, which minimizes (27). Then, for all $(x, y) \in \mathrm{spt}(\pi)$, we have that the dual constraints are saturated, *i.e.* for $\pi$-almost every $(x, y)$,

$$g^c(x) + g(y) = c(x, y). \tag{35}$$

If $\pi = (\mathrm{Id}, \mathrm{T})_{\#}\mu$ with $\mathrm{T} : \mathbb{R}^d \to \mathbb{R}^d$ such that $\mathrm{T}_{\#}\mu = \nu$, then we have for $\mu$-almost every $x$,

$$g^c(x) + g\big(\mathrm{T}(x)\big) = c\big(x, \mathrm{T}(x)\big), \tag{36}$$

which implies that

$$\mathrm{T}(x) \in \operatorname*{argmin}_y \; c(x, y) - g(y) - g^c(x) = \operatorname*{argmin}_y \; c(x, y) - g(y). \tag{37}$$

Assume now $\mu, \nu \in \mathcal{P}_{\mathrm{ac}, 2}(\mathbb{R}^d)$, and take $c(x, y) = \frac{1}{2}\|x - y\|_2^2$. In this case, by (34), the optimal potentials $y \mapsto \frac{1}{2}\|y\|_2^2 - g(y)$ and $x \mapsto \frac{1}{2}\|x\|_2^2 - g^c(x)$ are both convex, and we have

$$g^c(x) = \min_y \; \tfrac{1}{2}\|x - y\|_2^2 - g(y) = \tfrac{1}{2}\|x\|_2^2 - \max_y \; \langle x, y \rangle - \big(\tfrac{1}{2}\|y\|_2^2 - g(y)\big). \tag{38}$$

Moreover, on one hand, $x \mapsto \langle x, y \rangle$ is convex. On the other hand, $y \mapsto \frac{1}{2}\|y\|_2^2 - g(y)$ is convex, thus $y \mapsto \langle x, y \rangle - \big(\frac{1}{2}\|y\|_2^2 - g(y)\big)$ is concave. Hence, applying Danskin's theorem (Blondel & Roulet, 2024, Theorem 11.1), we get that $g^c$ is differentiable and we recover the well know formula

$$\nabla g^c(x) = x - \mathrm{T}(x) \iff \mathrm{T}(x) = x - \nabla g^c(x), \tag{39}$$

which also states that the OT map $\mathrm{T}$ is the gradient of the convex function $x \mapsto \frac{1}{2}\|x\|_2^2 - g^c(x)$.

Note that for $h : \mathbb{R}^d \to \mathbb{R}$ strictly convex and $c(x, y) = h(x - y)$, then we have that $\mathrm{T}(x) = x - (\nabla h)^{-1}\big(\nabla g^c(x)\big)$ (Santambrogio, 2015, Theorem 1.17).

## B.2. Unbalanced Optimal Transport

The Unbalanced Optimal Transport problem (Liero et al., 2018; Chizat et al., 2018; Séjourné et al., 2023) relaxes the hard marginal constraints by penalizing deviations with divergences. Denoting by $\mathcal{M}_+(\mathbb{R}^d)$ the space of positive measures over $\mathbb{R}^d$, this problem can be formalized as, for $\mu, \nu \in \mathcal{M}_+(\mathbb{R}^d)$,

$$\mathrm{UOT}_c(\mu, \nu) = \inf_{\gamma \in \mathcal{M}_+(\mathbb{R}^d \times \mathbb{R}^d)} \int c(x, y) \, \mathrm{d}\gamma(x, y) + \lambda_1 D(\pi^1_{\#}\gamma, \mu) + \lambda_2 D(\pi^2_{\#}\gamma, \nu), \tag{40}$$

for $D$ some divergence between positive measures, $\lambda_1, \lambda_2 > 0$. $D$ is often chosen as a $\varphi$-divergence (Séjourné et al., 2023), but other divergences can be chosen such as the MMD (Manupriya et al., 2024) or OT based distances (Mahey, 2024). In general, for $\gamma$ the optimal plan in (40), $\pi^1_{\#}\gamma \neq \mu$ and $\pi^2_{\#}\gamma \neq \nu$, but $\pi^1_{\#}\gamma$ and $\pi^2_{\#}\gamma$ have the same mass, *i.e.* $\gamma(\mathbb{R}^d \times \mathbb{R}^d) = \pi^1_{\#}\gamma(\mathbb{R}^d) = \pi^2_{\#}\gamma(\mathbb{R}^d)$. In particular, Eyring et al. (2024, Proposition 3.1) showed that for $D(\eta, \mu) = \mathrm{D}_f(\eta \| \mu)$, $\gamma$ is an optimal transport plan between $\pi^1_{\#}\gamma$ and $\pi^2_{\#}\gamma$, *i.e.*, it solves problem (27) between these measures. Moreover, for $c(x, y) = h(x - y)$ with $h$ strictly convex and $\mu$ absolutely continuous *w.r.t.* the Lebesgue measure, then $\gamma$ is unique, and induced by an OT map. Note also that (40) is equivalent to (Liero et al., 2018)

$$\mathrm{UOT}_c(\mu, \nu) = \inf_{\pi_1, \pi_2 \in \mathcal{M}_+(\mathbb{R}^d)} \mathrm{OT}_c(\pi_1, \pi_2) + \lambda_1 D(\pi_1, \mu) + \lambda_2 D(\pi_2, \nu), \tag{41}$$

optimizing directly over the marginals. This can be seen *e.g.* by contradiction. Indeed, if $\gamma$ is not an optimal coupling between the marginals of $\gamma$, then one can always find a better coupling between the two marginals, and hence $\gamma$ does not minimize the full objective.

For the case of $f$-divergence, this problem also admits for dual representation (Liero et al., 2018)

$$\mathrm{UOT}_c(\mu, \nu) = \sup_{(u, v) \in \Phi_c} - \int \lambda_1 f^*\big(-u(x)/\lambda_1\big) \, \mathrm{d}\mu(x) - \int \lambda_2 f^*\big(-v(y)/\lambda_2\big) \, \mathrm{d}\nu(y), \tag{42}$$

and the semi-dual

$$\mathrm{UOT}_c(\mu, \nu) = \sup_v \ - \int \lambda_1 f^*\big(-v^c(x)/\lambda_1\big)\, \mathrm{d}\mu(x) - \int \lambda_2 f^*\big(-v(y)/\lambda_2\big)\, \mathrm{d}\nu(y). \tag{43}$$

Taking $\lambda_1 \to \infty$, $\lambda_2 \to \infty$, (40) becomes equal to the OT problem (27) as it enforces hard constraints on the marginals. Taking only $\lambda_1 \to \infty$ enforces the first marginal of the plan $\gamma \in \mathcal{M}_+(\mathbb{R}^d \times \mathbb{R}^d)$ to be equal to $\mu$, and hence the mass of $\gamma$ is equal to the mass of $\mu$, $i.e.$ $\gamma(\mathbb{R}^d \times \mathbb{R}^d) = \mu(\mathbb{R}^d)$. We now restrict to $\mu \in \mathcal{P}_2(\mathbb{R}^d)$, then taking $\lambda_1 \to \infty$, we obtain the semi-relaxed UOT problem

$$\begin{aligned} \mathrm{sUOT}_c(\mu, \nu) &= \inf_{\gamma \in \mathcal{P}_2(\mathbb{R}^d \times \mathbb{R}^d),\, \pi^1_\# \gamma = \mu} \int c(x,y)\, \mathrm{d}\gamma(x,y) + \lambda_2 D(\pi^2_\# \gamma, \nu) \\ &= \inf_{\pi_2 \in \mathcal{P}_2(\mathbb{R}^d)} \mathrm{OT}_c(\mu, \pi_2) + \lambda_2 D(\pi_2, \nu). \end{aligned} \tag{44}$$

For $\mathrm{D}(\eta, \mu) = \mathrm{D}_f(\eta \| \mu)$, it also admits the (semi-)dual representation

$$\mathrm{sUOT}_c(\mu, \nu) = \sup_v \int v^c(x)\, \mathrm{d}\mu(x) - \int \lambda_2 f^*\big(-v(y)/\lambda_2\big)\, \mathrm{d}\nu(y). \tag{45}$$

By (Eyring et al., 2024, Proposition 3.1), noting $\gamma$ the optimal plan of (44) and $v$ the optimal potential of (45), $\pi^2_\# \gamma = (f^*)'(v) \cdot \nu$. Moreover, if $\mu \in \mathcal{P}_{\mathrm{ac},2}(\mathbb{R}^d)$ and $c(x,y) = h(x-y)$ with $h$ strictly convex, $\gamma = (\mathrm{Id}, \mathrm{T})_\# \mu$ where $\mathrm{T} = \mathrm{Id} - \nabla h^* \circ \nabla v$.

## B.3. Wasserstein Gradient

We recall the notion of Wasserstein gradient. For more details, we refer to (Ambrosio et al., 2008) or (Lanzetti et al., 2025).

**Definition B.1.** Let $\mathcal{F} : \mathcal{P}_2(\mathbb{R}^d) \to \mathbb{R}$. A Wasserstein gradient of $\mathcal{F}$ at $\mu \in \mathcal{P}_2(\mathbb{R}^d)$, if it exists, is defined as the map $\nabla_{\mathrm{W}_2} \mathcal{F}(\mu) \in L^2(\mu)$, which satisfies for any $\nu \in \mathcal{P}_2(\mathbb{R}^d)$, $\gamma \in \Pi_o(\mu, \nu)$,

$$\mathcal{F}(\nu) = \mathcal{F}(\mu) + \int \langle \nabla_{\mathrm{W}_2} \mathcal{F}(\mu)(x), y - x \rangle\, \mathrm{d}\gamma(x,y) + o\big(\mathrm{W}_2(\mu, \nu)\big). \tag{46}$$

The tangent space of $\mathcal{P}_2(\mathbb{R}^d)$ at $\mu \in \mathcal{P}_2(\mathbb{R}^d)$ is defined as

$$T_\mu \mathcal{P}_2(\mathbb{R}^d) = \overline{\{\nabla \psi,\ \psi \in C_c^\infty(\mathbb{R}^d)\}} \subset L^2(\mu), \tag{47}$$

where the closure is taken in $L^2(\mu)$, see $e.g.$ (Ambrosio et al., 2008, Definition 8.4.1). It is possible to show that $\nabla_{\mathrm{W}_2} \mathcal{F}(\mu) = \xi + \xi^\perp$ with $\xi \in T_\mu \mathcal{P}_2(\mathbb{R}^d)$, $\xi \in T_\mu \mathcal{P}_2(\mathbb{R}^d)^\perp$, and that $\int \langle \xi^\perp(x), y - x \rangle\, \mathrm{d}\gamma(x,y) = 0$. Moreover, $\xi$ is unique, see $e.g.$ (Lanzetti et al., 2025, Proposition 2.5). Hence, we can always restrict ourselves to $\nabla_{\mathrm{W}_2} \mathcal{F}(\mu) \in T_\mu \mathcal{P}_2(\mathbb{R}^d)$. Moreover, such Wasserstein gradient is a "strong Fréchet differential", implying that (46) also holds for non optimal couplings.

**Proposition B.2** (Proposition 2.12 in (Lanzetti et al., 2025)). *Let $\mu, \nu \in \mathcal{P}_2(\mathbb{R}^d)$, $\gamma \in \mathcal{P}_2(\mathbb{R}^d \times \mathbb{R}^d)$ any coupling and $\mathcal{F} : \mathcal{P}_2(\mathbb{R}^d) \to \mathbb{R}$ a Wasserstein differentiable functional at $\mu$ with Wasserstein gradient $\nabla_{\mathrm{W}_2} \mathcal{F}(\mu) \in T_\mu \mathcal{P}_2(\mathbb{R}^d)$. Then,*

$$\mathcal{F}(\nu) - \mathcal{F}(\mu) = \int \langle \nabla_{\mathrm{W}_2} \mathcal{F}(\mu)(x), y - x \rangle\, \mathrm{d}\gamma(x,y) + o\left(\sqrt{\int \|x - y\|_2^2\, \mathrm{d}\gamma(x,y)}\right). \tag{48}$$

Under regularity assumptions, the Wasserstein gradient of $\mathcal{F}$ can be computed in practice using the first variation $\mathcal{F}'(\mu)$ (Santambrogio, 2015, Definition 7.12), which is defined, if it exists, as the unique function (up to a constant) such that, for $\chi$ satisfying $\int \mathrm{d}\chi = 0$,

$$\frac{\mathrm{d}}{\mathrm{d}t} \mathcal{F}(\mu + t\chi)\Big|_{t=0} = \lim_{t \to 0} \frac{\mathcal{F}(\mu + t\chi) - \mathcal{F}(\mu)}{t} = \int \mathcal{F}'(\mu)\, \mathrm{d}\chi. \tag{49}$$

Then the Wasserstein gradient can be computed as $\nabla_{\mathrm{W}_2} \mathcal{F}(\mu) = \nabla \mathcal{F}'(\mu)$, see $e.g.$ (Chewi et al., 2024, Proposition 5.10).

# C. More details for JKO schemes with $f$-Divergences

In this section, we derive variational formulations for several $f$-divergences. We also introduce the Donsker–Varadhan representation of the KL divergence and highlight its advantages over the classical formulation. Finally, we discuss the method proposed by (Baptista et al., 2025) and show that it corresponds to GWF with the Donsker–Varadhan formula, up to a reparametrization.

## C.1. Variational Formulations of $f$-Divergences

We first recall the definition of $f$-divergences.

**Definition C.1** ($f$-divergence). Let $\mu, \nu \in \mathcal{P}_2(\mathbb{R}^d)$ such that $\mu \ll \nu$, and denote by $\frac{d\mu}{d\nu}$ the Radon–Nikodym derivative of $\mu$ with respect to $\nu$. Let $f : [0, \infty) \to \mathbb{R}$ be a convex function with $f(1) = 0$. The $f$-divergence between $\mu$ and $\nu$ is defined as:

$$D_f(\mu\|\nu) = \int f\left(\frac{d\mu}{d\nu}(x)\right) \, d\nu(x). \tag{50}$$

Given the choice of $f$, we can recover many well-known divergences. For instance, for $f(r) = r \log r$, we recover $D_f(\mu\|\nu) = \mathrm{KL}(\mu\|\nu)$, for $f(r) = -\log r$, we obtain $D_f(\mu\|\nu) = \mathrm{KL}(\nu\|\mu)$. We refer to (Polyanskiy & Wu, 2025, Chapter 7) for other examples. $f$-divergences admit the following lower bound.

**Lemma C.2.** *For all measurable functions $\phi : R^d \to \mathbb{R}$, we have:*

$$\int \phi \, d\mu - \int f^* \circ \phi \, d\nu \leq D_f(\mu\|\nu), \tag{51}$$

*where $f^*(y) = \sup_t\{yt - f(t)\}$ is the convex conjugate of $f$.*

*Proof.* Using the definition of $f^*$, for all $x \in \mathbb{R}^d$,

$$f^*\big(\phi(x)\big) = \sup_t \{\phi(x)t - f(t)\} \geq \phi(x) \cdot \frac{d\mu}{d\nu}(x) - f\left(\frac{d\mu}{d\nu}(x)\right), \tag{52}$$

so integrating both sides w.r.t. $\nu$ gives the desired inequality. $\qquad\square$

If we take the supremum in the previous lower bound, Theorem 7.6 of (Polyanskiy & Wu, 2025) guarantees that the supremum is attained and equal to the $f$-divergence:

$$D_f(\mu\|\nu) = \sup_\phi \left\{ \int \phi \, d\mu - \int f^* \circ \phi \, d\nu \right\}. \qquad \text{(Variational formulation)}$$

We now summarize the formula for some $f$-divergences of interest, that we will use in our applications.

**Reverse KL divergence.**
$$f(t) = t \log t, \quad f^*(t) = \sup_u\{tu - u \log u\} = \exp(t - 1). \tag{53}$$

Then:
$$\mathrm{KL}(\mu\|\nu) = \sup_\phi \left\{ \int \phi \, d\mu - \int \exp\big(\phi(y) - 1\big) \, d\nu(y) \right\}. \tag{54}$$

**$\chi^2$ divergence.**
$$f(t) = (t - 1)^2, \quad f^*(t) = \sup_x\{tx - (x - 1)^2\} = \frac{t^2}{4} + t. \tag{55}$$

Then:
$$\chi^2(\mu\|\nu) = \sup_\phi \left\{ \int \phi \, d\mu - \int \left(\frac{\phi^2}{4} + \phi\right) d\nu \right\} = \sup_g \left\{ \int 2g \, d\mu - 1 - \int g^2 \, d\nu \right\}. \tag{56}$$

for $\phi = 2g - 2$.

**Jensen–Shannon divergence (JS).**

$$f(t) = t \log \left( \frac{2t}{1+t} \right) + \log \left( \frac{2}{1+t} \right), \quad f^*(t) = -\log(2 - \exp(t)). \tag{57}$$

Then:

$$\mathrm{JS}(\mu \| \nu) = \sup_{\phi : \phi < \log 2} \left\{ \int \phi \, \mathrm{d}\mu + \int \log(2 - \exp(\phi)) \, \mathrm{d}\nu \right\}. \tag{58}$$

Or, if we let $g = \frac{1}{2} e^{\phi}$,

$$\mathrm{JS}(\mu \| \nu) = \sup_{g : \mathbb{R}^d \to [0,1]} \left\{ \log(2) + \int \log \circ g \, \mathrm{d}\mu + \int \log \left( 1 - g(y) \right) \, \mathrm{d}\nu(y) \right\}. \tag{59}$$

## C.2. Donsker-Varadhan Formulations

Let $\mu \ll \nu$, we recall the definition of the Kullback-Leibler divergence

$$\mathrm{KL}(\mu \| \nu) = \int \log \left( \frac{\mathrm{d}\mu}{\mathrm{d}\nu} \right) \mathrm{d}\mu. \tag{60}$$

Besides the variational lower bound of the previous section, the KL divergence admits another one, called the Donsker-Varadhan (DV) formula.

**Lemma C.3** (DV lower bound). *For all measurable functions $\phi : \mathbb{R}^d \to \mathbb{R}$, we have:*

$$\mathrm{KL}(\mu \| \nu) \geq \int \phi \, \mathrm{d}\mu - \log \int e^{\phi} \, \mathrm{d}\nu. \tag{61}$$

*Proof.* By Jensen's inequality we get:

$$\exp \left( \int \phi \, \mathrm{d}\mu - \mathrm{KL}(\mu \| \nu) \right) = \exp \left( \int \left( \phi - \log \left( \frac{\mathrm{d}\mu}{\mathrm{d}\nu} \right) \right) \mathrm{d}\mu \right) \leq \int \exp \left( \phi - \log \left( \frac{\mathrm{d}\mu}{\mathrm{d}\nu} \right) \right) \mathrm{d}\mu. \tag{62}$$

Now observe that:

$$\int \exp \left( \phi - \log \left( \frac{\mathrm{d}\mu}{\mathrm{d}\nu} \right) \right) \mathrm{d}\mu = \int \frac{e^{\phi}}{\frac{\mathrm{d}\mu}{\mathrm{d}\nu}} \, \mathrm{d}\mu = \int e^{\phi} \, \mathrm{d}\nu. \tag{63}$$

Applying the logarithm function to the previous Jensen's inequality hence implies the desired inequality. $\square$

Passing to the supremum in the RHS of Lemma C.3, which is achieved for $\phi = \log \circ \frac{\mathrm{d}\mu}{\mathrm{d}\nu}$, we get:

$$\mathrm{KL}(\mu \| \nu) = \sup_{\phi} \left\{ \int \phi \, \mathrm{d}\mu - \log \int e^{\phi} \, \mathrm{d}\nu \right\}. \qquad \text{(Donsker-Varadhan)}$$

**DV vs. Variational KL.** The Kullback–Leibler divergence admits both the Donsker–Varadhan (DV) and the variational representation via the convex conjugate of $f(t) = t \log t$. Both formulations are equivalent at optimality:

$$\mathrm{KL}(\mu \| \nu) = \sup_{\phi : \mathbb{R}^d \to \mathbb{R}} \left\{ \int \phi \, \mathrm{d}\mu - \log \int e^{\phi} \, \mathrm{d}\nu \right\}, \tag{64}$$

$$= \sup_{\phi : \mathbb{R}^d \to \mathbb{R}} \left\{ \int \phi \, \mathrm{d}\mu - \int e^{\phi(y)-1} \, \mathrm{d}\nu(y) \right\}. \tag{65}$$

To compare both formulations, we first use the inequality $x \geq \log(x) + 1$ for all $x > 0$ (by concavity of log) at $x = \int e^{\phi-1} \, \mathrm{d}\nu$. Hence,

$$\log \left( \int e^{\phi-1} \, \mathrm{d}\nu \right) + 1 = \log \left( \int e^{\phi} \, \mathrm{d}\nu \right) \leq \int e^{\phi-1} \, \mathrm{d}\nu. \tag{66}$$

As a result, for any function $\phi$, we have

$$\int \phi \, \mathrm{d}\mu - \int e^{\phi(y)-1} \, \mathrm{d}\nu(y) \le \int \phi \, \mathrm{d}\mu - \log \int e^\phi \, \mathrm{d}\nu. \tag{67}$$

This shows that the DV representation yields a tighter lower bound than the variational representation in the sense that for each choice of $\phi$, the obtained lower bound on KL in the RHS is larger.

### C.3. Descent scheme of (Baptista et al., 2025)

Baptista et al. (2025) propose to minimize

$$\mathcal{F}_\varepsilon(\mu) = \inf_{\eta \in \mathcal{P}_2(\mathbb{R}^d)} \mathrm{W}_2^2(\mu, \eta) + \varepsilon \mathrm{KL}(\eta || \nu) \tag{68}$$

using Wasserstein Gradient Descent. Below, we show that this scheme is equivalent to solving the JKO scheme using GWF with the Donsker-Varadhan formulation.

Let $\mu_\ell \in \mathcal{P}_{\mathrm{ac},2}(\mathbb{R}^d)$, $\eta_\ell^\star = \operatorname{argmin}_\eta \mathrm{W}_2^2(\mu_\ell, \eta) + \varepsilon \mathrm{KL}(\eta || \nu)$, using Proposition 3.2 for the KL, the $\ell$-th step of Wasserstein gradient descent of $\mathcal{F}_\varepsilon(\mu)$ with $\varepsilon = 2\tau$ and step size $\gamma = \frac{1}{2}$ is equivalent to the $\ell$-th step of the JKO scheme applied to the KL:

$$\mathrm{T}_\ell = \operatorname*{argmin}_{\mathrm{T}} \frac{1}{\varepsilon} \|\mathrm{T} - \mathrm{Id}\|_{L^2(\mu_\ell)}^2 + \mathrm{KL}(\mathrm{T}_{\#}\mu_\ell || \nu), \tag{69}$$

where $\mathrm{T}_\ell$ is the OT map between $\mu_\ell$ and $\mu_{\ell+1} := \eta_\ell^\star$. Denoting $h_\ell$ the Kantorovich potential such that $\mathrm{T}_\ell = \mathrm{Id} - \nabla h_\ell^c$, we get

$$\forall \ell \ge 0, \ \mu_{\ell+1} = (\mathrm{Id} - \nabla h_\ell^c)_{\#}\mu_\ell. \tag{70}$$

Baptista et al. (2025) proposed to find $h_\ell$ by solving a dual form of (68) derived through the Donsker-Varadhan formulation. They first showed that

$$\mathcal{F}_\varepsilon(\mu_\ell) = \sup_{(f,g) \in \Phi_c} \int f \, \mathrm{d}\mu_\ell - \varepsilon \log \left( \int e^{-g/\varepsilon} \, \mathrm{d}\nu \right), \tag{71}$$

where we recall that $\Phi_c = \{(f, g) \in L^1(\mu) \times L^1(\nu), \ f(x) + g(y) \le c(x, y) \text{ for } \mu \otimes \nu\text{-a.e. } (x, y)\}$. For $c(x, y) = \frac{1}{2}\|x-y\|_2^2$, we can do a similar reparametrization as (34), i.e. $\varphi = \frac{1}{2}\|\cdot\|_2^2 - f$, $\psi = \frac{1}{2}\|\cdot\|_2^2 - g$ and use $\psi = \varphi^*$, and we obtain

$$
\begin{aligned}
\mathcal{F}_\varepsilon(\mu_\ell) &= \frac{1}{2} \int \|x\|_2^2 \, \mathrm{d}\mu_\ell(x) + \sup_{\varphi \text{ convex}} - \int \varphi(x) \, \mathrm{d}\mu_\ell(x) - \varepsilon \log \left( \int e^{-\frac{\|y\|_2^2}{2\varepsilon}} e^{-\frac{\varphi^*(y)}{\varepsilon}} \, \mathrm{d}\nu(y) \right) \\
&= \frac{1}{2} \int \|x\|_2^2 \, \mathrm{d}\mu_\ell(x) - \inf_{\varphi \text{ convex}} \int \varphi(x) \, \mathrm{d}\mu_\ell(x) + \varepsilon \log \left( \int e^{-\frac{\|y\|_2^2}{2\varepsilon}} e^{-\frac{\varphi^*(y)}{\varepsilon}} \, \mathrm{d}\nu(y) \right).
\end{aligned}
\tag{72}
$$

Finally, they parametrize $\varphi^*$ by $\varphi^*(x) = \langle x, \nabla g(x) \rangle - \varphi(\nabla g(x))$ with $g$ convex, and solve

$$\mathcal{F}_\varepsilon(\mu_\ell) = \frac{1}{2} \int \|x\|_2^2 \, \mathrm{d}\mu_\ell(x) - \inf_{\varphi \text{ convex}} \sup_{g \text{ convex}} \int \varphi(x) \, \mathrm{d}\mu_\ell(x) + \varepsilon \log \left( \int e^{-\frac{\|y\|_2^2}{2\varepsilon}} e^{-\frac{\langle x, \nabla g(x) \rangle - \varphi(\nabla g(x))}{\varepsilon}} \, d\nu(y) \right), \tag{73}$$

using ICNNs (Amos et al., 2017) to parametrize $\varphi$ and $g$.

Hence, the scheme proposed in (Baptista et al., 2025) can be seen as a specific instance of GWF, i.e. generative modeling through JKO schemes. In their case, the objective is the KL, whose dual formulation is chosen as the Donsker-Varadhan formula, and where the optimized functions (in the dual form) are parametrized with ICNNs.

## D. UOT and JKO with MMD

In this section, we derive the dual of the UOT problem with IPM regularization. We then derive a new dual formulation for the squared MMD regularization and show how it can be used to obtain a JKO-based generative scheme. Finally, we present the full algorithm of the JKO MMD GAN, which corresponds to a slight adaptation of the previously derived scheme.

### D.1. Duality of UOT with IPM Regularization

Let $\mathcal{G}$ be a class of test functions. Following Manupriya et al. (2024), we assume that $\mathcal{G}$ is compact and absolutely convex (*i.e.* convex and such that $\lambda\mathcal{G} \subset \mathcal{G}$ for all $\lambda \in \mathbb{R}$ such that $|\lambda| \leq 1$). We now derive the dual of the UOT problem with IPM regularization between $\mu, \nu \in \mathcal{P}_2(\mathbb{R}^\lceil)$. Let $\lambda_1, \lambda_2 > 0$. We start from the primal IPM-based UOT objective:

$$
\begin{aligned}
\mathrm{UOT}_c(\mu,\nu) &= \min_{\gamma \in \mathcal{M}^+(\mathbb{R}^d \times \mathbb{R}^d)} \left\{ \int c(x,y)\,\mathrm{d}\gamma(x,y) + \lambda_1 \mathrm{IPM}_{\mathcal{G}}(\pi^1_\#\gamma, \mu) + \lambda_2 \mathrm{IPM}_{\mathcal{G}}(\pi^2_\#\gamma, \nu) \right\} \\
&= \min_{\gamma \in \mathcal{M}^+(\mathbb{R}^d \times \mathbb{R}^d)} \left\{ \int c(x,y)\,\mathrm{d}\gamma(x,y) + \lambda_1 \sup_{f \in \mathcal{G}} \left| \int f\,\mathrm{d}\mu - \int f\,\mathrm{d}(\pi^1_\#\gamma) \right| + \lambda_2 \sup_{g \in \mathcal{G}} \left| \int g\,\mathrm{d}\nu - \int g\,\mathrm{d}(\pi^2_\#\gamma) \right| \right\}.
\end{aligned}
\tag{74}
$$

Since $\mathcal{G}$ is compact, the supremums are attained. Moreover, as it is absolutely convex, $f \in \mathcal{G}$ implies $-f \in \mathcal{G}$, and thus we can remove the absolute values.

$$
= \min_{\gamma \in \mathcal{M}^+(\mathbb{R}^d \times \mathbb{R}^d)} \left\{ \int c(x,y)\,\mathrm{d}\gamma(x,y) + \max_{f \in \mathcal{G}} \left( \lambda_1 \int f\,\mathrm{d}\mu - \lambda_1 \int f\,\mathrm{d}(\pi^1_\#\gamma) \right) + \max_{g \in \mathcal{G}} \left( \lambda_2 \int g\,\mathrm{d}\nu - \lambda_2 \int g\,\mathrm{d}(\pi^2_\#\gamma) \right) \right\}.
\tag{75}
$$

Introducing the rescaled sets $\mathcal{G}(\lambda) := \{\lambda f,\ f \in \mathcal{G}\}$, we can rewrite the above as

$$
= \min_{\gamma \in \mathcal{M}^+(\mathbb{R}^d \times \mathbb{R}^d)} \left\{ \int c(x,y)\,\mathrm{d}\gamma(x,y) + \max_{f \in \mathcal{G}(\lambda_1)} \left( \int f\,\mathrm{d}\mu - \int f\,\mathrm{d}(\pi^1_\#\gamma) \right) + \max_{g \in \mathcal{G}(\lambda_2)} \left( \int g\,\mathrm{d}\nu - \int g\,\mathrm{d}(\pi^2_\#\gamma) \right) \right\}.
\tag{76}
$$

For convenience, define the lifted functions $\tilde{f}(x,y) := f(x)$, $\tilde{g}(x,y) := g(y)$. By applying Sion's minimax theorem for one compact set (Sion, 1958, Theorem 4.2) (since $\mathcal{G}(\lambda_1) \times \mathcal{G}(\lambda_2)$ is compact and convex, $\mathcal{M}^+(\mathbb{R}^d \times \mathbb{R}^d)$ is convex, the objective is affine in each variable, hence concave-convex and continuous in $(f,g)$), the problem becomes

$$
= \max_{f \in \mathcal{G}(\lambda_1),\ g \in \mathcal{G}(\lambda_2)} \left\{ \int f\,\mathrm{d}\mu + \int g\,\mathrm{d}\nu + \min_{\gamma \in \mathcal{M}^+(\mathbb{R}^d \times \mathbb{R}^d)} \int \left( c(x,y) - \tilde{f}(x,y) - \tilde{g}(x,y) \right) \mathrm{d}\gamma(x,y) \right\}.
\tag{77}
$$

The inner minimization reduces to

$$
\min_{\gamma} \int \left( c(x,y) - f(x) - g(y) \right) \mathrm{d}\gamma(x,y) = \begin{cases} 0, & \text{if } f(x) + g(y) \leq c(x,y) \text{ for } \gamma\text{-a.e.}(x,y), \\ -\infty, & \text{otherwise.} \end{cases}
\tag{78}
$$

Therefore, the dual formulation of UOT is

$$
\mathrm{UOT}_c(\mu,\nu) = \max_{\substack{f \in \mathcal{G}(\lambda_1),\ g \in \mathcal{G}(\lambda_2) \\ (f,g) \in \Phi_c}} \left\{ \int f\,\mathrm{d}\mu + \int g\,\mathrm{d}\nu \right\},
\tag{79}
$$

where we recall that $\Phi_c = \{(f,g) \in L^1(\mu) \times L^1(\nu),\ f(x) + g(y) \leq c(x,y) \text{ for } \mu \otimes \nu\text{-a.e. } (x,y)\}$.

The semi-relaxed UOT problem corresponds to $\lambda_1 = +\infty$. Thus, the dual (79) becomes

$$
\mathrm{sUOT}_c(\mu,\nu) = \max_{g \in \mathcal{G}(\lambda_2),\ (f,g) \in \Phi_c} \int f\,\mathrm{d}\mu + \int g\,\mathrm{d}\nu.
\tag{80}
$$

Then, using the $c$-transform $g^c$ and observing that for $\mu \otimes \nu$-almost every $(x,y)$, $g^c(x) + g(y) \leq c(x,y)$ and thus $(g^c, c) \in \Phi_c$, we obtain the semi-dual relaxation

$$
\mathrm{sUOT}_c(\mu,\nu) = \max_{g \in \mathcal{G}(\lambda_2)} \int g^c\,\mathrm{d}\mu + \int g\,\mathrm{d}\nu.
\tag{81}
$$

**Ball based IPMs.** The previous computations work in particular for IPMs where $\mathcal{G} = \{f, \|f\| \leq 1\}$ for some norm $\|\cdot\|$. This includes many popular choices. In particular, for $\|\cdot\|_{\mathcal{H}}$ the norm of a reproducing kernel Hilbert space (RKHS) $\mathcal{H}$ (Steinwart & Scovel, 2012), this yields the MMD, and for $\|f\|_L = \sup\{|f(x) - f(y)|/\|x-y\|_2,\ x \neq y\}$, this yields the Wasserstein-1 distance defined for $c(x,y) = \|x-y\|_2$. For such IPMs, we observe that $\mathcal{G}(\lambda) = \{f,\ \|f\| \leq \lambda\}$.

## D.2. Duality of UOT with Squared MMD Regularization

We now discuss the dual of the UOT problem regularized by the squared MMD. Unlike IPMs, the squared MMD is not written as a supremum of the form $\sup_{f \in \mathcal{G}} \left| \int f \, d\mu - \int f \, d\nu \right|$ and therefore the derivation used in the previous section does not apply directly. Instead, it admits a variational formulation over the whole RKHS with an additional quadratic penalty term. We recall this formulation below; see *e.g.* (Mroueh & Nguyen, 2021, Proposition 4) or (Mroueh & Rigotti, 2020, Proposition 1).

**Lemma D.1.** *Let* $\mu, \nu \in \mathcal{P}_2(\mathbb{R}^d)$, *then*

$$\mathrm{MMD}_k^2(\mu, \nu) = \sup_{f \in \mathcal{H}_k} 2 \int f \, d(\mu - \nu) - \|f\|_{\mathcal{H}}^2. \tag{82}$$

*Proof.* On one hand, for $f \in \mathcal{H}_k$, $x \in \mathbb{R}^d$, the reproducing property yields $f(x) = \langle f, k(x, \cdot) \rangle_{\mathcal{H}}$. Thus,

$$\begin{aligned} \int f(x) \, d(\mu - \nu)(x) &= \int f(x) \, d\mu(x) - \int f(x) \, d\nu(x) \\ &= \int \langle f, k(x, \cdot) \rangle_{\mathcal{H}} \, d\mu(x) - \int \langle f, k(x, \cdot) \rangle_{\mathcal{H}} \, d\nu(x) \\ &= \langle f, \int k(x, \cdot) \, d(\mu - \nu)(x) \rangle_{\mathcal{H}}. \end{aligned} \tag{83}$$

Moreover,

$$\begin{aligned} 2 \int f \, d(\mu - \nu) - \|f\|_{\mathcal{H}}^2 &= - \left\| \int k(x, \cdot) \, d(\mu - \nu)(x) \right\|_{\mathcal{H}}^2 + 2 \left\langle f, \int k(x, \cdot) \, d(\mu - \nu)(x) \right\rangle_{\mathcal{H}} - \|f\|_{\mathcal{H}}^2 \\ &\quad + \left\| \int k(x, \cdot) \, d(\mu - \nu)(x) \right\|_{\mathcal{H}}^2 \\ &= - \left\| f - \int k(x, \cdot) \, d(\mu - \nu)(x) \right\|_{\mathcal{H}}^2 + \left\| \int k(x, \cdot) \, d(\mu - \nu)(x) \right\|_{\mathcal{H}}^2. \end{aligned} \tag{84}$$

Hence, as $- \left\| f - \int k(x, \cdot) \, d(\mu - \nu)(x) \right\|_{\mathcal{H}}^2 \leq 0$, the supremum is attained for $f^* = \int k(x, \cdot) \, d(\mu - \nu)(x)$. And plugging $f^*$ into the objective of (82), we verify that we recover the squared MMD:

$$\begin{aligned} 2 \int f^* \, d(\mu - \nu) - \|f^*\|_{\mathcal{H}}^2 &= 2 \left\langle f^*, \int k(x, \cdot) \, d(\mu - \nu)(x) \right\rangle_{\mathcal{H}} - \|f^*\|_{\mathcal{H}}^2 \\ &= 2 \left\| \int k(x, \cdot) \, d(\mu - \nu)(x) \right\|_{\mathcal{H}}^2 - \left\| \int k(x, \cdot) \, d(\mu - \nu)(x) \right\|_{\mathcal{H}}^2 \\ &= \left\| \int k(x, \cdot) \, d(\mu - \nu)(x) \right\|_{\mathcal{H}}^2 \\ &= \mathrm{MMD}_k^2(\mu, \nu). \end{aligned} \tag{85}$$

$\square$

Next, we derive the dual of semi-relaxed UOT problem regularized by $\frac{1}{2} \mathrm{MMD}_k^2$.

**Proposition D.2.** *Let* $\mu, \nu \in \mathcal{P}_2(\mathbb{R}^d)$, $\lambda_2 > 0$ *and let* $k$ *be a bounded continuous positive definite kernel. The semi-relaxed UOT problem regularized by the squared MMD can be written equivalently as*

$$\begin{aligned} \mathrm{sUOT}_c(\mu, \nu) &= \inf_{\gamma \in \mathcal{M}_+(\mathbb{R}^d \times \mathbb{R}^d), \pi_\#^1 \gamma = \mu} \int c(x, y) \, d\gamma(x, y) + \tfrac{\lambda_2}{2} \mathrm{MMD}_k^2(\nu, \pi_\#^2 \gamma) \\ &= \sup_{\substack{f \in C_b(\mathbb{R}^d), g \in \mathcal{H} \\ (f, g) \in \Phi_c}} \int f \, d\mu + \int g \, d\nu - \tfrac{1}{2\lambda_2} \|g\|_{\mathcal{H}}^2 \\ &= \sup_{g \in \mathcal{H}} \int g^c \, d\mu + \int g \, d\nu - \tfrac{1}{2\lambda_2} \|g\|_{\mathcal{H}}^2. \end{aligned} \tag{86}$$

*Proof.* We first rewrite the marginal constraint through its convex dual:

$$\iota_{\{\pi^1_\# \gamma = \mu\}}(\gamma) = \sup_{f \in C_b(\mathbb{R}^d)} \left\{ \int f \, \mathrm{d}\mu - \int f \, \mathrm{d}(\pi^1_\# \gamma) \right\}, \tag{87}$$

where $\iota_{\{\pi^1_\# \gamma = \mu\}}(\gamma) = 0$ if $\pi^1_\# \gamma = \mu$, and $+\infty$ otherwise. Indeed, if $\pi^1_\# \gamma = \mu$, the quantity inside the supremum is always 0. If $\pi^1_\# \gamma \neq \mu$, then there exists $f \in C_b(\mathbb{R}^d)$ such that $\int f \, \mathrm{d}\mu - \int f \, \mathrm{d}(\pi^1_\# \gamma) \neq 0$, and scaling $f$ shows that the supremum is $+\infty$.

Using this identity, we obtain

$$
\begin{aligned}
\mathrm{sUOT}_c(\mu, \nu) &= \inf_{\gamma \in \mathcal{M}_+(\mathbb{R}^d \times \mathbb{R}^d), \pi^1_\# \gamma = \mu} \int c(x, y) \, \mathrm{d}\gamma(x, y) + \tfrac{\lambda_2}{2} \mathrm{MMD}^2_k(\nu, \pi^2_\# \gamma) \\
&= \inf_{\gamma \in \mathcal{M}_+(\mathbb{R}^d \times \mathbb{R}^d)} \int c(x, y) \, \mathrm{d}\gamma(x, y) + \iota_{\{\pi^1_\# \gamma = \mu\}}(\gamma) + \frac{\lambda_2}{2} \| m_\nu - m_{\pi^2_\# \gamma} \|^2_{\mathcal{H}}.
\end{aligned}
\tag{88}
$$

**Fenchel–Rockafellar formulation.** We now cast the problem in the form required to apply Fenchel–Rockafellar duality. Let us define $G : \mathcal{M}_+(\mathbb{R}^d \times \mathbb{R}^d) \to \mathbb{R}$ as $G(\gamma) = \int c(x, y) \, \mathrm{d}\gamma(x, y)$ for $\gamma \in \mathcal{M}_+(\mathbb{R}^d \times \mathbb{R}^d)$, $A : \mathcal{M}_+(\mathbb{R}^d \times \mathbb{R}^d) \to \mathcal{M}_+(\mathbb{R}^d) \times \mathcal{H}$ as $A\gamma = (\pi^1_\# \gamma, m_{\pi^2_\# \gamma})$ where $m_{\pi^2_\# \gamma}$ denotes the kernel mean embedding of $\pi^2_\# \gamma$, namely $m_{\pi^2_\# \gamma} = \int k(\cdot, y) \, \mathrm{d}(\pi^2_\# \gamma)(y)$, and $F : \mathcal{M}_+(\mathbb{R}^d) \times \mathcal{H} \to \mathbb{R}$ as $F(m_1, h) = \iota_{\{m_1 = \mu\}} + \frac{\lambda_2}{2} \| m_\nu - h \|^2_{\mathcal{H}}$. Then, (88) can be written as

$$\mathrm{sUOT}_c(\mu, \nu) = \inf_{\gamma \in \mathcal{M}_+(\mathbb{R}^d \times \mathbb{R}^d)} G(\gamma) + F(A\gamma). \tag{89}$$

On one hand, $A$ is a linear operator since both the pushforward and the mean embedding are linear. Moreover, since $k$ is bounded, $A$ is bounded for the total variation norm, hence continuous. Indeed, writing $\kappa := \sup_{y \in \mathbb{R}^d} \sqrt{k(y, y)} < +\infty$, then

$$\| A\gamma \| = \| \pi^1_\# \gamma \|_{\mathrm{TV}} + \| m_{\pi^2_\# \gamma} \|_{\mathcal{H}} \leq (1 + \kappa) \| \gamma \|_{\mathrm{TV}}.$$

Hence

$$\| A \|_{\mathrm{op}} := \sup_{\gamma \neq 0} \frac{\| A\gamma \|}{\| \gamma \|_{\mathrm{TV}}} \leq 1 + \kappa.$$

We can compute its adjoint $A^* : C_b(\mathbb{R}^d) \times \mathcal{H} \to C_b(\mathbb{R}^d \times \mathbb{R}^d)$ as follows. Let $f \in C_b(\mathbb{R}^d), g \in \mathcal{H}$:

$$
\begin{aligned}
\langle (f, g), A\gamma \rangle &= \langle f, \pi^1_\# \gamma \rangle + \langle g, m_{\pi^2_\# \gamma} \rangle_{\mathcal{H}} \\
&= \int f(x) \, \mathrm{d}(\pi^1_\# \gamma)(x) + \left\langle g, \int k(\cdot, y) \, \mathrm{d}(\pi^2_\# \gamma)(y) \right\rangle_{\mathcal{H}} \\
&= \int f(x) \, \mathrm{d}\gamma(x, y) + \int g(y) \, \mathrm{d}\gamma(x, y) = \int \big( f(x) + g(y) \big) \, \mathrm{d}\gamma(x, y)
\end{aligned}
\tag{90}
$$

hence $A^*(f, g) = f \oplus g$ where $f \oplus g(x, y) = f(x) + g(y)$ for all $x, y \in \mathbb{R}^d$.

$G$ is lower semi-continuous and convex because linear, and its convex conjugate is, for $h \in C_b(\mathbb{R}^d \times \mathbb{R}^d)$,

$$
\begin{aligned}
G^*(h) &= \sup_{\gamma \in \mathcal{M}_+(\mathbb{R}^d \times \mathbb{R}^d)} \int h(x, y) \, \mathrm{d}\gamma(x, y) - \int c(x, y) \, \mathrm{d}\gamma(x, y) \\
&= \sup_{\gamma \in \mathcal{M}_+(\mathbb{R}^d \times \mathbb{R}^d)} \int (h - c) \, \mathrm{d}\gamma(x, y) \\
&= \iota_{\{h \leq c\}}.
\end{aligned}
\tag{91}
$$

Moreover, $F(m_1, z) = F_1(m_1) + F_2(z)$ with $F_1(m_1) = \iota_{\{m_1 = \mu\}}$ and $F_2(z) = \frac{\lambda_2}{2} \| m_\nu - z \|^2_{\mathcal{H}}$. The convex conjugate of $F_1$ is, for and $f \in C_b(\mathbb{R}^d)$,

$$F_1^*(f) = \sup_{m_1 \in \mathcal{M}_+(\mathbb{R}^d)} \int f \, \mathrm{d}m_1 - \iota_{\{m_1 = \mu\}} = \int f \, \mathrm{d}\mu. \tag{92}$$

Moreover, for $g \in \mathcal{H}$, using the change of variable $u = z - m_\nu$, we obtain

$$
\begin{aligned}
F_2^*(g) &= \sup_{z \in \mathcal{H}} \langle g, z \rangle_\mathcal{H} - \frac{\lambda_2}{2} \| m_\nu - z \|_\mathcal{H}^2 \\
&= \langle g, m_\nu \rangle_\mathcal{H} + \sup_{u \in \mathcal{H}} \langle g, u \rangle_\mathcal{H} - \frac{\lambda_2}{2} \| u \|_\mathcal{H}^2
\end{aligned}
\tag{93}
$$

Observe that

$$
\begin{aligned}
\langle g, u \rangle_\mathcal{H} - \frac{\lambda_2}{2} \| u \|_\mathcal{H}^2 &= -\frac{\lambda_2}{2} \left( \| u \|_\mathcal{H}^2 - \frac{2}{\lambda_2} \langle g, u \rangle_\mathcal{H} \right) \\
&= -\frac{\lambda_2}{2} \left( \left\| u - \frac{g}{\lambda_2} \right\|_\mathcal{H}^2 - \frac{1}{\lambda_2^2} \| g \|_\mathcal{H}^2 \right) \\
&= -\frac{\lambda_2}{2} \left\| u - \frac{g}{\lambda_2} \right\|_\mathcal{H}^2 + \frac{1}{2\lambda_2} \| g \|_\mathcal{H}^2.
\end{aligned}
\tag{94}
$$

Hence, the sup in (93) is attained for $u = \frac{1}{\lambda_2} g$, and

$$
F_2^*(g) = \langle g, m_\nu \rangle_\mathcal{H} + \frac{1}{\lambda_2} \| g \|_\mathcal{H}^2 - \frac{1}{2\lambda_2} \| g \|_\mathcal{H}^2 = \int g \, \mathrm{d}\nu + \frac{1}{2\lambda_2} \| g \|_\mathcal{H}^2.
\tag{95}
$$

For any $\gamma \in \mathcal{P}(\mathbb{R}^d \times \mathbb{R}^d)$ such that $\pi_\#^1 \gamma = \mu$, $F$ is continuous at $A\gamma$ and $F(A\gamma) < +\infty$, and $G(\gamma) < +\infty$. Hence, applying the Fenchel–Rockafellar strong duality theorem, namely Theorem 3 of Rockafellar (1967), we get

$$
\begin{aligned}
\mathrm{sUOT}_c(\mu, \nu) &= \sup_{(f,g) \in C_b(\mathbb{R}^d) \times \mathcal{H}} -F^* \big( -(f,g) \big) - G^* \big( A^*(f,g) \big) \\
&= \sup_{(f,g) \in C_b(\mathbb{R}^d) \times \mathcal{H}} -F_1^*(-f) - F_2^*(-g) - \iota_{\{f \oplus g \le c\}} \\
&= \sup_{\substack{(f,g) \in C_b(\mathbb{R}^d) \times \mathcal{H} \\ (f,g) \in \Phi_c}} \int f \, \mathrm{d}\mu + \int g \, \mathrm{d}\nu - \frac{1}{2\lambda_2} \| g \|_\mathcal{H}^2.
\end{aligned}
\tag{96}
$$

Finally, using the c-transform and that $(f,g) \in \Phi_c$, we obtain semi-relaxed UOT problem

$$
\mathrm{sUOT}_c(\mu, \nu) = \sup_{g \in \mathcal{H}} \left\{ \int g^c \, \mathrm{d}\mu + \int g \, \mathrm{d}\nu - \frac{1}{2\lambda_2} \| g \|_\mathcal{H}^2 \right\}.
\tag{97}
$$

$\square$

### D.3. JKO with Squared MMD

Taking inspiration from the derivation of scalable JKO (Choi et al., 2024), we start from the dual of the semi-relaxed UOT with squared MMD regularization derived in Proposition D.2:

$$
\mathrm{sUOT}_c(\mu, \nu) = \sup_{g \in \mathcal{H}} \int g^c \, \mathrm{d}\mu + \int g \, \mathrm{d}\nu - \tfrac{1}{2\lambda_2} \| g \|_\mathcal{H}^2.
\tag{98}
$$

Then, we parametrize $g^c$ by a measurable map T as $g^c(x) = \inf_\mathrm{T} c(x, \mathrm{T}(x)) - g(\mathrm{T}(x))$, where in the following we will take $c(x, y) = \| x - y \|_2^2$. Plugging this into (98), we obtain the problem

$$
\sup_{g \in \mathcal{H}} \inf_\mathrm{T} \int \big( \| x - \mathrm{T}(x) \|_2^2 - g(\mathrm{T}(x)) \big) \, \mathrm{d}\mu(x) + \int g \, \mathrm{d}\nu - \tfrac{1}{2\lambda_2} \| g \|_\mathcal{H}^2.
\tag{99}
$$

Doing the change of variable $g = \lambda_2 u$, we obtain

$$
\sup_{u \in \mathcal{H}} \inf_\mathrm{T} \int \big( \| x - \mathrm{T}(x) \|_2^2 - \lambda_2 u(\mathrm{T}(x)) \big) \, \mathrm{d}\mu(x) + \int \lambda_2 u \, \mathrm{d}\nu - \tfrac{\lambda_2}{2} \| u \|_\mathcal{H}^2.
\tag{100}
$$

Then factorizing by $\lambda_2$, this is equivalent to

$$\sup_{u \in \mathcal{H}} \inf_{\mathrm{T}} \int \left( \tfrac{1}{\lambda_2} \|x - \mathrm{T}(x)\|_2^2 - u(\mathrm{T}(x)) \right) \mathrm{d}\mu(x) + \int u \, \mathrm{d}\nu - \tfrac{1}{2}\|u\|_{\mathcal{H}}^2. \tag{101}$$

In practice, we alternate between the optimization of T and $u$. T is parametrized as a neural network, while the optimal $u$ is known in closed-form. Indeed, for any T,

$$\sup_{u \in \mathcal{H}} \int \left( \tfrac{1}{\lambda_2} \|x - \mathrm{T}(x)\|_2^2 - u(\mathrm{T}(x)) \right) \mathrm{d}\mu(x) + \int u \, \mathrm{d}\nu - \tfrac{1}{2}\|u\|_{\mathcal{H}}^2 = \mathrm{cst} + \sup_{u \in \mathcal{H}} \int u \, \mathrm{d}(\nu - \mathrm{T}_{\#}\mu) - \tfrac{1}{2}\|u\|_{\mathcal{H}}^2$$
$$= \mathrm{cst} + \frac{1}{2}\mathrm{MMD}_k^2(\mathrm{T}_{\#}\mu, \nu), \tag{102}$$

where we used the variational formulation of the squared MMD provided in Lemma D.1, whose maximum is obtained for $u^* = \int k(x, \cdot) \, \mathrm{d}(\nu - \mathrm{T}_{\#}\mu)(x)$. Thus, to solve a step of the JKO we alternate between the update

$$u^* = \int k(x, \cdot) \, \mathrm{d}(\nu - \mathrm{T}_{\#}\mu)(x), \tag{103}$$

and a gradient descent step over the parameters of $\mathrm{T}_\theta$ as

$$\theta_{k+1} = \theta_k - \eta \nabla_\theta J(\theta_k) \quad \text{where} \quad J(\theta) = \int \left( \tfrac{1}{\lambda_2} \|x - \mathrm{T}_\theta(x)\|_2^2 - u^*(\mathrm{T}_\theta(x)) \right) \mathrm{d}\mu(x), \tag{104}$$

with $\lambda_2 = 2\tau$ as detailed in Algorithm 1.

### D.4. JKO MMD GANs

As shown in Section 3, a single step of the JKO scheme associated with a functional $\mathcal{F}$ can be written as a semi-dual unbalanced optimal transport problem. In particular, when $\mathcal{F} = \mathrm{MMD}_k^2$ and the transport map is parametrized by a neural network $\mathrm{T}_\theta$, the $\ell$-th JKO step amounts to solving the following problem:

$$\sup_{u \in \mathcal{H}} \inf_{\theta} \int \left( \tfrac{1}{\lambda_2} \|x - \mathrm{T}_\theta(x)\|_2^2 - u(\mathrm{T}_\theta(x)) \right) \mathrm{d}\mu_\ell(x) + \int u \, \mathrm{d}\nu - \tfrac{1}{2}\|u\|_{\mathcal{H}}^2. \tag{105}$$

As discussed in Appendix D.3, the inner maximization over $u$ admits a closed-form solution. Indeed, for any fixed transport map $\mathrm{T}_\theta$, the optimal function $u^*$ is given by the witness function

$$u^* = \int k(x, \cdot) \, \mathrm{d}(\nu - \mathrm{T}_\theta \# \mu_\ell)(x),$$

and the maximized value coincides with $\tfrac{1}{2}\mathrm{MMD}_k^2(\mathrm{T}_\theta \# \mu_\ell, \nu)$. As a result, solving a JKO step reduces to alternating between updating the transport map $\mathrm{T}_\theta$ and evaluating the corresponding witness function, as detailed in Algorithm 1.

**Learning the embedding in the kernel.** In high-dimensional settings such as image generation, using a fixed kernel $k$ often leads to poor performance. Static kernels may fail to capture the evolving geometry of the transported distribution $\mathrm{T}_\theta \# \mu_\ell$, and MMD-based flows are known to suffer from both statistical and computational limitations in high dimensions.

To address these issues, we consider a parametrized kernel of the form

$$k_\phi(x, y) = k(h_\phi(x), h_\phi(y)),$$

where $h$ is a neural network with parameters $\phi$ embedding the data into a lower-dimensional latent space. This modification leads to a joint optimization problem over the transport map $\mathrm{T}_\theta$ and the kernel parameters $\phi$, yielding the following extended formulation:

$$\max_{\phi} \sup_{u_\phi \in \mathcal{H}_\phi} \inf_{\theta} \int \left( \tfrac{1}{\lambda_2} \|x - \mathrm{T}_\theta(x)\|_2^2 - u_\phi(\mathrm{T}_\theta(x)) \right) \mathrm{d}\mu_\ell(x) + \int u_\phi \, \mathrm{d}\nu - \tfrac{1}{2}\|u_\phi\|_{\mathcal{H}_\phi}^2. \tag{106}$$

---

**Algorithm 2** Primal–Dual Training Procedure of $\mathrm{MMD}^2$ with learned embedded kernel

---

**Inputs**: $K$ JKO steps, $N$ inner steps, $k$ a p.s.d. kernel, steps sizes $\eta_h$ and $\eta_{\mathrm{T}}$
Initialize $\mathrm{T}_{\theta_0^N} = I_d$
**for** $\ell = 1, \ldots, K$ **do**
    **for** $i = 1, \ldots, N$ **do**
        Sample $(z_j)_{j=1}^n \sim \mu_0$, $(y_j)_{j=1}^m \sim \nu$
        $\phi_\ell^{i+1} = \phi_\ell^i + \eta_h \nabla_\phi \mathrm{MMD}_{k_\phi}\big(\{\mathrm{T}_{\theta_\ell^i}(z_j)\}_{j=1}^n, \{y_j\}_{j=1}^m\big)$
        Sample $(z_j)_{j=1}^n \sim \mu_0$, $(y_j)_{j=1}^m \sim \nu$
        Compute $x_j = \mathrm{T}_{\theta_\ell^i}(z_j)$ for $j = 1, \ldots, n$
        Define $g = \frac{1}{n} \sum_{j=1}^n k_{\phi_\ell^{i+1}}(x_j, \cdot) - \frac{1}{m} \sum_{j=1}^m k_{\phi_\ell^{i+1}}(y_j, \cdot)$
        $J(\theta_\ell^i) = \frac{1}{n} \sum_{j=1}^n \left( \frac{1}{2\tau} \big\| \mathrm{T}_{\theta_{\ell-1}^N}(z_j) - x_j \big\|_2^2 - g(x_j) \right)$
        Update $\theta_\ell^{i+1} = \theta_\ell^i - \eta_{\mathrm{T}} \nabla_\theta J(\theta_\ell^i)$.
    **end for**
    $\theta_{\ell+1}^1 = \theta_\ell^N$
    $\phi_{\ell+1}^1 = \phi_\ell^N$
**end for**

---

This problem naturally leads to an alternating optimization scheme. For fixed $\phi$, the transport map is updated by minimizing

$$\theta^* \in \operatorname*{argmin}_\theta \int \left( \tfrac{1}{\lambda_2} \|x - \mathrm{T}_\theta(x)\|_2^2 - u_\phi^*\big(\mathrm{T}_\theta(x)\big) \right) \, \mathrm{d}\mu_\ell(x), \tag{107}$$

where $u_\phi^*$ denotes the witness function associated with the kernel $k_\phi$. Conversely, for fixed $\theta$, using the variational formulation of the squared MMD given in Lemma D.1, the optimization with respect to $\phi$ reduces to

$$\phi^* \in \operatorname*{argmax}_\phi \tfrac{1}{2} \mathrm{MMD}_{k_\phi}^2(\mathrm{T}_\theta \# \mu_\ell, \nu), \tag{108}$$

which encourages the kernel to maximally discriminate between the transported samples and the target distribution.

**Connection with MMD GANs.** This alternating scheme admits two complementary interpretations. On the one hand, it can be seen as learning a latent representation $\phi$ in which the JKO flow is carried out in a lower-dimensional feature space, reducing the complexity of the transport dynamics. On the other hand, it can be interpreted as an adversarial procedure, where the learned kernel $k_\phi$ plays the role of an adaptive discriminator. In this view, the update of $\mathrm{T}_\theta$ coincides with a gradient step minimizing a regularized squared MMD objective. Indeed, the gradient of the witness function term

$$-\nabla_\theta \int u_\phi^*\big(\mathrm{T}_\theta(x)\big) \, \mathrm{d}\mu_\ell(x)$$

is equal to $\nabla_\theta \mathrm{MMD}_{k_\phi}^2(\mathrm{T}_\theta \# \mu_\ell, \nu)$. Consequently, the resulting update rules are equivalent to those used in MMD GANs (Li et al., 2017; Bińkowski et al., 2018), augmented with a Wasserstein proximal regularization on the generator. As discussed in Section 3.2, this framework can be interpreted as a Wasserstein proximal formulation of MMD GANs (Lin et al., 2021).

The full training procedure is detailed in Algorithm 2.

# E. Summary of the GWF Framework

We summarize in Figure 3 the unifying results of our paper, showing the methods discussed in the paper which belong to the GWF framework.

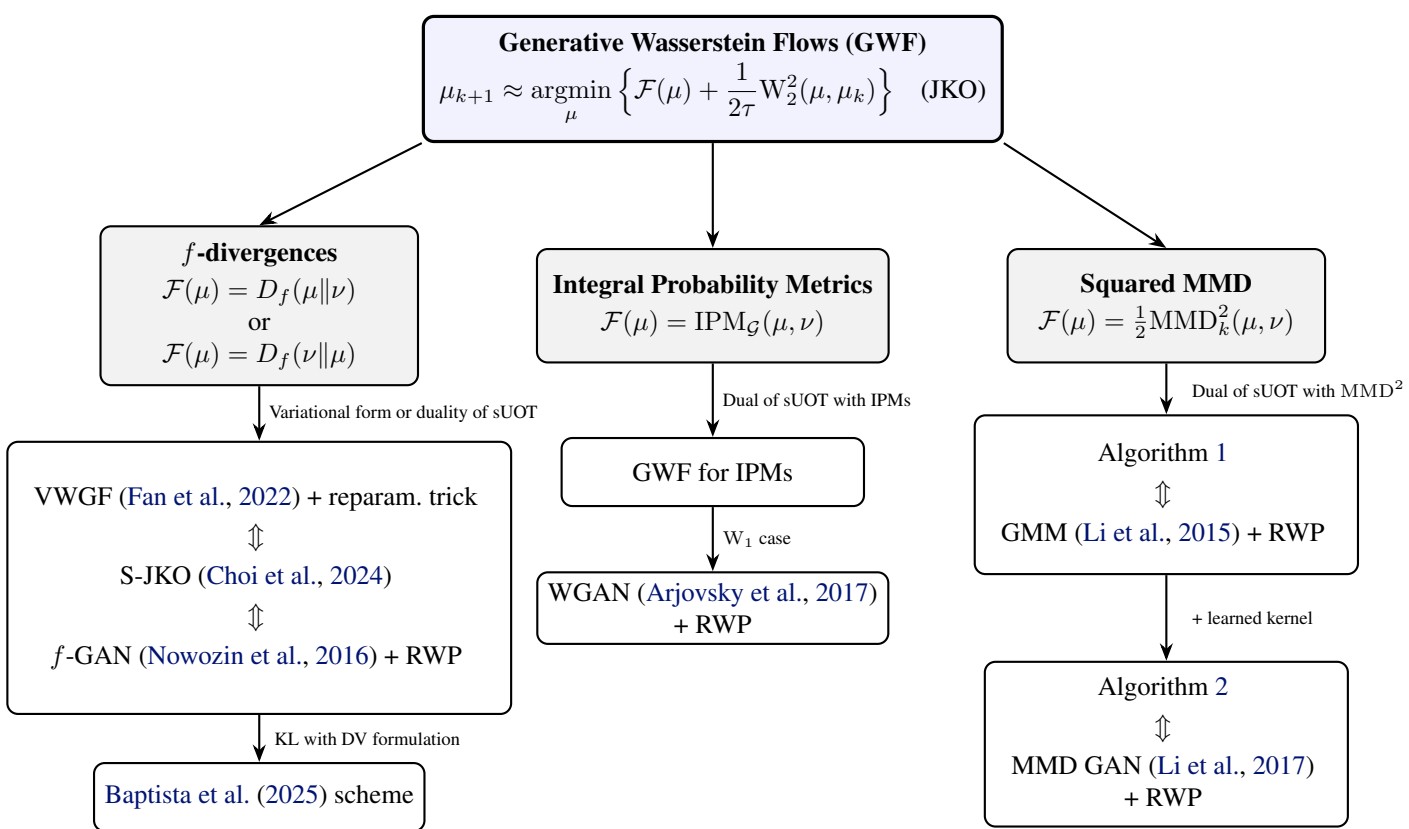

*RWP* stands for *Relaxed Wasserstein Proximal*, a regularization technique introduced in Lin et al. (2021).

> **Note on the orientation of $f$-divergences.** Classical $f$-GAN is typically written for $F(\mu) = D_f(\nu\|\mu)$, whereas VWGF and S-JKO are primarily presented in our paper for $F(\mu) = D_f(\mu\|\nu)$. When instantiated with the *same* objective functional $F$, the schemes coincide. When comparing opposite orientations, the correspondence is exact for symmetric divergences (e.g. Jensen–Shannon), while forward KL in one formulation corresponds to reverse KL in the other. This ambiguity does not arise for IPMs and $\mathrm{MMD}^2$, which are symmetric.

*Figure 3.* Schematic view of the unification results in our paper. All considered methods fit into the same Generative Wasserstein Flow (GWF) framework, obtained from parametric JKO steps with different choices of objective functional $F$.

# F. Proofs

## F.1. Proof of Proposition 3.1

Assume that $\nu, \mu \in \mathcal{P}_{\mathrm{ac},2}(\mathbb{R}^d)$. The original objective is given by

$$\mathcal{L}(\mathrm{T}, h) = \int \left( \tfrac{1}{2\tau} \|\mathrm{T} - \mathrm{Id}\|_2^2 + h \circ \mathrm{T} \right) \mathrm{d}\mu - \int f^* \circ h \, \mathrm{d}\nu, \tag{109}$$

where we use the change of variable $h \mapsto -h$ for (9).

We can reparametrize T with a coupling $\gamma \in \mathcal{P}_2(\mathbb{R}^d \times \mathbb{R}^d)$ such that $\pi^1_\# \gamma = \mu$. Hence, let us focus first on

$$\tilde{\mathcal{L}}(\gamma, h) = \int \left( \tfrac{1}{2\tau} \|y - x\|_2^2 + h(y) \right) \mathrm{d}\gamma(x, y) - \int f^* \big( h(y) \big) \mathrm{d}\nu(y). \tag{110}$$

$\tilde{\mathcal{L}}$ is convex in $\gamma$ (as linear) and concave in $h$ (as $f^*$ is convex), hence applying the minimax theorem (see *e.g.* (Liero et al., 2018, Theorem 2.4)), we get that

$$\inf_{\gamma \in \mathcal{P}_2(\mathbb{R}^d \times \mathbb{R}^d),\, \pi^1_\# \gamma = \mu} \sup_h \tilde{\mathcal{L}}(\gamma, h) = \sup_h \inf_{\gamma \in \mathcal{P}_2(\mathbb{R}^d \times \mathbb{R}^d),\, \pi^1_\# \gamma = \mu} \tilde{\mathcal{L}}(\gamma, h). \tag{111}$$

The left-hand side of (111) implies that the optimal $\gamma$ is necessary of the form $\gamma = (\mathrm{Id}, \mathrm{T})_\# \mu$ with T an optimal transport map between $\mu$ and $\pi^2_\# \gamma$ for the cost $c(x, y) = \tfrac{1}{2\tau} \|x - y\|_2^2$. Indeed, the left-hand side is of the form

$$\inf_{\gamma \in \mathcal{P}_2(\mathbb{R}^d \times \mathbb{R}^d),\, \pi^1_\# \gamma = \mu} \sup_h \tilde{\mathcal{L}}(\gamma, h) = \inf_{\gamma \in \mathcal{P}_2(\mathbb{R}^d \times \mathbb{R}^d),\, \pi^1_\# \gamma = \mu} \int \tfrac{1}{2\tau} \|x - y\|_2^2 \, \mathrm{d}\gamma(x, y) + \mathrm{D}_f(\pi^2_\# \gamma \| \nu), \tag{112}$$

and $\mu \in \mathcal{P}_{\mathrm{ac},2}(\mathbb{R}^d)$, thus by (Eyring et al., 2024, Proposition 3.1), the optimal plan $\gamma$ is given by an OT map between $\mu$ and $\pi^2_\# \gamma$. Hence, we have

$$\inf_{\mathrm{T}} \sup_h \mathcal{L}(\mathrm{T}, h) = \inf_{\gamma \in \mathcal{P}_2(\mathbb{R}^d \times \mathbb{R}^d), \pi^1_\# \gamma = \mu} \sup_h \tilde{\mathcal{L}}(\gamma, h) = \sup_h \inf_{\gamma \in \mathcal{P}_2(\mathbb{R}^d \times \mathbb{R}^d),\, \pi^1_\# \gamma = \mu} \tilde{\mathcal{L}}(\gamma, h). \tag{113}$$

For the other side, note that we always have on one hand

$$\sup_h \inf_{\mathrm{T}} \mathcal{L}(\mathrm{T}, h) \le \inf_{\mathrm{T}} \sup_h \mathcal{L}(\mathrm{T}, h) = \sup_h \inf_{\gamma \in \mathcal{P}_2(\mathbb{R}^d \times \mathbb{R}^d),\, \pi^1_\# \gamma = \mu} \tilde{\mathcal{L}}(\gamma, h). \tag{114}$$

On the other hand, $\{(\mathrm{Id}, \mathrm{T})_\# \mu,\ \mathrm{T} : \mathbb{R}^d \to \mathbb{R}^d \text{ measurable}\}$ only gives a subclass of couplings with first marginal $\mu$, hence

$$\inf_{\gamma \in \mathcal{P}_2(\mathbb{R}^d \times \mathbb{R}^d),\, \pi^1_\# \gamma = \mu} \tilde{\mathcal{L}}(\gamma, h) \le \inf_{\mathrm{T}} \mathcal{L}(\mathrm{T}, h), \tag{115}$$

and thus

$$\sup_h \inf_{\gamma \in \mathcal{P}_2(\mathbb{R}^d \times \mathbb{R}^d),\, \pi^1_\# \gamma = \mu} \tilde{\mathcal{L}}(\gamma, h) \le \sup_h \inf_{\mathrm{T}} \mathcal{L}(\mathrm{T}, h). \tag{116}$$

Therefore, we can conclude using (114) and (116) that

$$\sup_h \inf_{\gamma \in \mathcal{P}_2(\mathbb{R}^d \times \mathbb{R}^d),\, \pi^1_\# \gamma = \mu} \tilde{\mathcal{L}}(\gamma, h) = \sup_h \inf_{\mathrm{T}} \mathcal{L}(\mathrm{T}, h), \tag{117}$$

and thus

$$\sup_h \inf_{\mathrm{T}} \mathcal{L}(\mathrm{T}, h) = \inf_{\mathrm{T}} \sup_h \mathcal{L}(\mathrm{T}, h). \tag{118}$$

## F.2. Proof of Proposition 3.2

Let $\nu \in \mathcal{P}_{\mathrm{ac},2}(\mathbb{R}^d)$, recall that Baptista et al. (2025) introduced the proximal OT divergence as, for all $\mu \in \mathcal{P}_2(\mathbb{R}^d)$,

$$\mathcal{F}_\varepsilon(\mu) = \inf_{\eta \in \mathcal{P}_2(\mathbb{R}^d)} \mathrm{W}_2^2(\mu, \eta) + \varepsilon \mathrm{D}_f(\eta \| \nu). \tag{119}$$

We claim in Proposition 3.2 that the forward discretization of (119) coincides with the JKO scheme of $\mathcal{F}(\mu) = \mathrm{D}_f(\mu \| \nu)$ for $\varepsilon = 2\tau$. We detail this equivalence below, by first describing the JKO update of $\mathcal{F}$, and then the forward update of $\mathcal{F}_\varepsilon$.

**Minimization of $D_f$ via JKO scheme.** We recall that the JKO scheme to minimize $\mathcal{F}(\mu) = D_f(\mu||\nu)$ is, for all $\ell \geq 0$,

$$\mu_{\ell+1} = \underset{\mu \in \mathcal{P}_2(\mathbb{R}^d)}{\text{argmin}} \ \frac{1}{2\tau} W_2^2(\mu_\ell, \mu) + D_f(\mu||\nu), \tag{120}$$

As $\nu \in \mathcal{P}_{\text{ac},2}(\mathbb{R}^d)$, $D_f(\mu||\nu) < \infty$ if and only if $\mu \in \mathcal{P}_{\text{ac},2}(\mathbb{R}^d)$. Thus, leveraging Brenier's theorem, the JKO scheme is equivalent to

$$\begin{cases} T_{\ell+1} = \text{argmin}_{T \in L^2(\mu_\ell)} \ \frac{1}{2\tau} \int \|x - T(x)\|_2^2 \, d\mu_\ell(x) + D_f(T_{\#}\mu_\ell||\nu) \\ \mu_{\ell+1} = (T_{\ell+1})_{\#}\mu_\ell. \end{cases} \tag{121}$$

The following lemma states that $T_{\ell+1}$ is necessarily the OT map between $\mu_\ell$ and $\mu_{\ell+1}$.

**Lemma F.1.** $T_{\ell+1}$ in (121) is the optimal transport map between $\mu_\ell$ and

$$\mu_{\ell+1} = \underset{\eta}{\text{argmin}} \ W_2^2(\mu_\ell, \eta) + 2\tau D_f(\eta||\nu). \tag{122}$$

*Proof.* Suppose $T_{\ell+1}$ is not the OT map, then there exists $\tilde{T}$ such that $\|\text{Id} - \tilde{T}\|_{L^2(\mu_\ell)}^2 < \|\text{Id} - T_{\ell+1}\|_{L^2(\mu_\ell)}^2$ and $\tilde{T}_{\#}\mu_\ell = \mu_{\ell+1}$. Then we would have:

$$\|\text{Id} - \tilde{T}\|_{L^2(\mu_\ell)}^2 + 2\tau D_f(\tilde{T}_{\#}\mu_\ell||\nu) < \|\text{Id} - T_{\ell+1}\|_{L^2(\mu_\ell)}^2 + 2\tau D_f(T_{\ell+1\#}\mu_\ell||\nu), \tag{123}$$

contradicting the optimality of $T_{\ell+1}$, since $T_{\ell+1\#}\mu_\ell = \mu_{\ell+1}$. $\square$

We can now apply the second part of Theorem 1.17 of (Santambrogio, 2015) with the cost $c_\tau(x, y) = \frac{1}{2\tau}\|x - y\|_2^2$. Since:

$$h(z) = \frac{1}{2\tau}\|z\|^2 \Rightarrow \nabla h(z) = \frac{1}{\tau}z \Rightarrow (\nabla h)^{-1}(u) = \tau u, \tag{124}$$

we obtain:

$$T_{\ell+1}(x) = x - (\nabla h)^{-1}\big(\nabla \phi_{\mu_\ell, \mu_{\ell+1}}^{c_\tau}(x)\big) = x - \tau \nabla \phi_{\mu_\ell, \mu_{\ell+1}}^{c_\tau}(x), \tag{125}$$

where $\phi_{\mu_\ell, \mu_{\ell+1}}^{c_\tau}$ is the Kantorovich potential between $\mu_\ell$ and $\mu_{\ell+1}$ for $c_\tau$. Therefore, the JKO update becomes:

$$\mu_{\ell+1} = T_{\ell+1\#}\mu_\ell = (\text{Id} - \tau \nabla \phi_{\mu_\ell, \mu_{\ell+1}}^{c_\tau})_{\#}\mu_\ell. \tag{126}$$

We now show that the Wasserstein Gradient Descent scheme of (119) coincides with this update.

**Wasserstein Gradient Descent of ProxOT** (119). A Wasserstein gradient descent step for (119) is, for all $\ell \geq 0$, $\gamma > 0$,

$$\mu_{\ell+1} = \big(\text{Id} - \gamma \nabla_{W_2}\mathcal{F}_\varepsilon(\mu_\ell)\big)_{\#}\mu_\ell. \tag{127}$$

Let us first compute the Wasserstein gradient of (119).

**Lemma F.2.** Let $\mu, \nu \in \mathcal{P}_{\text{ac},2}(\mathbb{R}^d)$. Let us define

$$\eta_{\mu,\nu}^\star = \underset{\eta \in \mathcal{P}_2(\mathbb{R}^d)}{\text{argmin}} \ W_2^2(\eta, \mu) + \varepsilon D_f(\eta||\nu). \tag{128}$$

Then, the Wasserstein gradient of $\mathcal{F}_\varepsilon$ is given by $\nabla_{W_2}\mathcal{F}_\varepsilon(\mu) = \varepsilon \nabla \phi_{\mu,\eta^\star}^{c_\varepsilon}$, with $\phi_{\mu,\eta^\star}$ the Kantorovich potential between $\mu$ and $\eta_{\mu,\nu}^\star$ for $c_\varepsilon(x, y) = \frac{1}{\varepsilon}\|x - y\|_2^2$.

*Proof.* (128) is equivalent to (Liero et al., 2018)

$$\begin{cases} \gamma^\star \in \text{argmin}_{\gamma \in \mathcal{P}_2(\mathbb{R}^d \times \mathbb{R}^d), \ \pi_{\#}^1 \gamma = \mu} \ \int \frac{1}{\varepsilon}\|x - y\|_2^2 \, d\gamma(x, y) + D_f(\pi_{\#}^2 \gamma||\nu) \\ \eta_{\mu,\nu}^\star = \pi_{\#}^2 \gamma^\star, \end{cases} \tag{129}$$

with $\pi^2 : (x, y) \mapsto y$. Moreover, $\gamma^\star$ is necessarily an optimal coupling between $\mu$ and $\pi_\#^2 \gamma^\star$, as otherwise, we could find another coupling with the same second marginal, but with lower cost (same argument as Lemma F.1). Let $c_\varepsilon(x, y) = \frac{1}{\varepsilon}\|x - y\|_2^2$, and denote by $\phi_{\mu,\eta}$ the optimal Kantorovich potential between $\mu$ and $\eta_{\mu,\nu}^\star$ for the cost $c_\varepsilon$, which satisfies for $\gamma^\star$-almost every $(x, y)$, $\phi_{\mu,\eta}(y) + \phi_{\mu,\eta}^{c_\varepsilon}(x) = c_\varepsilon(x, y)$. Hence, we have the equality

$$
\begin{aligned}
\mathcal{F}_\varepsilon(\mu) &= \varepsilon \cdot \left( \int \frac{1}{\varepsilon}\|x - y\|_2^2 \, \mathrm{d}\gamma^\star(x, y) + \mathrm{D}_f(\pi_\#^2 \gamma^\star \| \nu) \right) \\
&= \varepsilon \cdot \left( \int \phi_{\mu,\eta}^{c_\varepsilon}(x) \, \mathrm{d}\mu(x) - \int f^*\big( -\phi_{\mu,\eta}(y) \big) \, \mathrm{d}\nu(y) \right),
\end{aligned}
\tag{130}
$$

using the dual formulation and optimality conditions (see *e.g.* (Eyring et al., 2024, Proposition 3.1).

Let $\xi : \mathbb{R}^d \to \mathbb{R}^d$ be a diffeomorphism, $s > 0$. Using the dual formulation, we get

$$
\begin{aligned}
\mathcal{F}_\varepsilon\big((\mathrm{Id} + s\xi)_\# \mu\big) &\geq \varepsilon \left( \int \phi_{\mu,\eta}^{c_\varepsilon}(x) \, \mathrm{d}\big((\mathrm{Id} + s\xi)_\# \mu\big)(x) - \int f^*\big( -\phi_{\mu,\eta}(y) \big) \, \mathrm{d}\nu(y) \right) \\
&= \varepsilon \left( \int \phi_{\mu,\eta}^{c_\varepsilon}\big(x + s\xi(x)\big) \, \mathrm{d}\mu(x) - \int f^*\big( -\phi_{\mu,\eta}(y) \big) \, \mathrm{d}\nu(y) \right).
\end{aligned}
\tag{131}
$$

Hence,

$$
\frac{\mathcal{F}_\varepsilon\big((\mathrm{Id} + s\xi)_\# \mu\big) - \mathcal{F}_\varepsilon(\mu)}{\varepsilon s} \geq \int \frac{\phi_{\mu,\eta}^{c_\varepsilon}\big(x + s\xi(x)\big) - \phi_{\mu,\eta}^{c_\varepsilon}(x)}{s} \, \mathrm{d}\mu(x).
\tag{132}
$$

Taking the limit $s \to 0$ by Lebesgue's dominated convergence theorem,

$$
\liminf_{s \to 0} \frac{\mathcal{F}_\varepsilon\big((\mathrm{Id} + s\xi)_\# \mu\big) - \mathcal{F}_\varepsilon(\mu)}{\varepsilon s} \geq \int \langle \nabla \phi_{\mu,\eta}^{c_\varepsilon}(x), \xi(x) \rangle \, \mathrm{d}\mu(x).
\tag{133}
$$

On the other hand, observe that $\mathrm{T} = \mathrm{Id} - \frac{\varepsilon}{2} \nabla \phi_{\mu,\eta}^{c_\varepsilon}$ is an OT map between $\mu$ and $\eta_{\mu,\nu}^\star$ (by (Santambrogio, 2015, Theorem 1.17). Hence $\gamma = (\mathrm{Id} + s\xi, \mathrm{Id} - \frac{\varepsilon}{2} \nabla \phi_{\mu,\eta}^{c_\varepsilon})_\# \mu \in \Pi\big((\mathrm{Id} + s\xi)_\# \mu, \eta_{\mu,\nu}^\star\big)$. Thus,

$$
\begin{aligned}
\mathcal{F}_\varepsilon\big((\mathrm{Id} + s\xi)_\# \mu\big) &\leq \int \|x - y\|_2^2 \, \mathrm{d}\gamma(x, y) + \varepsilon \mathrm{D}_f(\pi_\#^2 \gamma \| \nu) \\
&= \int \|x + s\xi(x) - x + \tfrac{\varepsilon}{2} \nabla \phi_{\mu,\nu}^{c_\varepsilon}(x)\|_2^2 \, \mathrm{d}\mu(x) + \varepsilon \mathrm{D}_f(\eta_{\mu,\nu}^\star \| \nu) \\
&= \int \|x - (x - \tfrac{\varepsilon}{2} \nabla \phi_{\mu,\nu}^{c_\varepsilon}(x))\|_2^2 \, \mathrm{d}\mu(x) \\
&\quad + \varepsilon \mathrm{D}_f(\eta_{\mu,\nu}^\star \| \nu) + s\varepsilon \int \langle \xi(x), \nabla \phi_{\mu,\nu}^{c_\varepsilon}(x) \rangle \, \mathrm{d}\mu(x) + s^2 \int \|\xi(x)\|_2^2 \, \mathrm{d}\mu(x) \\
&= \mathrm{W}_2^2(\mu, \eta_{\mu,\nu}^\star) + \varepsilon \mathrm{D}_f(\eta_{\mu,\nu}^\star \| \nu) + s\varepsilon \int \langle \xi(x), \nabla \phi_{\mu,\nu}^{c_\varepsilon}(x) \rangle \, \mathrm{d}\mu(x) + s^2 \int \|\xi(x)\|_2^2 \, \mathrm{d}\mu(x) \\
&= \mathcal{F}_\varepsilon(\mu) + s\varepsilon \int \langle \xi(x), \nabla \phi_{\mu,\nu}^{c_\varepsilon}(x) \rangle \, \mathrm{d}\mu(x) + s^2 \int \|\xi(x)\|_2^2 \, \mathrm{d}\mu(x).
\end{aligned}
\tag{134}
$$

Hence,

$$
\limsup_{s \to 0} \frac{\mathcal{F}_\varepsilon\big((\mathrm{Id} + s\xi)_\# \mu\big) - \mathcal{F}_\varepsilon(\mu)}{\varepsilon s} \leq \int \langle \nabla \phi_{\mu,\nu}^{c_\varepsilon}(x), \xi(x) \rangle \, \mathrm{d}\mu(x).
\tag{135}
$$

Thus, we can conclude that $\nabla_{\mathrm{W}_2} \mathcal{F}_\varepsilon(\mu) = \varepsilon \nabla \phi_{\mu,\eta}^{c_\varepsilon}$. $\qquad\square$

Hence, applying Lemma F.2 with $\mu = \mu_\ell$ and $\eta_{\mu_\ell,\nu}^\star = \mathrm{argmin}_\eta \ \mathrm{W}_2^2(\eta, \mu_\ell) + \varepsilon \mathrm{D}_f(\eta \| \nu)$, we get

$$
\mu_{\ell+1} = (\mathrm{Id} - \gamma \varepsilon \nabla \phi_{\mu_\ell, \eta_{\mu_\ell,\nu}^\star}^{c_\varepsilon})_\# \mu_\ell.
\tag{136}
$$

Taking $\varepsilon = 2\tau$ and $\gamma = \frac{1}{2}$, and observing that in that case, $\phi^{c_\varepsilon} = \phi^{c_\tau}$, we recover the update rule of the JKO scheme (126).

### F.3. Proof of Proposition 3.3

The standard VWGF derivation in Section 3 is written for the functional $\tilde{\mathcal{F}}(\mu) = D_f(\mu\|\nu)$, which leads, after the reparametrization trick, to the update rule given in Equation (12).

We now apply the same derivation to the functional $\mathcal{F}(\mu) = D_f(\nu\|\mu)$. Using the variational representation of the $f$-divergence in this orientation gives

$$D_f(\nu\|\mu) = \sup_h \left\{ \mathbb{E}_\nu[h(y)] - \mathbb{E}_\mu[f^*(h(x))] \right\}. \tag{137}$$

The corresponding VWGF step reads

$$\inf_{\tilde{T}} \frac{1}{2\tau} \mathbb{E}_{\mu_\ell} \left[ \|\tilde{T} - \mathrm{Id}\|_2^2 \right] + \sup_h \left\{ \mathbb{E}_\nu[h(y)] - \mathbb{E}_{\mu_\ell} \left[ f^*(h(\tilde{T}(x))) \right] \right\}. \tag{138}$$

Using the reparameterization trick described in Section 3.2 yields

$$\inf_T \sup_h \left\{ \mathbb{E}_\nu[h(y)] - \mathbb{E}_{\mu_0}[f^*(h(T(x)))] + \frac{1}{2\tau} \mathbb{E}_{\mu_0} \left[ \|T - T_\ell\|_2^2 \right] \right\}, \tag{139}$$

which coincides with the RWP-$f$-GAN objective introduced by Lin et al. (2021).

Therefore,

$$\text{VWGF applied to } D_f(\nu\|\mu) \equiv \text{RWP-}f\text{-GAN applied to } D_f(\nu\|\mu).$$

We now relate this formulation to the orientation used in the paper. Defining the adjoint generator $\tilde{f}(r) := r f(1/r)$, we have the standard identity

$$D_f(\nu\|\mu) = D_{\tilde{f}}(\mu\|\nu). \tag{140}$$

Consequently,

$$\text{RWP-}f\text{-GAN on } D_f(\nu\|\mu) \equiv \text{VWGF on } D_{\tilde{f}}(\mu\|\nu).$$

This shows that the two schemes correspond to the same proximal update once applied to the same orientation of the divergence, and therefore differ only by the choice of orientation in their original formulations.

### F.4. Proof of Proposition 6.1

We first compute the gradient of $\theta \mapsto \mathcal{F}(\mu_\theta)$ where $\mu_\theta = F_{\theta\#}\mu$.

**Proposition F.3.** *Let $\mathcal{F} : \mathcal{P}_2(\mathbb{R}^d) \to \mathbb{R}$ be a functional that is Wasserstein differentiable. Let $F_\theta : \mathbb{R}^d \to \mathbb{R}^d$ be a smooth map parametrized by $\theta \in \mathbb{R}^p$, and define $\mu_\theta = F_{\theta\#}\mu$ for some fixed reference measure $\mu \in \mathcal{P}_2(\mathbb{R}^d)$. Then the gradient of $\theta \mapsto \mathcal{F}(F_{\theta\#}\mu)$ is*

$$\nabla_\theta \mathcal{F}(F_{\theta\#}\mu) = \int \partial_\theta F_\theta(x)^\top \nabla_{W_2}\mathcal{F}(F_{\theta\#}\mu)\big(F_\theta(x)\big) \, d\mu(x). \tag{141}$$

*Proof.* Let $h \in \mathbb{R}^p$. As Wasserstein gradients are "strong gradients" (Lanzetti et al., 2025, Proposition 2.12), using the coupling $\gamma = (F_\theta, F_{\theta+h})_{\#}\mu \in \Pi(F_{\theta\#}\mu, F_{\theta+h\#}\mu)$, we obtain

$$\mathcal{F}(F_{\theta+h\#}\mu) - \mathcal{F}(F_{\theta\#}\mu) = \int \langle \nabla_{W_2}\mathcal{F}(F_{\theta\#}\mu)(x), y - x \rangle \, d\big((F_\theta, F_{\theta+h})_{\#}\mu\big)(x, y)$$

$$+ o\left( \sqrt{\int \|x - y\|^2 \, d\big((F_\theta, F_{\theta+h})_{\#}\mu\big)(x, y)} \right) \tag{142}$$

$$= \int \langle \nabla_{W_2}\mathcal{F}(F_{\theta\#}\mu)\big(F_\theta(x)\big), F_{\theta+h}(x) - F_\theta(x) \rangle \, d\mu(x) + o(\|h\|), \tag{143}$$

where we justify the $o(\|h\|)$ after.

Now apply the first-order Taylor expansion for vector-valued functions (Nasirzadeh et al., 2011):

$$F_{\theta+h}(x) = F_\theta(x) + \partial_\theta F_\theta(x)\, h + o(\|h\|). \tag{144}$$

Substituting (144) into (143), we get

$$\mathcal{F}(F_{\theta+h\#}\mu) - \mathcal{F}(F_{\theta\#}\mu) = \int \langle \nabla_{W_2}\mathcal{F}(F_{\theta\#}\mu)\big(F_\theta(x)\big), \partial_\theta F_\theta(x)h \rangle \,\mathrm{d}\mu(x) + o(\|h\|). \tag{145}$$

Extracting the linear term in $h$ yields

$$\mathcal{F}(F_{\theta+h\#}\mu) - \mathcal{F}(F_{\theta\#}\mu) = \left\langle \int \partial_\theta F_\theta(x)^\top \nabla_{W_2}\mathcal{F}(F_{\theta\#}\mu)\big(F_\theta(x)\big) \,\mathrm{d}\mu(x), h \right\rangle + o(\|h\|). \tag{146}$$

Thus, by definition of the gradient,

$$\nabla_\theta \mathcal{F}(F_{\theta\#}\mu) = \int \partial_\theta F_\theta(x)^\top \nabla_{W_2}\mathcal{F}(F_{\theta\#}\mu)\big(F_\theta(x)\big) \,\mathrm{d}\mu(x). \tag{147}$$

**Justification of the $o(\|h\|)$ control.** We verify that the remainder term in (142) is indeed $o(\|h\|)$. From (144), we compute

$$I(h) := \int \|F_\theta(x) - F_{\theta+h}(x)\|^2 \,\mathrm{d}\mu(x) = \int \|\partial_\theta F_\theta(x)h + o(\|h\|)\|^2 \,\mathrm{d}\mu(x) \tag{148}$$

$$\leq 2\left( \int \|\partial_\theta F_\theta(x)h\|^2 \,\mathrm{d}\mu(x) + o(\|h\|^2) \right) \tag{149}$$

$$= 2h^\top \left( \int \partial_\theta F_\theta(x)^\top \partial_\theta F_\theta(x) \,\mathrm{d}\mu(x) \right)h + o(\|h\|^2). \tag{150}$$

Let $G(\theta) := \int \partial_\theta F_\theta(x)^\top \partial_\theta F_\theta(x) \,\mathrm{d}\mu(x)$ then $G(\theta) \succeq 0$. Then $I(h) \leq (\lambda_{\max}(G(\theta)) + o(1))\|h\|^2$. Thus $\sqrt{I(h)} \leq C\|h\|$ for some $C > 0$ and small enough $\|h\|$. If the remainder satisfies $r(h) = o(\sqrt{I(h)})$, then

$$\frac{r(h)}{\|h\|} = \frac{r(h)}{\sqrt{I(h)}} \cdot \frac{\sqrt{I(h)}}{\|h\|} \xrightarrow[h\to 0]{} 0 \cdot C = 0, \tag{151}$$

so indeed $r(h) = o(\|h\|)$. This justifies the step in (143) and completes the proof. $\square$

We are now ready to show Proposition 6.1.

*Proof of Proposition 6.1.* From Proposition F.3, the parameter dynamics are

$$\dot\theta_t = -\nabla_\theta \mathcal{F}(\mu_{\theta_t}) = -\int \partial_\theta F_{\theta_t}(y)^\top \nabla_{W_2}\mathcal{F}(\mu_{\theta_t})\big(F_{\theta_t}(y)\big) \,\mathrm{d}\mu(y). \tag{152}$$

For $x_t \sim \mu_{\theta_t}$, there exists $z \sim \mu$ such that $x_t = F_{\theta_t}(z)$. Since $\dot x_t = v_{\theta_t}(x_t)$, using the chain rule, we identify $v_{\theta_t}$ with

$$v_{\theta_t}(x_t) = \partial_\theta F_{\theta_t}(z) \cdot \dot\theta_t = -\partial_\theta F_{\theta_t}(z) \cdot \int \partial_\theta F_{\theta_t}(y)^\top \nabla_{W_2}\mathcal{F}(\mu_{\theta_t})\big(F_{\theta_t}(y)\big) \,\mathrm{d}\mu(y). \tag{153}$$

Consequently, $(\mu_{\theta_t})_t$ solves (weakly) the continuity equation

$$\partial_t \mu_{\theta_t} = \nabla \cdot \Big( \mu_{\theta_t}\, \mathcal{H}_{\theta_t}[\nabla_{W_2}\mathcal{F}(\mu_{\theta_t})] \Big), \tag{154}$$

where the operator $\mathcal{H}_{\theta_t}$ is defined for any $f \in L^2(\mu_\theta)$ and $x \in \mathbb{R}^d$ as

$$\big[\mathcal{H}_\theta f\big](x) := \partial_\theta F_\theta\big(F_\theta^{-1}(x)\big)\, \mathcal{G}_\theta(f). \tag{155}$$

$\square$

## F.5. Proof of Proposition 6.2

We consider the flow induced by

$$\dot{\theta}_t = -G(\theta_t)^{-1}\nabla_\theta \mathcal{F}(\mu_{\theta_t}), \tag{156}$$

where $G(\theta) = \int \partial_\theta F_\theta(z)^\top \partial F_\theta(z)\, \mathrm{d}\mu(z)$.

First, we show that $v_\theta = -\tilde{\mathcal{H}}_{\theta_t}\nabla_{\mathrm{W}_2}\mathcal{F}(\mu_{\theta_t})$. For $x_t \sim \mu_{\theta_t}$, there exists $z \sim \mu$ such that $x_t = F_{\theta_t}(z)$. Since $\dot{x}_t = v_{\theta_t}(x_t)$, using the chain rule, we identify $v_{\theta_t}$ with

$$v_{\theta_t}(x_t) = \partial_\theta F_{\theta_t}(z)\dot{\theta}_t = -\partial_\theta F_{\theta_t}(z)G(\theta)^{-1}\nabla_\theta \mathcal{F}(\mu_{\theta_t}). \tag{157}$$

Hence, by Proposition F.3,

$$v_{\theta_t}(x_t) = -\partial_\theta F_{\theta_t}(z)G(\theta_t)^{-1}\int \partial_\theta F_{\theta_t}(y)^\top \nabla_{\mathrm{W}_2}\mathcal{F}(F_{\theta_t\#}\mu)\big(F_{\theta_t}(y)\big)\, \mathrm{d}\mu(y) = -\big[\tilde{\mathcal{H}}_{\theta_t}\nabla_{\mathrm{W}_2}\mathcal{F}(\mu_{\theta_t})\big]\big(F_{\theta_t}(z)\big). \tag{158}$$

In particular $v_\theta$ belongs to $V_\theta := \{F_\theta(x) \mapsto \partial_\theta F_\theta(x)\,u \; : \; u \in \mathbb{R}^p\}$. Let us now show that $\tilde{\mathcal{H}}_\theta$ defines a projection operator. For any $f \in L^2(\mu_\theta)$, $x \in \mathbb{R}^d$,

$$
\begin{aligned}
\tilde{\mathcal{H}}_\theta\big(\tilde{\mathcal{H}}_\theta f\big)\big(F_\theta(x)\big) &= \partial_\theta F_\theta(x)G(\theta)^{-1}\int \partial_\theta F_\theta(z)^\top \tilde{\mathcal{H}}_\theta f\big(F_\theta(z)\big)\, \mathrm{d}\mu(z)\\
&= \partial_\theta F_\theta(x)\,G(\theta)^{-1}\int \partial_\theta F_\theta(z)^\top \Big(\partial_\theta F_\theta(z)\,G(\theta)^{-1}\int \partial_\theta F_\theta(y)^\top f\big(F_\theta(y)\big)\, \mathrm{d}\mu(y)\Big)\, \mathrm{d}\mu(z)\\
&= \partial_\theta F_\theta(x)\,G(\theta)^{-1}\Big(\int \partial_\theta F_\theta(z)^\top \partial_\theta F_\theta(z)\, \mathrm{d}\mu(z)\Big) G(\theta)^{-1}\int \partial_\theta F_\theta(y)^\top f\big(F_\theta(y)\big)\, \mathrm{d}\mu(y)\\
&= \partial_\theta F_\theta(x)\,G(\theta)^{-1}G(\theta)G(\theta)^{-1}\int \partial_\theta F_\theta(y)^\top f\big(F_\theta(y)\big)\, \mathrm{d}\mu(y)\\
&= \partial_\theta F_\theta(x)\,G(\theta)^{-1}\int \partial_\theta F_\theta(y)^\top f\big(F_\theta(y)\big)\, \mathrm{d}\mu(y) = \tilde{\mathcal{H}}_\theta f\big(F_\theta(x)\big).
\end{aligned} \tag{159}
$$

Hence, $\tilde{\mathcal{H}}_\theta$ is a projection. Now let us show that it is a self adjoint operator. Let $f, g \in L^2(\mu_\theta)$, then

$$
\begin{aligned}
\langle \tilde{\mathcal{H}}_\theta f, g\rangle_{L^2(\mu_\theta)} &= \int \langle \tilde{\mathcal{H}}_\theta f\big(F_\theta(y)\big), g\big(F_\theta(y)\big)\rangle\, \mathrm{d}\mu(y)\\
&= \int \langle \partial_\theta F_\theta(y)G(\theta)^{-1}\int \partial_\theta F_\theta(z)^\top f\big(F_\theta(z)\big)\, \mathrm{d}\mu(z), g\big(F_\theta(y)\big)\rangle\, \mathrm{d}\mu(y)\\
&= \int \Big\langle \int \partial_\theta F_\theta(z)^\top f\big(F_\theta(z)\big)\, \mathrm{d}\mu(z), G(\theta)^{-1}\partial_\theta F_\theta(y)^\top g\big(F_\theta(y)\big)\Big\rangle\, \mathrm{d}\mu(y)\\
&= \Big\langle \int \partial_\theta F_\theta(z)^\top f\big(F_\theta(z)\big)\, \mathrm{d}\mu(z), \int G(\theta)^{-1}\partial_\theta F_\theta(y)^\top g\big(F_\theta(y)\big)\, \mathrm{d}\mu(y)\Big\rangle\\
&= \int \langle f\big(F_\theta(z)\big), \partial_\theta F_\theta(z)\int G(\theta)^{-1}\partial_\theta F_\theta(y)^\top g\big(F_\theta(y)\big)\, \mathrm{d}\mu(y)\rangle\, \mathrm{d}\mu(z)\\
&= \int \langle f\big(F_\theta(z)\big), \tilde{\mathcal{H}}_\theta g\big(F_\theta(z)\big)\rangle\, \mathrm{d}\mu(z)\\
&= \langle f, \tilde{\mathcal{H}}_\theta g\rangle_{L^2(\mu_\theta)}.
\end{aligned} \tag{160}
$$

Hence, by (Rudin, 1991, Theorem 12.14), $\tilde{\mathcal{H}}_\theta$ is an orthogonal projection operator onto $V_\theta := \{F_\theta(x) \mapsto \partial_\theta F_\theta(x)\,u \; : \; u \in \mathbb{R}^p\}$.

# G. Parametric Flows

In this section, we clarify the theory presented in Section 6 and derive the geometric structure induced by the parametrization of probability distributions we consider. We define the associated metric tensor and Riemannian gradient in parameter space

and relate this construction to the framework of (Li & Zhao, 2019; Chen & Li, 2020). We use this geometric perspective to clarify the practical implementation of JKO-type schemes in parametrized models and to refine Proposition 6.3 by making its assumptions explicit. Finally, as an illustrative example, we apply this construction to Gaussian distributions and show that it recovers the classical Bures-Wasserstein gradient.

### G.1. Geometry induced by the parametrization

In this subsection, we define a metric induced by the parametrization and connect it to the operators $\mathcal{H}$ and $\tilde{\mathcal{H}}$ introduced in Section 6.

Let $\mu \in \mathcal{P}_2(\mathbb{R}^d)$ be a reference distribution, and let

$$F_\theta : \mathbb{R}^d \to \mathbb{R}^d, \qquad \theta \in \Theta \subset \mathbb{R}^p. \tag{161}$$

We consider the parametrized family of probability measures

$$\mu_\theta := F_{\theta\#}\mu, \tag{162}$$

and define the associated space

$$\mathcal{M}_\mu := \left\{ \mu_\theta = F_{\theta\#}\mu : \theta \in \Theta \right\} \subset \mathcal{P}_2(\mathbb{R}^d). \tag{163}$$

We endow this family with a finite-dimensional geometric structure induced by the parametrization and view $\mathcal{M}_\mu$ as a parametrized manifold embedded in the Wasserstein space $(\mathcal{P}_2(\mathbb{R}^d), W_2)$.

**Admissible velocity fields.** Let $(\theta_t)_t$ be a smooth curve in $\Theta$ such that $\theta_{t=0} = \theta$, and denote

$$\mu_{\theta_t} := F_{\theta_t \#}\mu. \tag{164}$$

Assuming that $F_{\theta_t}$ is invertible, and proceeding as in Section 6, differentiating with respect to time yields the continuity equation

$$\partial_t \mu_{\theta_t} + \nabla \cdot \left( \mu_{\theta_t} v_t \right) = 0, \tag{165}$$

where the associated velocity field is given by

$$v_t(x) = \partial_\theta F_{\theta_t}\left( F_{\theta_t}^{-1}(x) \right) \dot{\theta}_t. \tag{166}$$

Evaluating at time $t = 0$, we obtain the admissible velocity field in $\mu_\theta$:

$$v_0(x) = \partial_\theta F_\theta\left( F_\theta^{-1}(x) \right) \dot{\theta}_0. \tag{167}$$

More generally, for any $h \in \mathbb{R}^p$, we define the associated admissible velocity field by

$$v_h(x) := \partial_\theta F_\theta\left( F_\theta^{-1}(x) \right) h. \tag{168}$$

**Tangent space.** The tangent space to $\mathcal{M}_\mu$ at $\mu_\theta$ is given by

$$T_{\mu_\theta}\mathcal{M}_\mu = \left\{ -\nabla \cdot (\mu_\theta v_h) : h \in \mathbb{R}^p \right\}, \tag{169}$$

where the associated velocity field satisfies $v_h \in L^2(\mu_\theta; \mathbb{R}^d)$ and write as (168). We therefore introduce the space of realizable velocity fields

$$V_\theta := \left\{ v_h : h \in \mathbb{R}^p \right\} \subset L^2(\mu_\theta; \mathbb{R}^d), \tag{170}$$

which is a finite-dimensional subspace of the Hilbert space $L^2(\mu_\theta; \mathbb{R}^d)$. We endow this space with the inner product induced by $L^2(\mu_\theta)$.

**Riemannian metric.** As in Section 6, we define the matrix

$$G(\theta) := \int \partial_\theta F_\theta(z)^\top \partial_\theta F_\theta(z) \, \mathrm{d}\mu(z) \in \mathbb{R}^{p \times p}. \tag{171}$$

Observe that this matrix naturally appears when computing the squared $L^2(\mu_\theta)$-norm of realizable velocity fields. Indeed, applying the transfer lemma,

$$\|v_h\|_{L^2(\mu_\theta)}^2 = \int \left\| \partial_\theta F_\theta \big( F_\theta^{-1}(x) \big) h \right\|^2 \mathrm{d}\mu_\theta(x) = h^\top G(\theta) h. \tag{172}$$

Assuming that $G(\theta)$ is positive definite, it defines a Riemannian metric on the parameter space $\Theta$ and, equivalently, on the manifold $\mathcal{M}_\mu$. More precisely, for two tangent vectors associated with parameters $h_1, h_2 \in \mathbb{R}^p$, we define

$$g_{\mu_\theta}(v_{h_1}, v_{h_2}) := \langle v_{h_1}, v_{h_2} \rangle_{L^2(\mu_\theta)} = h_1^\top G(\theta) h_2. \tag{173}$$

**Riemannian gradient.** We define the operator

$$(\widetilde{\mathcal{H}}_\theta f)(x) := \partial_\theta F_\theta \big( F_\theta^{-1}(x) \big) G(\theta)^{-1} \int \partial_\theta F_\theta(z)^\top f(z) \, \mathrm{d}\mu_\theta(z). \tag{174}$$

By Proposition 6.2, $\widetilde{\mathcal{H}}_\theta$ is the orthogonal projection from $L^2(\mu_\theta; \mathbb{R}^d)$ onto the subspace $V_\theta$.

**Proposition G.1.** *Let $\mathcal{F} : \mathcal{P}_2(\mathbb{R}^d) \to \mathbb{R}$ be a smooth functional. Then the Riemannian gradient of $\mathcal{F}$ on the manifold $\mathcal{M}_\mu$, with respect to the metric $g_{\mu_\theta}$, is given by*

$$\mathrm{grad}_{\mathcal{M}_\mu} \mathcal{F}(\mu_\theta) = -\nabla \cdot \left( \mu_\theta \, \widetilde{\mathcal{H}}_\theta \nabla \frac{\delta \mathcal{F}}{\delta \mu}(\mu_\theta) \right). \tag{175}$$

*Proof.* By definition of the Riemannian gradient, for every smooth curve $(\mu_t)_t$ in $\mathcal{M}_\mu$ such that $\mu_{t=0} = \mu_{\theta_0}$, we have

$$g_{\mu_{\theta_0}}\big(\mathrm{grad}_{\mathcal{M}_\mu} \mathcal{F}(\mu_{\theta_0}), \partial_t \mu_t|_{t=0}\big) = \frac{\mathrm{d}}{\mathrm{d}t} \mathcal{F}(\mu_t) \Big|_{t=0}. \tag{176}$$

Moreover, if

$$\partial_t \mu_t|_{t=0} \in T_{\mu_{\theta_0}} \mathcal{M}_\mu, \tag{177}$$

then there exists $h \in \mathbb{R}^p$ and $v_h \in V_{\theta_0}$ such that

$$\partial_t \mu_t \big|_{t=0} = -\nabla \cdot (\mu_{\theta_0} v_h). \tag{178}$$

Using the chain rule for functionals, we obtain

$$\frac{\mathrm{d}}{\mathrm{d}t} \mathcal{F}(\mu_t) \Big|_{t=0} = \int \frac{\delta \mathcal{F}}{\delta \mu}(\mu_{\theta_0})(x) \, \partial_t \mu_t(x) \Big|_{t=0} \mathrm{d}x. \tag{179}$$

Substituting the continuity equation yields

$$\frac{\mathrm{d}}{\mathrm{d}t} \mathcal{F}(\mu_t) \Big|_{t=0} = -\int \frac{\delta \mathcal{F}}{\delta \mu}(\mu_{\theta_0})(x) \nabla \cdot \big( \mu_{\theta_0}(x) v_h(x) \big) \, \mathrm{d}x. \tag{180}$$

An integration by parts gives

$$\frac{\mathrm{d}}{\mathrm{d}t} \mathcal{F}(\mu_t) \Big|_{t=0} = \int \left\langle \nabla \frac{\delta \mathcal{F}}{\delta \mu}(\mu_{\theta_0})(x), \, v_h(x) \right\rangle \mathrm{d}\mu_{\theta_0}(x). \tag{181}$$

Since $\widetilde{\mathcal{H}}_{\theta_0}$ is the orthogonal projection onto $V_{\theta_0}$ and $v_h \in V_{\theta_0}$, we may write

$$\frac{\mathrm{d}}{\mathrm{d}t} \mathcal{F}(\mu_t) \Big|_{t=0} = \int \left\langle \widetilde{\mathcal{H}}_{\theta_0} \left( \nabla \frac{\delta \mathcal{F}}{\delta \mu}(\mu_{\theta_0}) \right)(x), \, v_h(x) \right\rangle \mathrm{d}\mu_{\theta_0}(x). \tag{182}$$

Therefore,

$$\frac{\mathrm{d}}{\mathrm{d}t}\mathcal{F}(\mu_t)\Big|_{t=0} = \left\langle \widetilde{\mathcal{H}}_{\theta_0}\left(\nabla\frac{\delta\mathcal{F}}{\delta\mu}(\mu_{\theta_0})\right), v_h \right\rangle_{L^2(\mu_{\theta_0})}. \tag{183}$$

On the other hand, by definition of the Riemannian metric,

$$g_{\mu_{\theta_0}}\left(\mathrm{grad}_{\mathcal{M}_\mu}\mathcal{F}(\mu_{\theta_0}), \partial_t\mu_t|_{t=0}\right) = \langle v, v_h \rangle_{L^2(\mu_{\theta_0})}, \tag{184}$$

where $v \in V_{\theta_0}$ is the velocity field associated with $\mathrm{grad}_{\mathcal{M}_\mu}\mathcal{F}(\mu_{\theta_0})$, namely

$$\mathrm{grad}_{\mathcal{M}_\mu}\mathcal{F}(\mu_{\theta_0}) = -\nabla\cdot(\mu_{\theta_0}v). \tag{185}$$

Since the above identity holds for every $h \in \mathbb{R}^p$, hence for every $v_h \in V_{\theta_0}$, we conclude that

$$v = \widetilde{\mathcal{H}}_{\theta_0}\left(\nabla\frac{\delta\mathcal{F}}{\delta\mu}(\mu_{\theta_0})\right), \quad \text{hence } \mathrm{grad}_{\mathcal{M}_\mu}\mathcal{F}(\mu_{\theta_0}) = -\nabla\cdot\left(\mu_{\theta_0}\widetilde{\mathcal{H}}_{\theta_0}\left(\nabla\frac{\delta\mathcal{F}}{\delta\mu}(\mu_{\theta_0})\right)\right), \tag{186}$$

which concludes the proof. $\qquad\square$

This computation highlights the necessity of considering the preconditioned flow

$$\dot{\theta}_t = -G(\theta_t)^{-1}\nabla_\theta\mathcal{F}(\mu_{\theta_t}), \tag{187}$$

as introduced in Section 6. Without this correction, the operator $\mathcal{H}_\theta$ defined in Proposition 6.1 is not, in general, a projection operator, and the associated velocity field $\mathcal{H}_\theta\left(\nabla\frac{\delta\mathcal{F}}{\delta\mu}(\mu_\theta)\right)$ does not correspond to the opposite of a proper Riemannian gradient on $\mathcal{M}_\mu$. This distinction reflects the difference between the Euclidean gradient in parameter coordinates and the intrinsic Riemannian gradient associated with the metric tensor $G(\theta)$.

**Proposition G.2.** *The operator $\mathcal{H}_\theta$, defined in Proposition 6.1, is a projection if and only if*

$$G(\theta) = I_p. \tag{188}$$

*In general, $\mathcal{H}_\theta$ is not an orthogonal projector.*

*Proof.* For any $f \in L^2(\mu_\theta; \mathbb{R}^d)$, a direct computation yields

$$\mathcal{H}_\theta\big(\mathcal{H}_\theta f\big)(x) = \partial_\theta F_\theta\big(F_\theta^{-1}(x)\big)\int \partial_\theta F_\theta\big(F_\theta^{-1}(z)\big)^\top\left(\partial_\theta F_\theta\big(F_\theta^{-1}(z)\big)\int \partial_\theta F_\theta\big(F_\theta^{-1}(y)\big)^\top f(y)\,\mathrm{d}\mu_\theta(y)\right)\mathrm{d}\mu_\theta(z) \tag{189}$$

$$= \partial_\theta F_\theta\big(F_\theta^{-1}(x)\big)\,G(\theta)\int \partial_\theta F_\theta\big(F_\theta^{-1}(y)\big)^\top f(y)\,\mathrm{d}\mu_\theta(y). \tag{190}$$

Therefore, $\mathcal{H}_\theta^2 = \mathcal{H}_\theta$ if and only if $G(\theta) = I_p$. $\qquad\square$

### G.2. Relation with Wasserstein Natural Gradient Flows

In the previous subsection, the metric tensor,

$$G(\theta) := \int \partial_\theta F_\theta(z)^\top \partial_\theta F_\theta(z)\,\mathrm{d}\mu(z), \tag{191}$$

was introduced as the $L^2(\mu_\theta)$ inner product of velocity fields generated by the parametrization.

We now relate this construction to the metric tensor introduced in (Li & Montúfar, 2018; Li & Zhao, 2019; Li et al., 2019; Chen & Li, 2020), which is defined as the exact pullback of the Wasserstein metric onto the parametric manifold.

**Wasserstein Metric.** We briefly recall the definition of the Wasserstein Riemannian metric, see *e.g.* (Li & Zhao, 2019) for more details. Let $\mu \in \mathcal{P}_2(\mathbb{R}^d)$. For tangent vectors $\xi_1, \xi_2 \in T_\mu\mathcal{P}_2(\mathbb{R}^d)$ represented as $\xi_1 = -\mathrm{div}(\mu\nabla\psi_1)$ and $\xi_2 = -\mathrm{div}(\mu\nabla\psi_2)$ with $\psi_1, \psi_2 : \mathbb{R}^d \to \mathbb{R} \in C_c^\infty(\mathbb{R}^d)$, the Wasserstein metric tensor $g_\mu^W : T_\mu\mathcal{P}_2(\mathbb{R}^d) \times T_\mu\mathcal{P}_2(\mathbb{R}^d) \to \mathbb{R}$ is defined as

$$g_\mu^W(\xi_1, \xi_2) = \int \nabla\psi_1(x)^\top \nabla\psi_2(x)\,\mathrm{d}\mu(x). \tag{192}$$

**Exact Pullback Metric.** Let

$$\Phi : \Theta \to \mathcal{P}_2(\mathbb{R}^d), \qquad \theta \mapsto \mu_\theta = (F_\theta)_\# \mu. \tag{193}$$

The Wasserstein information matrix (Li & Zhao, 2019) is defined as the pullback of the Wasserstein metric $g_{\mu_\theta}^{\mathrm{W}}$:

$$\big(G_{\mathrm{W}}(\theta)\big)_{ij} = g_{\mu_\theta}^{\mathrm{W}}\big(d\Phi_\theta[e_i], d\Phi_\theta[e_j]\big), \tag{194}$$

where $d\Phi_\theta : T_\theta \Theta \to T_{\mu_\theta} \mathcal{P}_2(\mathbb{R}^d)$ denotes the differential of the map $\theta \mapsto \mu_\theta$ and $(e_i)_{i=1}^p$ the canonical basis of $\mathbb{R}^p$. Then $d\Phi_\theta[e_i] = \partial_{\theta_i} \mu_\theta$ denotes the tangent vector induced by varying the parameter $\theta_i$. As shown in Appendix G.1, a parameter variation in direction $h \in \mathbb{R}^p$ induces the velocity field $v_h(x) = \partial_\theta F_\theta\big(F_\theta^{-1}(x)\big)h$. In particular, for $h = e_i$, the velocity field associated with $d\Phi_\theta[e_i]$ is $x \mapsto \partial_{\theta_i} F_\theta\big(F_\theta^{-1}(x)\big)$. Hence $d\Phi_\theta[e_i] = \partial_{\theta_i}\mu_\theta = -\nabla \cdot \big(\mu_\theta \partial_{\theta_i} F_\theta\big(F_\theta^{-1}(x)\big)\big)$, and

$$\big(G_{\mathrm{W}}(\theta)\big)_{ij} = \int \nabla \psi_i(x)^\top \nabla \psi_j(x) \, \mathrm{d}\mu_\theta(x), \tag{195}$$

where $\nabla \psi_i$ is the Wasserstein representative of $d\Phi_\theta[e_i]$, *i.e.* $\psi_i$ solves

$$\nabla \cdot \big(\mu_\theta \nabla \psi_i(x)\big) = \nabla \cdot \big(\mu_\theta \partial_{\theta_i} F_\theta\big(F_\theta^{-1}(x)\big)\big). \tag{196}$$

**Explicit form of the metric tensor.** Li et al. (2019) state that, in dimension one, the metric tensor $G_{\mathrm{W}}$ coincides with $G$. We show below that, more generally, it coincides when the velocity fields generated by the parametrization are gradients of scalar potentials.

**Proposition G.3.** *Assume that the velocity fields generated by the parametrization satisfy*

$$\partial_{\theta_i} F_\theta\big(F_\theta^{-1}(x)\big) = \nabla \phi_i(x) \tag{197}$$

*for some scalar potentials $\phi_i$, with $\nabla \phi_i \in L^2(\mu_\theta)$, such that the integration by parts below is valid. Then the metric tensor introduced in (Li & Zhao, 2019) coincides with the matrix defined in the previous subsection:*

$$G_{\mathrm{W}}(\theta) = G(\theta). \tag{198}$$

*Proof.* By definition of the pullback of the Wasserstein metric, the metric tensor reads

$$\big(G_{\mathrm{W}}(\theta)\big)_{ij} = \int \nabla \psi_i(x) \cdot \nabla \psi_j(x) \, \mathrm{d}\mu_\theta(x), \tag{199}$$

where the potentials $\psi_i$ satisfy

$$\nabla \cdot \big(\mu_\theta \nabla \psi_i\big) = \nabla \cdot \Big(\mu_\theta \, \partial_{\theta_i} F_\theta\big(F_\theta^{-1}(x)\big)\Big). \tag{200}$$

By assumption, there exists a scalar potential $\phi_i$ such that

$$\partial_{\theta_i} F_\theta\big(F_\theta^{-1}(x)\big) = \nabla \phi_i(x). \tag{201}$$

Therefore both potentials $\psi_i$ and $\phi_i$ satisfy the same equation and we have

$$\nabla \cdot \big(\mu_\theta \nabla \psi_i\big) = \nabla \cdot \big(\mu_\theta \nabla \phi_i\big). \tag{202}$$

Subtracting the two equations gives

$$\nabla \cdot \big(\mu_\theta \nabla(\psi_i - \phi_i)\big) = 0. \tag{203}$$

Testing this equation against $\psi_i - \phi_i$ and integrating by parts, which is valid by assumption, yields

$$\int \|\nabla(\psi_i - \phi_i)(x)\|^2 \, \mathrm{d}\mu_\theta(x) = 0, \tag{204}$$

Hence $\nabla \psi_i = \nabla \phi_i = \partial_{\theta_i} F_\theta\big(F_\theta^{-1}(x)\big)$ $\mu_\theta$-a.e.

Therefore

$$\left(G_{\mathrm{W}}(\theta)\right)_{ij} = \int \partial_{\theta_i} F_\theta\left(F_\theta^{-1}(x)\right) \cdot \partial_{\theta_j} F_\theta\left(F_\theta^{-1}(x)\right) \mathrm{d}\mu_\theta(x). \tag{205}$$

Using a change of variable we obtain

$$\left(G_{\mathrm{W}}(\theta)\right)_{ij} = \int \partial_{\theta_i} F_\theta(z) \cdot \partial_{\theta_j} F_\theta(z) \, \mathrm{d}\mu(z). \tag{206}$$

In matrix form, this yields

$$G_{\mathrm{W}}(\theta) = \int \partial_\theta F_\theta(z)^\top \partial_\theta F_\theta(z) \, \mathrm{d}\mu(z), \tag{207}$$

which coincides with the matrix $G(\theta)$ defined in the previous subsection. $\qquad\square$

**Interpretation.** The previous result shows that the matrix $G(\theta)$ introduced in the previous subsection coincides with the metric tensor $G_{\mathrm{W}}(\theta)$ only under an additional structural assumption on the parametrization.

Therefore, in the general case, the matrix $G(\theta)$ should be interpreted as the metric induced by the parametrization through the $L^2(\mu_\theta)$ inner product on realizable velocity fields, and it does not coincide with the Wasserstein information matrix.

### G.3. Proof of Proposition 6.3

In practice, as discussed in Section 3.2, solving the parametrized JKO scheme

$$\begin{cases} \theta_{\ell+1} \in \underset{\theta}{\operatorname{argmin}} \; \mathcal{F}\left((F_\theta)_{\#}\mu_\ell\right) + \dfrac{1}{2\tau}\|F_\theta - \mathrm{Id}\|_{L^2(\mu_\ell)}^2, \\ \mu_{\ell+1} = (F_{\theta_{\ell+1}})_{\#}\mu_\ell, \end{cases} \tag{208}$$

is costly because it involves repeated compositions of transport maps.

To avoid this, a reparametrization trick (Section 3.2) allows us to rewrite the problem in terms of a fixed base measure $\mu$, leading to the equivalent optimization problem

$$\theta_{\ell+1} \in \underset{\theta\in\mathbb{R}^p}{\operatorname{argmin}} \; \Phi_\ell(\theta) := \frac{1}{2\tau}\|F_\theta - F_{\theta_\ell}\|_{L^2(\mu)}^2 + \mathcal{F}(\mu_\theta), \qquad \mu_\theta = (F_\theta)_{\#}\mu. \tag{209}$$

The following proposition clarifies the connection between the reparametrized JKO update and the preconditioned gradient flow associated with the functional $\mathcal{F}$.

**Proposition G.4.** *Let $\mu \in \mathcal{P}_2(\mathbb{R}^d)$ and $F_\theta : \mathbb{R}^d \to \mathbb{R}^d$ be a parametric map. Define the pushforward measure $\mu_\theta := F_\theta\#\mu$ and consider the objective $\theta \mapsto \mathcal{F}(\mu_\theta)$. Let $(\theta_\ell)_{\ell\geq 0}$ be the solutions of (209). Assume*

1. *$L^2(\mu)$-**Fréchet differentiability of $\theta \mapsto F_\theta$**: For every $\theta$ there exists a linear operator $h \mapsto \partial_\theta F_\theta h \in L^2(\mu; \mathbb{R}^d)$ such that*

$$\|F_{\theta+h} - F_\theta - \partial_\theta F_\theta h\|_{L^2(\mu)} = o(\|h\|_2) \qquad \text{as } h \to 0.$$

2. ***Square-integrability of the Jacobian:** For every $\theta$ in a neighborhood of $\theta_\ell$,*

$$\partial_\theta F_\theta \in L^2(\mu), \qquad \|\partial_\theta F_\theta\|_{L^2(\mu)} < \infty.$$

3. ***Bound on increment of parameters:** For all $\ell \geq 0$, $\|\theta_{\ell+1} - \theta_\ell\|_2 = O(\tau)$ as $\tau \to 0$.*

*Then, we have the expansion*

$$G(\theta_{\ell+1}) \frac{(\theta_{\ell+1} - \theta_\ell)}{\tau} = -\nabla_\theta \mathcal{F}(\mu_{\theta_{\ell+1}}) + o(1) \qquad \text{as } \tau \to 0. \tag{210}$$

*Consequently, when $G$ is invertible and $(\theta_{\ell+1} - \theta_\ell)/\tau \to \dot\theta_t$ for $t = \ell\tau$ as $\tau \to 0$, the update rule defines, at first order, a backward (implicit) discretization of the preconditioned gradient flow*

$$\dot\theta_t = -G(\theta_t)^{-1} \nabla_\theta \mathcal{F}(\mu_{\theta_t}). \tag{211}$$

*Proof.* Fix $\ell \geq 0$ and define $\Delta_\ell := \theta_{\ell+1} - \theta_\ell \in \mathbb{R}^p$. Since $\theta_{\ell+1}$ minimizes $\Phi_\ell$ in (209), the first-order optimality condition yields

$$\nabla_\theta \Phi_\ell(\theta_{\ell+1}) = 0. \tag{212}$$

Write

$$\Phi_\ell(\theta) = \frac{1}{\tau} Q_\ell(\theta) + \mathcal{F}(\mu_\theta), \qquad Q_\ell(\theta) := \frac{1}{2}\|F_\theta - F_{\theta_\ell}\|^2_{L^2(\mu)}.$$

Differentiating with respect to $\theta$ gives

$$\nabla_\theta Q_\ell(\theta) = \int \left(\partial_\theta F_\theta(z)\right)^\top \left(F_\theta(z) - F_{\theta_\ell}(z)\right) \mathrm{d}\mu(z). \tag{213}$$

Thus (212) becomes

$$\frac{1}{\tau} \int \left(\partial_\theta F_{\theta_{\ell+1}}(z)\right)^\top \left(F_{\theta_{\ell+1}}(z) - F_{\theta_\ell}(z)\right) \mathrm{d}\mu(z) + \nabla_\theta \mathcal{F}(\mu_{\theta_{\ell+1}}) = 0. \tag{214}$$

**Step 1: Linearization of the metric term.** Using the $L^2(\mu)$-Fréchet differentiability of the mapping $\theta \mapsto F_\theta$, we can write, $\mu$-almost everywhere

$$F_{\theta_{\ell+1}} - F_{\theta_\ell} = \partial_\theta F_{\theta_{\ell+1}} \Delta_\ell + r_\ell, \qquad \|r_\ell\|_{L^2(\mu)} = o(\|\Delta_\ell\|_2). \tag{215}$$

Plugging (215) into the integral term in (214) yields

$$\int (\partial_\theta F_{\theta_{\ell+1}})^\top (F_{\theta_{\ell+1}} - F_{\theta_\ell}) \, \mathrm{d}\mu = \int (\partial_\theta F_{\theta_{\ell+1}})^\top (\partial_\theta F_{\theta_{\ell+1}} \Delta_\ell) \, \mathrm{d}\mu + \int (\partial_\theta F_{\theta_{\ell+1}})^\top r_\ell \, \mathrm{d}\mu. \tag{216}$$

The first term equals

$$G(\theta_{\ell+1})\Delta_\ell.$$

The second term can be bounded using Cauchy–Schwarz:

$$\left| \int (\partial_\theta F_{\theta_{\ell+1}})^\top r_\ell \, \mathrm{d}\mu \right| \leq \|\partial_\theta F_{\theta_{\ell+1}}\|_{L^2(\mu)} \|r_\ell\|_{L^2(\mu)} = o(\|\Delta_\ell\|_2),$$

using the square-integrability of the Jacobian.

Therefore,

$$\int (\partial_\theta F_{\theta_{\ell+1}})^\top (F_{\theta_{\ell+1}} - F_{\theta_\ell}) \, \mathrm{d}\mu = G(\theta_{\ell+1})\Delta_\ell + o(\|\Delta_\ell\|_2). \tag{217}$$

Plugging (217) into (214) yields

$$\frac{1}{\tau}\Big(G(\theta_{\ell+1})\Delta_\ell + o(\|\Delta_\ell\|_2)\Big) + \nabla_\theta \mathcal{F}(\mu_{\theta_{\ell+1}}) = 0.$$

**Step 2: Limit as $\tau \to 0$.** By definition of the little-$o$ notation, there exists a function $\varepsilon(s) \to 0$ as $s \to 0$ such that

$$o(\|\Delta_\ell\|_2) = \varepsilon(\|\Delta_\ell\|_2) \|\Delta_\ell\|_2.$$

Therefore,

$$\frac{o(\|\Delta_\ell\|_2)}{\tau} = \varepsilon(\|\Delta_\ell\|_2) \frac{\|\Delta_\ell\|_2}{\tau}.$$

By Assumption 3, $\|\Delta_\ell\|_2 = O(\tau)$ and thus $\|\Delta_\ell\|_2 \to 0$ and $\|\Delta_\ell\|_2/\tau = O(1)$ as $\tau \to 0$. Hence,

$$\frac{o(\|\Delta_\ell\|_2)}{\tau} = o(1).$$

We finally obtain

$$G(\theta_{\ell+1})\frac{\Delta_\ell}{\tau} = -\nabla_\theta \mathcal{F}(\mu_{\theta_{\ell+1}}) + o(1), \qquad (\tau \to 0),$$

This shows that the proximal update defines, at first order, an implicit (backward Euler-type) discretization of the preconditioned gradient flow (211). $\qquad\square$

We next provide assumptions on the map $\theta \mapsto F_\theta$ under which Assumption 3 in Proposition G.4 is satisfied.

**Proposition G.5.** *Let $(\theta_\ell)_{\ell \geq 0}$ be the solutions of (209), and assume 1 and 2 from Proposition G.4. Additionally, assume*

1. ***Local regularity of the objective:*** *The map $\theta \mapsto \mathcal{F}(\mu_\theta)$ is differentiable in a neighborhood of $\theta_\ell$, and its gradient is continuous.*

2. ***Non-degeneracy of the parametrization:*** *There exists $c > 0$ such that for all $\theta \in \mathbb{R}^p$,*

$$\|F_\theta - F_{\theta_\ell}\|^2_{L^2(\mu)} \geq c\|\theta - \theta_\ell\|^2_2,$$

*then Assumption 3 in Proposition G.4 is satisfied.*

*Proof.* Since $\theta_{\ell+1}$ minimizes $\Phi_\ell$, we have $\Phi_\ell(\theta_{\ell+1}) \leq \Phi_\ell(\theta_\ell)$, this yields

$$\frac{1}{2\tau}\|F_{\theta_{\ell+1}} - F_{\theta_\ell}\|^2_{L^2(\mu)} \leq \mathcal{F}(\mu_{\theta_\ell}) - \mathcal{F}(\mu_{\theta_{\ell+1}}). \tag{218}$$

Since $\theta \mapsto \mathcal{F}(\mu_\theta)$ is differentiable with continuous gradient in a neighborhood of $\theta_\ell$, it is locally Lipschitz there. Therefore, for $\tau$ small enough, there exists $C > 0$ such that

$$\mathcal{F}(\mu_{\theta_\ell}) - \mathcal{F}(\mu_{\theta_{\ell+1}}) \leq C\|\theta_{\ell+1} - \theta_\ell\|_2 = C\|\Delta_\ell\|_2.$$

Moreover, by the non-degeneracy assumption,

$$\|F_{\theta_{\ell+1}} - F_{\theta_\ell}\|^2_{L^2(\mu)} \geq c\|\theta_{\ell+1} - \theta_\ell\|^2_2 = c\|\Delta_\ell\|^2_2.$$

Combining the last two inequalities with (218), we obtain

$$\frac{c}{2\tau}\|\Delta_\ell\|^2_2 \leq C\|\Delta_\ell\|_2.$$

If $\Delta_\ell \neq 0$, dividing by $\|\Delta_\ell\|_2$ gives

$$\|\Delta_\ell\|_2 \leq \frac{2C}{c}\tau.$$

Hence,

$$\|\Delta_\ell\|_2 = O(\tau) \qquad \text{as } \tau \to 0. \tag{219}$$

$\square$

As a direct consequence of Proposition G.4, we obtain the following interpretation. When solving the reparametrized JKO scheme (209) in practice, we are not merely performing a Euclidean descent in parameter space. Rather, in the small step regime $\tau \to 0$, the induced dynamics at the distribution level can be interpreted as a projected Wasserstein gradient flow of $\mathcal{F}$. Accordingly, the parameter dynamics are consistent, at first order, with the preconditioned gradient flow

$$\dot{\theta}(t) = -G\big(\theta(t)\big)^{-1}\nabla_\theta\mathcal{F}(\mu_{\theta(t)}).$$

Therefore, although $G(\theta)$ is never formed explicitly in practice, the proximal structure of the JKO step implicitly introduces a Wasserstein preconditioning of the dynamics when $\tau$ is small.

**Strength of the assumptions and practical relevance.** The assumptions used in Proposition G.5 are admittedly strong and are generally not satisfied in large-scale neural network models. In particular, the invertibility of the matrix $G(\theta)$ and the non-degeneracy of the parametrization $\theta \mapsto F_\theta$ are structural conditions that rarely hold globally in highly overparameterized settings.

The invertibility assumption on $G(\theta)$ can be relaxed in practice by replacing the inverse with the Moore–Penrose pseudoinverse, as commonly done in (Zuo et al., 2025; Dumont et al., 2026).

The non-degeneracy condition is more restrictive. It requires that small changes in the model output imply proportionally small changes in the parameters, which may fail for highly non-convex parametrizations such as neural networks.

Despite these limitations, the result remains informative beyond the idealized setting considered in the proposition. In practice, even when the assumptions are violated, the parametric JKO updates empirically exhibit dynamics that closely follow the predicted preconditioned gradient flow when the step size $\tau$ is small, see Appendix J.1.

Therefore, while the assumptions are not expected to hold exactly in neural network architectures, the analysis provides theoretical insight into the mechanism underlying the method and helps explain the stability and qualitative behavior observed in experiments.

### G.4. Application to the Bures–Wasserstein geometry

We now illustrate the geometric constructions introduced in the previous subsections on a concrete example, namely the family of Gaussian distributions endowed with the Bures–Wasserstein metric. We show that, when restricted to Gaussian distributions with either isotropic covariance matrices or diagonal covariance matrices, the two metrics $G$ and $G_{\mathrm{W}}$, introduced above, coincide.

**The Isotropic Bures–Wasserstein metric.** We consider the manifold of Gaussian distributions with isotropic covariance matrices

$$\mathrm{IG} := \left\{ \mathcal{N}(m, \sigma^2 I_d) : m \in \mathbb{R}^d, \quad \sigma > 0 \right\}.$$

For two Gaussian measures $\mu_1 = \mathcal{N}(m_1, \sigma_1^2 I_d)$, $\mu_2 = \mathcal{N}(m_2, \sigma_2^2 I_d)$,

the squared Bures-Wasserstein distance reduces to

$$\mathrm{W}_2^2(\mu_1, \mu_2) = \|m_1 - m_2\|^2 + d(\sigma_1 - \sigma_2)^2.$$

This expression defines a Riemannian metric on the manifold IG, called the Bures–Wasserstein metric.

**Gaussian parametrization.** Let the reference distribution be the standard Gaussian $\mu = \mathcal{N}(0, I_d)$, and consider linear transformations

$$F_\theta(x) = \sigma x + m, \qquad \theta = (m, \sigma), \text{ where } m \in \mathbb{R}^d, \qquad \sigma > 0. \tag{220}$$

Since the map $F_\theta$ is linear with invertible matrix $\sigma I_d$, it admits a smooth inverse, which allows us to write explicitly the velocity field $\partial_\theta F_\theta \big( F_\theta^{-1}(x) \big)$. In particular, the pushforward distribution is given by $\mu_\theta = F_{\theta \#} \mu = \mathcal{N}(m, \sigma^2 I_d)$.

**Metric tensor.** The Jacobian of the parametrization satisfies

$$\partial_\theta F_\theta \big( F_\theta^{-1}(x) \big) h = a + \frac{s}{\sigma}(x - m), \tag{221}$$

for directions $h = (a, s)$, $a \in \mathbb{R}^d$, $s \in \mathbb{R}$.

The associated metric tensor therefore reads

$$G(\theta) = \int \partial_\theta F_\theta(z)^\top \partial_\theta F_\theta(z) \, \mathrm{d}\mu(z) = \begin{pmatrix} I_d & 0 \\ 0 & d \end{pmatrix}. \tag{222}$$

**Proposition G.6.** *In the isotropic Gaussian setting, the matrix $G(\theta)$ coincides with the intrinsic Bures–Wasserstein metric tensor on the manifold* IG.

*Proof.* From the previous computation,

$$v_h(x) = \partial_\theta F_\theta \big( F_\theta^{-1}(x) \big) h = a + \frac{s}{\sigma}(x - m). \tag{223}$$

This velocity field is the gradient of the quadratic function

$$\psi(x) = a^\top x + \frac{s}{2\sigma} \|x - m\|^2. \tag{224}$$

Hence admissible velocity fields belong to the gradient subspace of the Wasserstein tangent space. By Proposition G.3, the induced metric tensor therefore coincides with the pullback of the Wasserstein metric, which is the Bures–Wasserstein metric on the Gaussian manifold. □

**Bures–Wasserstein gradient.** Let

$$g(x) := \nabla \frac{\delta \mathcal{F}}{\delta \mu}(\mu_\theta)(x) \tag{225}$$

denote the Wasserstein gradient.

The Riemannian gradient on the parametric manifold, with respect to our metric, is given in Proposition G.1 by

$$v_\theta = -\widetilde{\mathcal{H}}_\theta g. \tag{226}$$

We now compute this operator explicitly.

From the definition of the projection operator,

$$\widetilde{\mathcal{H}}_\theta g = \partial_\theta F_\theta \big(F_\theta^{-1}(\cdot)\big) G(\theta)^{-1} \int \partial_\theta F_\theta(z)^\top g\big(F_\theta(z)\big)\, \mathrm{d}\mu(z). \tag{227}$$

Using the explicit form, for $h = (a, s)$,

$$\partial_\theta F_\theta\big(F_\theta^{-1}(x)\big) h = a + \frac{s}{\sigma}(x - m), \tag{228}$$

we obtain

$$\nabla_{\mathcal{M}_\mu} \mathcal{F}(\mu_\theta)(x) = \alpha + \beta(x - m), \tag{229}$$

with coefficients

$$\alpha = \int g(x)\, \mathrm{d}\mu_\theta(x), \quad \beta = \frac{1}{d\sigma^2} \int g(x) \cdot (x - m)\, \mathrm{d}\mu_\theta(x). \tag{230}$$

This is the well-known expression of the Bures–Wasserstein gradient, see, e.g., (Petit-Talamon et al., 2026, Appendix A.4). Hence, we recover it here as a direct application of the projection operator introduced in Section 6.

The same construction can be carried out in the diagonal Gaussian setting.

## H. Practical Algorithm for GWF

In this appendix, we detail the practical ingredients required to implement the Generative Wasserstein Flow (GWF) framework introduced in the main text. We successively describe the losses associated with each divergence, the regularization strategies used for the discriminator, and a generic training algorithm covering all considered objectives. We denote by $h_\phi$ a neural network modeling the critic, and by $\mathrm{T}_\theta$ a neural network modeling the transport map (or generator).

**Losses.** As explained in Section 3 and Section 4, GWF can be derived for IPMs and $f$-divergences. We detail here the losses used in our implementation for the different divergences considered throughout the experiments. Let $(y_j)_{j=1}^n \sim \nu$ denote real samples and $(z_j)_{j=1}^n \sim \mu_0$ noise samples, with generated samples given by $x_j = \mathrm{T}_\theta(z_j)$. We denote by $d_{\text{real},j} = h_\phi(y_j)$ and $d_{\text{fake},j} = h_\phi(x_j)$ the discriminator outputs on real and generated samples, respectively.

**Losses for $f$-divergences.** For $f$-divergences, the losses directly follow from the semi-dual of the UOT formulation (9) and the variational representations derived in Appendix C.1. They coincide with the standard $f$-GAN objectives applied to the reverse divergence, with a JKO regularization term.

$$\mathcal{L}_h^{\text{KL}} = \frac{1}{n} \sum_{j=1}^{n} \left( d_{\text{fake},j} + e^{-d_{\text{real},j}-1} \right),$$

$$\mathcal{L}_h^{\text{KL-DV}} = \frac{1}{n} \sum_{j=1}^{n} d_{\text{fake},j} \; + \; \log\left( \frac{1}{n} \sum_{j=1}^{n} e^{-d_{\text{real},j}} \right),$$

$$\mathcal{L}_h^{\text{JS}} = \frac{1}{n} \sum_{j=1}^{n} \left( (d_{\text{fake},j})_+ + (-d_{\text{real},j})_+ \right),$$

$$\mathcal{L}_h^{\chi^2} = \frac{1}{n} \sum_{j=1}^{n} \left( d_{\text{fake},j} + \tfrac{1}{4} d_{\text{real},j}^2 - d_{\text{real},j} \right).$$

$$\mathcal{L}_{\text{T}}^{f\text{-div}} = -\frac{1}{n} \sum_{j=1}^{n} d_{\text{fake},j}.$$

**Losses for $\text{MMD}^2$ and kernel-based objectives.** For the squared MMD objective, the losses follow from the equivalence between our Algorithm 2 and the JKO-regularized MMD GAN, presented in Appendix D.4. We denote by $k$ the (possibly learned) kernel described in Appendix I.

$$\mathcal{L}_h^{\text{MMD}} = -\text{MMD}_k^2(d_{\text{real}}, d_{\text{fake}}),$$
$$\mathcal{L}_{\text{T}}^{\text{MMD}} = \text{MMD}_k^2(d_{\text{real}}, d_{\text{fake}}).$$

In our experiments, we also consider alternative objectives related to $\text{MMD}^2$, namely sMMD (Arbel et al., 2018) and ckMMD (Zhang et al., 2025). The sMMD objective controls the Lipschitz constant of the critic through a learnable scaling factor, while ckMMD considers an extended function class, resulting in a computable lower bound of the MMD. These variants lead to the following losses.

$$\mathcal{L}_h^{\text{sMMD}} = -s \, \text{MMD}_k^2(d_{\text{real}}, d_{\text{fake}}),$$
$$\mathcal{L}_{\text{T}}^{\text{sMMD}} = s \, \text{MMD}_k^2(d_{\text{real}}, d_{\text{fake}}),$$

where $s$ denotes the scaling factor introduced in Arbel et al. (2018).

$$\mathcal{L}_h^{\text{ckMMD}} = -\frac{1}{n} \sum_{j=1}^{n} \left( k(z_j, d_{\text{real},j}) - k(z_j, d_{\text{fake},j}) \right),$$

$$\mathcal{L}_{\text{T}}^{\text{ckMMD}} = -\frac{1}{n} \sum_{j=1}^{n} k(z_j, d_{\text{fake},j}).$$

**Losses for Wasserstein-1.** As discussed in Section 4, the GWF framework can also be instantiated for Integral Probability Metrics (IPMs) such as the Wasserstein-1 distance. In this case, we rely on the semi-unbalanced optimal transport (sUOT) formulation for IPMs derived in Appendix D.1,

$$\text{sUOT}_c(\mu, \nu) = \max_{g \in \mathcal{G}(\lambda_2)} \int g^c \, d\mu + \int g \, d\nu. \tag{231}$$

At the $\ell$-th JKO step, the $c$-transform $g^c$ can be parameterized by a measurable map T as $g^c(x) = \inf_{\mathrm{T}} c\big(x, \mathrm{T}(x)\big) - g\big(\mathrm{T}(x)\big)$. Introducing the rescaled potential $h = \lambda_2 g = 2\tau g$, this yields the following problem:

$$\sup_{h \in \mathcal{G}(1)} \inf_{\mathrm{T}} \int \left( \tfrac{1}{2\tau} \|\mathrm{Id} - \mathrm{T}\|_2^2 - h \circ \mathrm{T} \right) \mathrm{d}\mu_\ell + \int h \, \mathrm{d}\nu. \tag{232}$$

Enforcing the 1-Lipschitz constraint on the discriminator therefore leads to the following losses (without the JKO regularization):

$$\mathcal{L}_h^{\mathrm{W}_1} = \frac{1}{n} \sum_{j=1}^{n} \Big( d_{\mathrm{fake},j} - d_{\mathrm{real},j} \Big),$$

$$\mathcal{L}_{\mathrm{T}}^{\mathrm{W}_1} = -\frac{1}{n} \sum_{j=1}^{n} d_{\mathrm{fake},j}.$$

In practice, this corresponds to a JKO-regularized WGAN.

**Regularization of the discriminator.** GWF are closely related to GAN-based models (see Section 3.2). For all considered divergences, the discriminator is required to satisfy a certain degree of regularity in order to stabilize the adversarial optimization, as in GANs. In particular, for the Wasserstein-1 distance, the discriminator must be 1-Lipschitz.

During training, we enforce this regularity by adding an $R_1$ gradient penalty to the discriminator loss. The resulting discriminator objective takes the form

$$\mathcal{L}_h \; \leftarrow \; \mathcal{L}_h \; + \; \lambda_{\mathrm{gp}} (\|\nabla_z h_\phi(z)\|_2 - c)^2, \tag{233}$$

where $\lambda_{\mathrm{gp}}$ denotes the gradient penalty weight, and $z$ denotes either a real sample $y \sim \nu$ or an interpolated sample between real and generated data. $c = 1$ only when the chosen divergence is $\mathrm{W}_1$ (and $c = 0$ otherwise).

**GWF Algorithm.** Using the reparametrization trick introduced in Section 3.2, together with the losses derived above and the discriminator regularization, we can define a generic training algorithm for Generative Wasserstein Flows. The resulting procedure is summarized in Algorithm 3. It corresponds to a JKO regularization of the reverse $f$-divergence GAN objectives, and to a JKO regularization of GAN training for symmetric objectives such as $\mathrm{MMD}^2$ and $\mathrm{W}_1$.

## I. Implementation Details

In this section, we provide implementation details for all experiments presented in the paper. We describe the architectures used for the transport maps and critics, the optimization of kernels for MMD-based objectives, the training and optimization setup, and the evaluation protocol used during experiments.

**Architectures for generators and discriminators.** Throughout our experiments, we use several backbone architectures to parametrize both the transport map T (generator) and the critic $h$. Each generator is paired with a discriminator of comparable capacity, i.e., with the same order of magnitude in parameter count and standard architectural design choices.

- **U-Net ("UNet").** The generator is a U-Net with approximately 18M parameters, adapted with minor modifications from the diffusion U-Net architecture of Ho et al. (2020). The corresponding discriminator is a ResNet-style architecture with stride-2 downsampling stages, an attention block at the $8 \times 8$ (or $7 \times 7$) resolution, global sum pooling, and a linear head. The base channel width of the discriminator is decoupled from that of the generator.

- **NCSN++-style ("LargeNet").** The generator is a large network with approximately 48M parameters, adapted with light modifications from the public score-based implementation of Song et al. (2021), and previously used in Choi et al. (2024). The associated discriminator follows a small-image NCSN++-style design, with successive downsampling blocks, a minibatch standard-deviation channel, global sum pooling, and a linear head.

- **ResNet ("SmallNet").** The generator is a compact residual network with approximately 1M parameters, also used in Choi et al. (2024). The corresponding discriminator is a lightweight ResNet composed of an initial downsampling residual block, followed by several residual blocks, ReLU nonlinearities, global sum pooling, and a linear head.

---

**Algorithm 3** Training Procedure of Generative Wasserstein Flows

---

**Inputs:** number of outer (JKO) steps $K$, number of inner steps $N$, step size $\tau$, divergence $D$, steps sizes $\eta_h$ and $\eta_{\mathrm{T}}$
Initialize $\mathrm{T}_{\theta_0} = \mathrm{Id}$
**for** $\ell = 1, \ldots, K$ **do**
  **for** $i = 1, \ldots, N$ **do**
    Sample noise $(z_j)_{j=1}^n \sim \mu_0$ and real samples $(y_j)_{j=1}^n \sim \nu$
    Compute $x = (\mathrm{T}_{\theta_\ell^i}(z_j))_{j=1}^n$
    Compute $d_{\mathrm{real}} = h_{\phi_\ell^i}(y), \quad d_{\mathrm{fake}} = h_{\phi_\ell^i}(x)$
    Update:
$$\phi_\ell^{i+1} = \phi_\ell^i - \eta_h \nabla_\phi \mathcal{L}_h^D(d_{\mathrm{real}}, d_{\mathrm{fake}})$$

    Sample noise $(z_j)_{j=1}^n \sim \mu_0$ and real samples $(y_j)_{j=1}^n \sim \nu$
    Compute $x = (\mathrm{T}_{\theta_\ell^i}(z_j))_{j=1}^n$
    Compute $d_{\mathrm{real}} = h_{\phi_\ell^{i+1}}(y), \quad d_{\mathrm{fake}} = h_{\phi_\ell^{i+1}}(x)$
    Define the JKO objective

$$J(\theta_\ell^i) = \frac{1}{n} \sum_{j=1}^n \left( \frac{1}{2\tau} \big\| \mathrm{T}_{\theta_{\ell-1}^N}(z_j) - x_j \big\|_2^2 + \mathcal{L}_{\mathrm{T}}^D(d_{\mathrm{real}}, d_{\mathrm{fake}}) \right)$$

    Update:
$$\theta_\ell^{i+1} = \theta_\ell^i - \eta_{\mathrm{T}} \nabla_\theta J(\theta_\ell^i)$$

  **end for**
  Set $\theta_{\ell+1}^1 \leftarrow \theta_\ell^N$ and $\phi_{\ell+1}^1 \leftarrow \phi_\ell^N$
**end for**

---

- **2D MLP.** For low-dimensional experiments, both the generator and the discriminator are implemented as simple multilayer perceptrons.

- **ResNet MMD GAN.** For MMD-based GAN experiments, we use the ResNet generator and discriminator architecture proposed in Zhang et al. (2025). The generator contains approximately 4M parameters.

**Optimization of the kernel** For MMD-based objectives, the choice of the kernel plays a crucial role. Rather than relying on a single fixed kernel, we follow the approach of Zhang et al. (2025) and consider a learnable combination of positive semi-definite kernels.

More precisely, we define the kernel as
$$k(x, y) = \sum_{i=1}^m w_i \, k_i(x, y), \tag{234}$$

where $\{k_i\}_{i=1}^m$ is a collection of predefined kernels and $w_i \geq 0$ are learnable weights. They are optimized jointly with both the critic and generator parameters through backpropagation.

In our experiments, we consider the following sum of kernels:

- a Gaussian kernel with fixed bandwidth: $k_{\mathrm{Gauss}}(x, y) = \exp\left( -\frac{\|x-y\|_2^2}{2\sigma^2} \right)$,

- a mixture of RBF kernels at multiple scales: $k_{\mathrm{RBF\text{-}mix}}(x, y) = \sum_{q=1}^K \exp\left( -\frac{\|x-y\|_2^2}{2\sigma_q^2} \right), \quad \sigma_q = 2^{q-1}\sigma_0$,

- a Laplacian kernel: $k_{\mathrm{Lap}}(x, y) = \exp\left( -\frac{\|x-y\|_1}{\sigma} \right)$,

- an exponential kernel: $k_{\mathrm{Exp}}(x, y) = \exp\left( -\frac{\|x-y\|_2}{\sigma} \right)$,

- Matérn kernels with smoothness parameters $3/2$: $k_{\mathrm{Matérn\text{-}3/2}}(x, y) = \alpha \left( 1 + \frac{\sqrt{3}\|x-y\|_2}{\ell} \right) \exp\left( -\frac{\sqrt{3}\|x-y\|_2}{\ell} \right)$,

- a positive semi-definite Riesz kernel: $k_{\mathrm{Riesz}}(x, y) = -\|x - y\|_2 + \|x\|_2 + \|y\|_2$.

**Optimization.** We train all models with the Adam optimizer (betas (0.5, 0.9)). Learning rates are $\eta_T = 2 \times 10^{-4}$ for the generator $T$ and $\eta_h = 1 \times 10^{-4}$ for the discriminator $h$. We apply a cosine annealing scheduler with minimum learning rate $\eta_{\min} = 5 \times 10^{-5}$ for both networks. For *U-Net* and *LargeNet* we also maintain an exponential moving average (EMA) of the parameters with decay 0.9999, which is known to improve stability and evaluation performance in adversarial training (Polyak & Juditsky, 1992; Yaz et al., 2019).

**Evaluation** To evaluate we mainly use the Fréchet Inception Distance (FID) (Heusel et al., 2017). Images are resized to $299 \times 299$, the expected input of Inception-v3. For MNIST, we replicate the single channel to RGB before resizing. We use the same preprocessing for all models and, unless noted, the same number of generated samples as in the reference set for metric computation. We use the `pytorch-fid` package to compute these values.

## J. Additional Experiments

This appendix presents additional experiments illustrating the theoretical results of Section 6 and Appendices G.2 and G.3 through a 2d comparison between the preconditioned, the unpreconditioned flow and the Natural Wasserstein Gradient Flow, the limitations of Algorithm 1, and complementary experiments and visualizations for Generative Wasserstein Flows.

### J.1. Preconditioned Gradient Flow and Practical JKO

In this section, we provide numerical illustrations of the connection between the practical JKO scheme and the preconditioned gradient flow discussed in Section 6. We consider the parametric setting

$$\min_{\theta \in \mathbb{R}^p} \ \mathcal{F}(\mu_\theta), \qquad \mu_\theta = (F_\theta)_\# \mu. \tag{235}$$

As discussed in Section 3.2, the JKO scheme associated with this parametrization can be written as

$$\theta_{\ell+1} = \arg\min_{\theta \in \mathbb{R}^p} \frac{1}{2\tau} \|F_\theta - F_{\theta_\ell}\|^2_{L^2(\mu)} + \mathcal{F}(\mu_\theta). \tag{236}$$

Alternatively, one may ignore the Wasserstein geometry and consider the standard Euclidean gradient flow

$$\dot{\theta}_t = -\nabla_\theta \mathcal{F}(\mu_{\theta_t}). \tag{237}$$

The result of Proposition G.4 shows that, under suitable regularity assumptions and in the small-step regime, the practical JKO scheme is related to the preconditioned gradient flow

$$\dot{\theta}_t = -G(\theta_t)^{-1} \nabla_\theta \mathcal{F}(\mu_{\theta_t}), \tag{238}$$

where

$$G(\theta) = \int \partial_\theta F_\theta(z)^\top \partial_\theta F_\theta(z) \, \mathrm{d}\mu(z) \in \mathbb{R}^{p \times p}. \tag{239}$$

The goal of the following experiments is to illustrate this connection in two settings: first in a two-dimensional example where parameter trajectories can be visualized, and then in a small neural-network example closer to our generative modeling experiments.

**A two-dimensional toy example.** We first consider a simple example in which the evolution of the parameters can be visualized explicitly. Let

$$F_{(a,b)}(z) = \big(\mathrm{softplus}(a)z_1, \ \mathrm{softplus}(b)z_2\big), \qquad z \sim \mathcal{N}(0, I_2), \tag{240}$$

and consider the Gaussian target distribution

$$\nu = \mathcal{N}\big(0, \mathrm{diag}(\sigma_1^2, \sigma_2^2)\big), \qquad \sigma_1 = 2, \qquad \sigma_2 = 0.5. \tag{241}$$

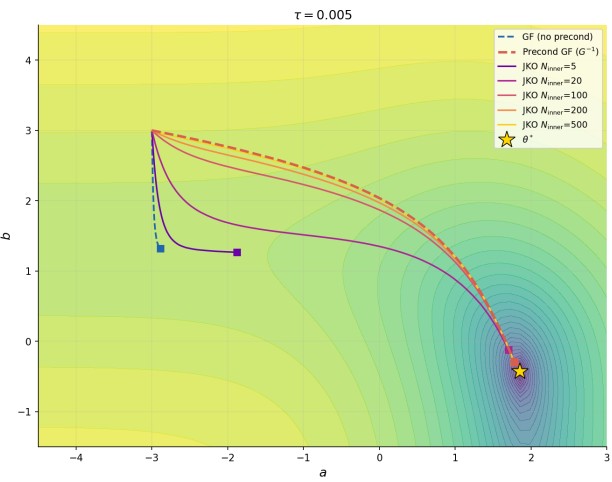

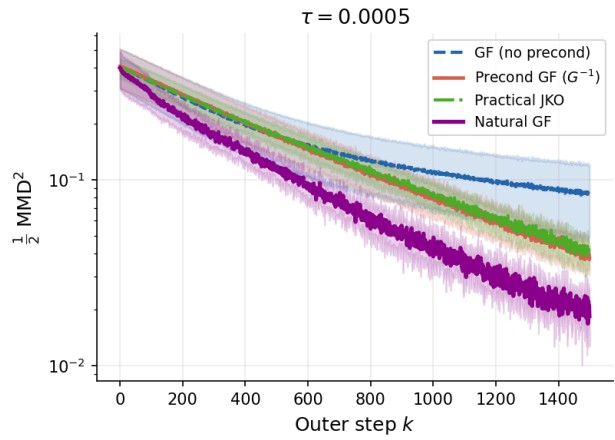

*(a)* Two-dimensional toy example. Increasing $N_{\text{inner}}$ makes the practical JKO trajectory closer to the preconditioned gradient flow.

*(b)* Neural-network example. For small $\tau$, practical JKO and the preconditioned gradient flow display similar behavior, while the Euclidean gradient flow converges more slowly.

*Figure 4.* Numerical illustration of the connection between practical JKO and the preconditioned gradient flow. Left: parameter trajectories in a two-dimensional toy example. Right: decay of the squared MMD objective for a shallow neural-network generator.

We minimize the squared MMD functional with Riesz kernel between $\mu_\theta$ and $\nu$. In this case, the optimal parameters are known explicitly ($a^\star = \text{softplus}^{-1}(2), b^\star = \text{softplus}^{-1}(0.5)$).

We compare the Euclidean gradient flow in Equation (237), the preconditioned gradient flow in Equation (238), the practical JKO scheme in Equation (236), and the kernelized approximation of the Wasserstein natural gradient flow proposed by Arbel et al. (2020). For the practical JKO scheme, the inner optimization problem is solved by gradient descent, and we vary the number of inner steps $N_{\text{inner}} \in \{1, 5, 20, 50, 100, 200\}$. Increasing $N_{\text{inner}}$ corresponds to solving the JKO subproblem more accurately. We use a step size $\lambda_2 = 2\tau = 0.01$ and plot the resulting trajectories together with the MMD landscape and the optimal parameter $\theta^\star = (a^\star, b^\star)$ in Figure 4a.

When $N_{\text{inner}} = 1$, the inner optimization is initialized at $\theta_\ell$, and the gradient of the proximal term vanishes at this point. Therefore, the resulting update exactly coincides with one Euclidean gradient descent step on $\mathcal{F}(\mu_\theta)$. As $N_{\text{inner}}$ increases, the practical JKO trajectory becomes closer to the preconditioned gradient flow. This is consistent with Proposition G.4: when the JKO subproblem is solved more accurately and $\tau$ is small, the practical JKO update follows the geometry induced by $G(\theta)$ rather than the standard Euclidean geometry in parameter space.

**A neural-network example.** We then consider a setting closer to the generative modeling experiments. The target distribution is a two-dimensional distribution supported on three circles, and the objective $\mathcal{F}$ is the squared MMD functional with a Riesz kernel. The generator $F_\theta$ is parametrized by a shallow neural network with approximately 200 parameters.

As in the previous experiment, we compare four optimization strategies:

- the Euclidean gradient flow in Equation (237);

- the preconditioned gradient flow in Equation (238);

- the practical JKO scheme in Equation (236), where the inner subproblem is solved by gradient descent for a fixed number of iterations;

- a kernelized approximation of the Wasserstein natural gradient flow, following Arbel et al. (2020).

We use $K = 1500$ outer optimization steps and $N_{\text{inner}} = 50$ inner steps to solve the JKO subproblem. The JKO step size is set to $\tau = 0.0005$. Since $G(\theta)$ may be singular for neural networks, we use the Moore–Penrose pseudoinverse $G(\theta)^\dagger$ in the preconditioned gradient flow.

The evolution of the squared MMD objective is reported in Figure 4b. For small $\tau$, and when the inner optimization is solved with enough gradient steps, the practical JKO scheme behaves similarly to the preconditioned gradient flow. In contrast, the Euclidean gradient flow converges more slowly, suggesting that the Euclidean geometry in parameter space is poorly adapted to the distributional objective.

The kernelized Wasserstein natural gradient flow follows a different dynamics from the preconditioned gradient flow. This is consistent with Proposition G.3: the metric $G(\theta)$ induced by the parametrization coincides with the Wasserstein information metric only when the parameter-induced velocity fields are gradient fields. This condition is not expected to hold for a general neural-network parametrization.

Finally, we emphasize that each JKO step requires solving an inner optimization problem, here with $N_{\text{inner}} = 50$ gradient descent iterations, making the method significantly more computationally expensive than a single gradient flow update. Nevertheless, for large neural networks, explicitly forming and inverting the matrix $G(\theta)$ is prohibitive. In this regime, practical JKO can be viewed as a tractable way to approximate the preconditioned gradient flow without explicitly computing $G(\theta)$.

### J.2. Limitations of Algorithm 1

In this section, we conduct experiments on MNIST to assess the practical behavior of Algorithm 1. In low-dimensional settings, such as two-dimensional toy datasets, this algorithm performs well (see Appendix J.1). For higher-dimensional data such as MNIST, the resulting samples are of poor quality.

Indeed, in Algorithm 1, we update the generator parameters with

$$-\nabla_\theta \int g\big(\mathrm{T}_\theta(x)\big)\,\mathrm{d}\mu_\ell(x),$$

which is equal to $\nabla_\theta \mathrm{MMD}^2(\mathrm{T}_\theta \# \mu_\ell, \nu)$. Hence this amounts to minimizing a parametrized, regularized squared MMD objective, similarly to Generative Moment Matching (GMM) models (Li et al., 2015), up to an additional regularization term. As a result, the method lacks a sufficiently expressive critic to discriminate complex image distributions.

Nevertheless, we emphasize that MMD flows themselves can perform well in high-dimensional settings. In particular, recent work (Hertrich et al., 2024) demonstrates that particle-based MMD flows can generate realistic images. However, their approach follows a different paradigm from ours. In their method, the MMD flow is first simulated directly at the particle level, producing trajectories that transport samples toward the data distribution. A neural network is then trained in a second stage to regress this transport map from the simulated particle trajectories. By contrast, in Algorithm 1, we directly learn a parametric transport map without relying on explicit particle trajectories. We hypothesize that this direct parametric learning problem is significantly more challenging, especially in high-dimensional settings.

To partially alleviate this issue, we experimented with enriching the kernel by incorporating a learned feature representation. Specifically, we introduce a small encoder trained via an autoencoding objective, and define the kernel in the corresponding latent space, as in (Deng et al., 2026). This modification leads to some qualitative improvement, but the generated samples remain unsatisfactory.

Following the ideas of improved GMM models (Li et al., 2017), we therefore move towards learning the kernel embedding itself through an adversarial objective. This leads to the JKO-regularized MMD GAN formulation described in Algorithm 2, whose empirical performance is discussed in Section 5.

Qualitative results illustrating these claims are shown in Figure 5.

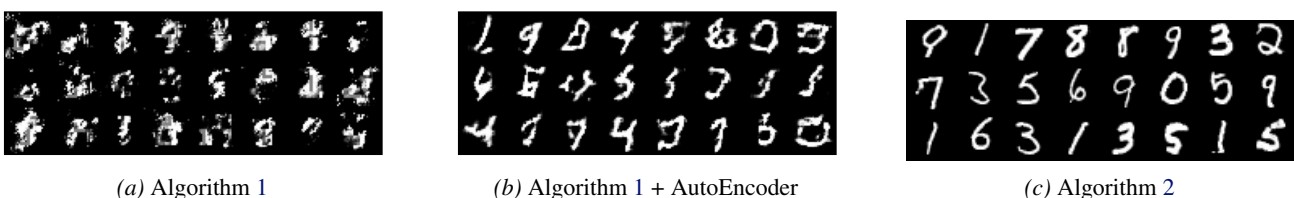

*(a)* Algorithm 1                    *(b)* Algorithm 1 + AutoEncoder                    *(c)* Algorithm 2

*Figure 5.* Generated samples obtained with $\mathrm{MMD}^2$-based generative methods on MNIST.

## J.3. Additional Results on GWF

In this section, we report additional experimental results that complement the main experiments and further support the conclusions of the paper. These results analyze the role of inner optimization accuracy in the JKO scheme, compare the behavior of KL and reverse KL formulations, provide complementary quantitative evaluations, and assess the additional computational cost and qualitative behavior of GWF.

**Impact of inner optimization accuracy in GWF.** We investigate the importance of accurately solving the inner optimization problem arising at each JKO step. To this end, we vary the number of inner optimization steps $N \in \{100, 1000, 2000, 3000\}$ for Algorithm 3 and analyze the resulting sample quality. The evolution of the FID as a function of $N$ is shown in Figure 6, and quantitative results are summarized in Table 2.

We observe that the number of inner optimization steps has a critical impact on the performance of GWF across all considered divergences. Using too few inner steps ($N = 100$) leads to severely degraded performance, indicating that the JKO subproblem is poorly approximated in this regime.

As $N$ increases, the FID consistently improves and eventually stabilizes, with diminishing returns beyond $N \approx 2000$ for most divergences. This suggests that a sufficiently accurate resolution of the inner problem is necessary for the JKO regularization to be effective, while excessively large values of $N$ provide limited additional benefit relative to their computational cost.

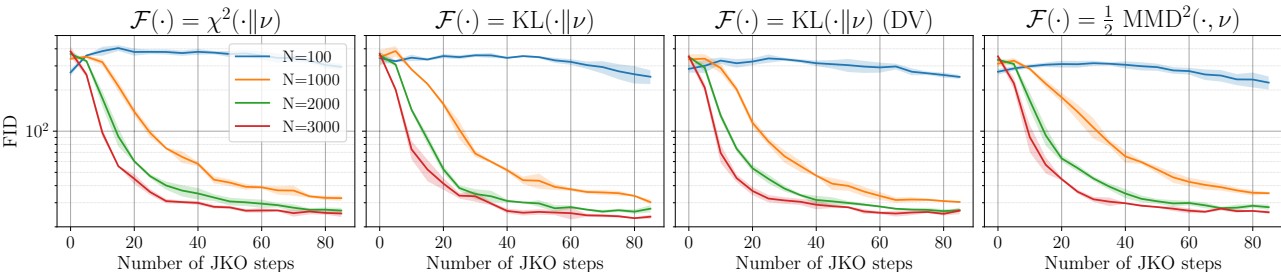

*Figure 6.* Effect of inner optimization accuracy on GWF training. We report the evolution of the FID as a function of the total number of inner optimization steps $N$.

*Table 2.* FID across divergences for varying numbers of inner optimization steps $N$ (averaged over 3 runs).

| Divergence | $N = 100$ | $N = 1000$ | $N = 2000$ | $N = 3000$ |
|---|---|---|---|---|
| $\chi^2$ | $295.16 \pm 15.23$ | $32.40 \pm 1.34$ | $26.29 \pm 1.24$ | $\mathbf{25.04 \pm 1.43}$ |
| KL | $249.94 \pm 28.69$ | $30.31 \pm 1.32$ | $27.13 \pm 1.62$ | $\mathbf{23.72 \pm 0.90}$ |
| KL (DV) | $249.19 \pm 8.45$ | $30.38 \pm 0.37$ | $\mathbf{26.47 \pm 0.73}$ | $26.24 \pm 1.46$ |
| MMD | $226.36 \pm 24.83$ | $35.22 \pm 0.71$ | $27.83 \pm 1.29$ | $\mathbf{25.56 \pm 0.71}$ |

**Additional quantitative results.** We report additional quantitative results corresponding to the experiments discussed in Section 5. In particular, Table 3 compares the DV and classical KL formulation across architectures, while Table 4 summarizes the impact of the JKO step size for various divergences using the *Small-Net* architecture.

We also report complementary evaluation metrics in Table 5, together with their evolution along training in Figure 7. Kernel Inception Distance (KID) (Bińkowski et al., 2018) and Inception Score (IS) (Salimans et al., 2016) are included to complement FID and assess robustness of the conclusions across metrics. Importantly, all three metrics lead to consistent conclusions across divergences and JKO step sizes, indicating that our findings do not depend on a specific evaluation metric.

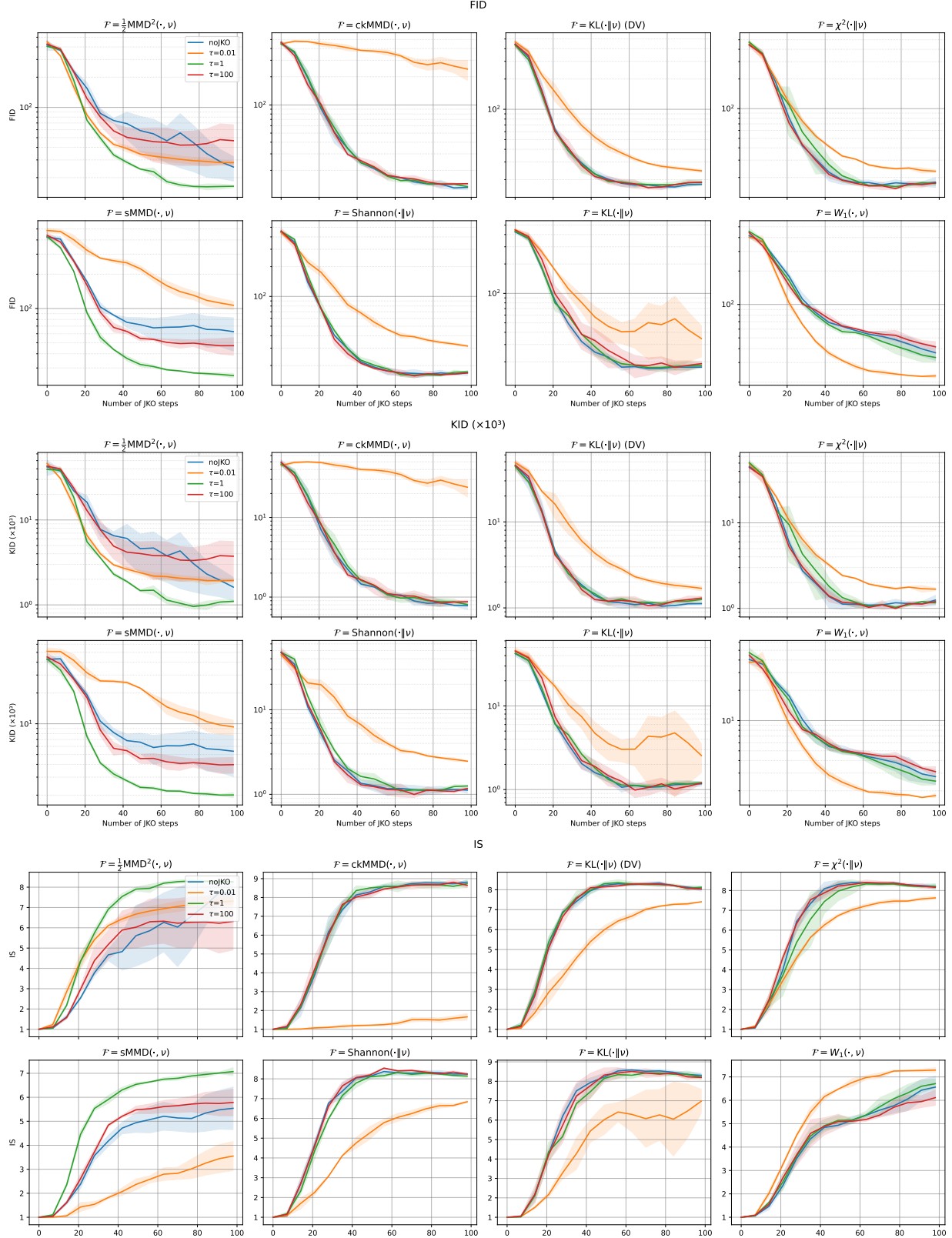

*Figure 7.* Evolution of generative performance metrics along the JKO iterations for different divergences and step sizes $\tau$. Top row: Fréchet Inception Distance (FID) and Kernel Inception Distance (KID). Bottom: Inception Score (IS). These complementary metrics provide a consistent evaluation of generative performance across training.

*Table 3.* Comparison of the DV and classical KL variational formulations at the final JKO iteration for different network architectures on CIFAR-10.

| Architecture | JKO step | DV (mean ± std) | KL (mean ± std) |
|---|---|---|---|
| *Small-Net* | 150 | **25.67 ± 0.44** | 26.90 ± 0.99 |
| *Large-Net*[*] | 50 | **9.35 ± 1.01** | 10.24 ± 1.09 |
| *U-Net* | 80 | **24.66 ± 2.77** | 30.06 ± 0.70 |

[*]*For* Large-Net, *results are reported at the best-performing JKO iteration (40 steps)*

*Table 4.* FID for different divergences and JKO step sizes on CIFAR-10 using *Small-Net* (averaged over 3 runs).

| Divergence | no JKO | JKO $\tau = 0.01$ | JKO $\tau = 1$ | JKO $\tau = 100$ |
|---|---|---|---|---|
| ckMMD | 13.96 ± 0.56 | 129.38 ± 5.19 | **12.78 ± 0.88** | 14.12 ± 0.67 |
| $\chi^2$ | 15.60 ± 0.36 | 22.26 ± 0.57 | **15.01 ± 0.12** | 15.87 ± 0.82 |
| KL | 15.78 ± 0.78 | 26.94 ± 2.26 | **15.47 ± 0.63** | 17.28 ± 1.81 |
| KL-DV | 16.68 ± 0.85 | 25.19 ± 0.28 | **16.56 ± 0.67** | 16.88 ± 0.74 |
| MMD | 49.39 ± 13.48 | 28.36 ± 0.72 | **15.76 ± 0.08** | 21.43 ± 2.58 |
| Shannon | **14.60 ± 0.53** | 32.24 ± 1.63 | 15.92 ± 0.40 | 15.23 ± 0.87 |
| sMMD | 59.79 ± 7.79 | 135.27 ± 10.58 | **26.79 ± 0.23** | 55.59 ± 8.22 |
| Wasserstein-1 | 35.37 ± 8.28 | **22.66 ± 1.08** | 28.91 ± 0.99 | 35.35 ± 3.01 |

*Table 5.* Final generative performance on CIFAR-10 across divergences and JKO step sizes $\tau$ using *Small-Net*. We report Fréchet Inception Distance (FID ↓), Kernel Inception Distance (KID ↓), and Inception Score (IS ↑) as mean ± standard deviation over 3 seeds.

| Divergence | noJKO | $\tau = 0.01$ | $\tau = 1$ | $\tau = 100$ |
|---|---|---|---|---|
| ckMMD | **13.16** ± 0.76 / **0.78** ± 0.08 / **8.81** ± 0.09 | 241.52 ± 58.26 / 23.99 ± 5.84 / 1.66 ± 0.18 | 13.42 ± 0.22 / 0.81 ± 0.02 / 8.73 ± 0.20 | 14.41 ± 0.16 / 0.88 ± 0.04 / 8.65 ± 0.13 |
| $\chi^2$ | 17.93 ± 2.04 / 1.25 ± 0.17 / 8.20 ± 0.14 | 23.16 ± 0.70 / 1.67 ± 0.07 / 7.63 ± 0.08 | **17.39** ± 0.90 / **1.16** ± 0.05 / **8.21** ± 0.11 | 17.63 ± 0.35 / 1.20 ± 0.05 / 8.16 ± 0.07 |
| KL | **17.62** ± 0.18 / **1.17** ± 0.07 / **8.30** ± 0.12 | 34.54 ± 12.19 / 2.55 ± 0.96 / 6.97 ± 0.62 | 18.30 ± 1.03 / 1.21 ± 0.03 / 8.22 ± 0.15 | 19.13 ± 1.08 / 1.20 ± 0.02 / 8.20 ± 0.03 |
| KL-DV | **17.98** ± 0.34 / **1.12** ± 0.05 / 8.04 ± 0.12 | 24.46 ± 0.84 / 1.69 ± 0.11 / 7.39 ± 0.00 | 18.72 ± 1.28 / 1.25 ± 0.11 / **8.12** ± 0.06 | 18.86 ± 0.46 / 1.30 ± 0.06 / 8.04 ± 0.05 |
| MMD | 25.48 ± 7.00 / 1.62 ± 0.50 / 7.71 ± 0.55 | 28.16 ± 1.18 / 1.94 ± 0.09 / 7.31 ± 0.22 | **16.39** ± 0.50 / **1.10** ± 0.04 / **8.27** ± 0.02 | 46.62 ± 20.53 / 3.74 ± 1.84 / 6.32 ± 1.41 |
| Shannon | **16.84** ± 1.01 / **1.12** ± 0.09 / **8.25** ± 0.07 | 31.53 ± 0.63 / 2.46 ± 0.04 / 6.84 ± 0.04 | 17.26 ± 0.14 / 1.26 ± 0.04 / 8.14 ± 0.10 | 16.87 ± 0.22 / 1.17 ± 0.02 / 8.24 ± 0.04 |
| sMMD | 62.58 ± 20.26 / 5.38 ± 2.35 / 5.54 ± 0.89 | 106.66 ± 9.06 / 9.31 ± 1.51 / 3.55 ± 0.61 | **25.73** ± 1.11 / **2.01** ± 0.11 / **7.07** ± 0.16 | 47.09 ± 8.29 / 3.97 ± 0.69 / 5.78 ± 0.56 |
| Wasserstein-1 | 36.77 ± 5.48 / 2.77 ± 0.46 / 6.56 ± 0.47 | **22.75** ± 0.85 / **1.79** ± 0.07 / **7.29** ± 0.10 | 33.34 ± 3.13 / 2.50 ± 0.21 / 6.71 ± 0.20 | 41.57 ± 4.74 / 3.10 ± 0.38 / 6.11 ± 0.33 |

[*]*The runs used in this table correspond to different random seeds than those reported in Table 4, which explains the slight differences in FID values.*

**Computational Cost of JKO.** Tables 6 and 7 provide measurements of the additional computational cost associated with JKO regularization and different network architectures. We observe that the additional computational cost introduced by JKO remains modest (mean overhead $\approx$ +8.9%), and is largely independent of the underlying architecture.

*Table 6.* JKO computational overhead measured as seconds per 1000 generator iterations (sec/kiter) on CIFAR-10 (batch size 256, H100 GPU, averaged over 3 seeds).

| Architecture | Divergence | sec/kiter w/o JKO | sec/kiter w/ JKO | JKO Overhead |
|---|---|---|---|---|
| *Small-Net* | KL | 141 | 162 | +14.8% |
| *Small-Net* | KL-DV | 141 | 162 | +14.4% |
| *Small-Net* | $\chi^2$ | 143 | 155 | +8.3% |
| *Small-Net* | Shannon | 143 | 154 | +7.7% |
| *Small-Net* | W-1 | 156 | 169 | +8.4% |
| *Small-Net* | ckMMD | 160 | 172 | +7.4% |
| *Small-Net* | sMMD | 232 | 244 | +5.1% |
| *Small-Net* | MMD | 377 | 386 | +2.5% |
| UNet | KL-DV | 245 | 275 | +12.1% |
| ResNet-MMD GAN | MMD | 93 | 100 | +8.1% |
| *Large-Net* | KL | 1491 | 1574 | +5.6% |

sec/kiter = seconds per 1000 generator iterations.

*Table 7.* Per-iteration computational cost by architecture on CIFAR-10 (batch size 256, H100 GPU). Costs are reported in seconds per 1000 generator iterations.

| Architecture | Divergence | sec/kiter | vs. *Small-Net* (same divergence) |
|---|---|---|---|
| *Small-Net* | KL-DV | 150 | ×1.0 |
| UNet | KL-DV | 260 | ×1.7 |
| *Large-Net* | KL-DV | 1574 | ×10.0 |
| ResNet-MMD GAN | MMD | 96 | ×0.3 |
| *Small-Net* | MMD | 384 | ×1.0 |

**Forward vs reverse KL.** As discussed in Section 3, the GWF algorithm can be applied to either $\mathcal{F}(\mu) = \mathrm{D}_f(\mu\|\nu)$ (original VGWF formulation (Fan et al., 2022)) or $\mathcal{F}(\mu) = \mathrm{D}_f(\nu\|\mu)$ (original RWP-$f$-GAN formulation (Lin et al., 2021)). It is well known that minimizing $\mathcal{F}(\mu) = \mathrm{KL}(\mu\|\nu)$ may lead to mode-seeking behavior, while minimizing $\mathcal{F}(\mu) = \mathrm{KL}(\nu\|\mu)$ typically encourages mode-covering.

Empirically, however, we did not observe major convergence differences between the two orientations in our setting. The usual mode-seeking behavior associated with $\mathrm{KL}(\mu\|\nu)$ appears mitigated in practice, likely because the variational formulations used in the GWF framework depend only on expectations and are therefore less sensitive to support mismatch than the classical density-based formulation. This observation is illustrated in Figure 8.

**Generated samples.** We conclude with qualitative results illustrating the sample quality achieved by GWF with the $\mathrm{MMD}^2$ on MNIST (Figure 9) and with the KL (DV) on CIFAR-10 (Figure 13). We further visualize the training dynamics through intermediate snapshots (Figures 10 and 11). Finally, to assess potential memorization, we report a nearest-neighbor analysis comparing generated samples to the training dataset (Figure 12).

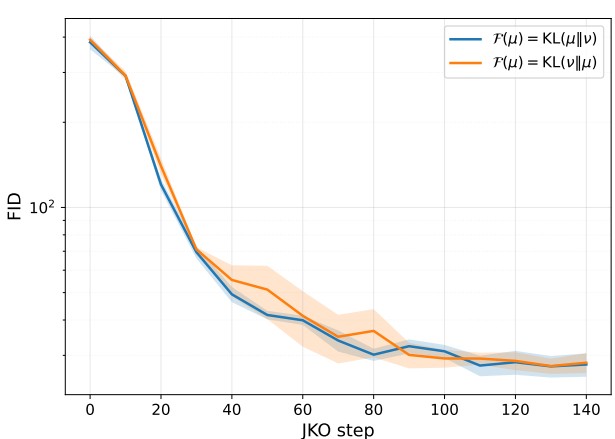

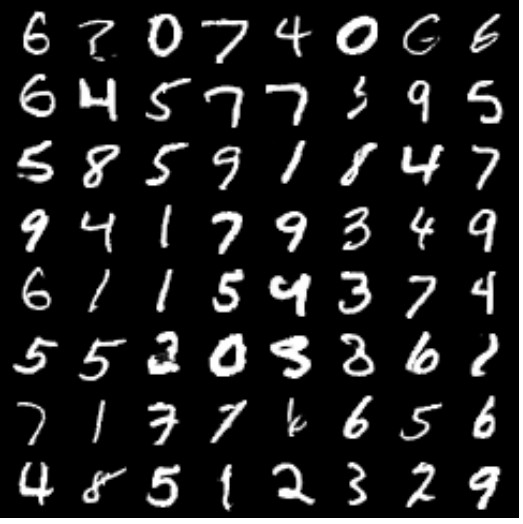

*Figure 8.* Comparison of the convergence behavior of the GWF algorithm when minimizing $\mathrm{KL}(\mu\|\nu)$ and $\mathrm{KL}(\nu\|\mu)$. We report the evolution of the FID score as a function of the JKO step, averaged over multiple runs.

*Figure 9.* Generated samples from GWF with the $\mathrm{MMD}^2$ and $\tau = 0.5$ trained on MNIST with *ResNet MMD GAN* (FID $= 0.95$).

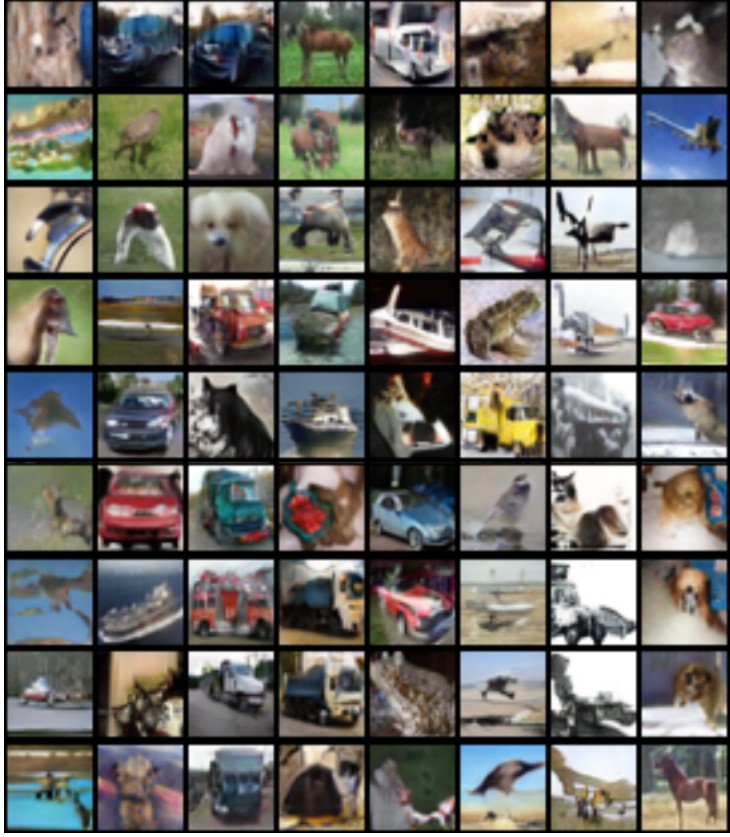

*Figure 10.* Evolution of generated samples for JKO-regularized GWF with KL (DV) on CIFAR-10, using *Large-Net* and step size $\tau = 0.2$. Snapshots are shown from top to bottom every $10\,000$ inner optimization steps.

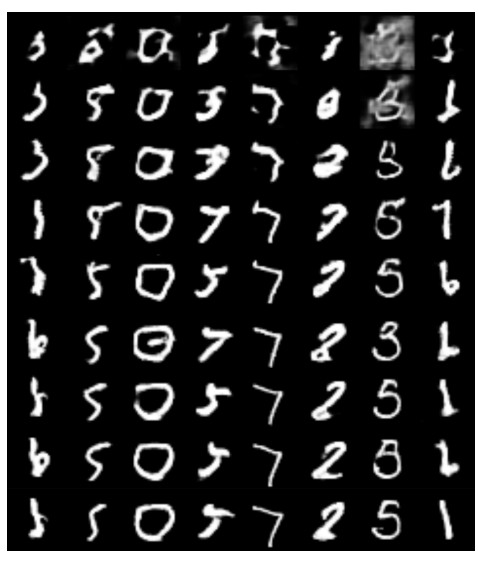
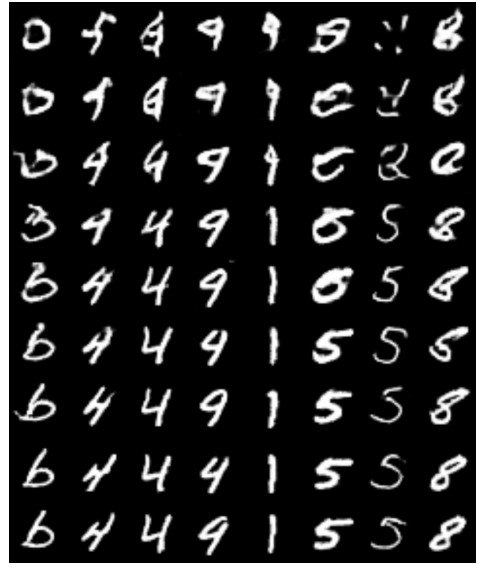

*(a)* GWF ($\tau = 0.1$)         *(b)* Unregularized GWF (no JKO)

*Figure 11.* Evolution of generated samples for GWF with $\mathrm{MMD}^2$ during training. Snapshots are shown from top to bottom every $N = 5\,000$ inner optimization steps, starting from $N = 1000$.

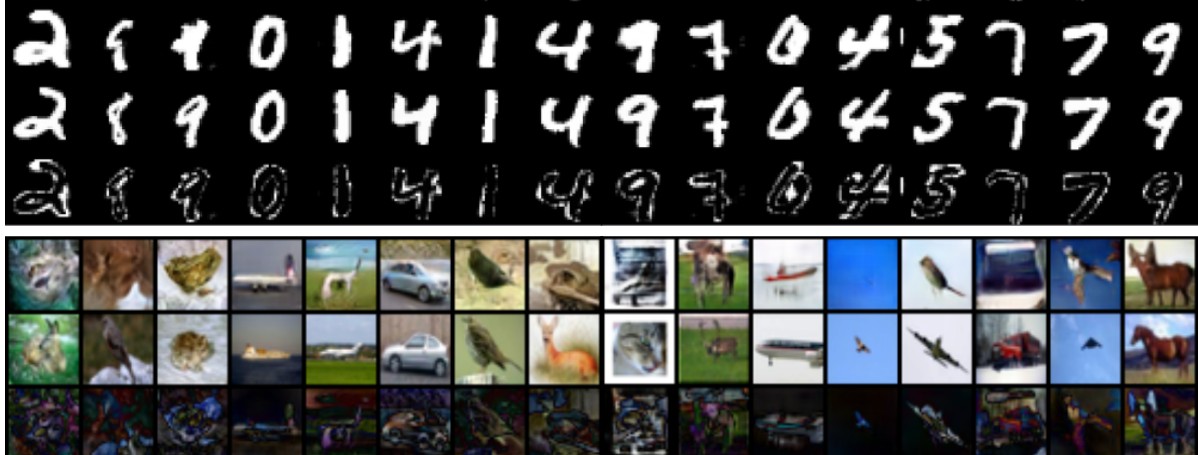

*Figure 12.* Nearest-neighbor analysis to assess potential memorization on CIFAR-10 and MNIST. For each generated sample, we compute the closest image in the training dataset using the $\ell_2$ distance in pixel space. From top to bottom, the first row shows generated samples, the second row shows the corresponding nearest neighbors from the training dataset, and the third row displays the pixelwise differences between them. The CIFAR-10 examples were generated using the Large-Net architecture with the KL-DV formulation (final FID = 9), while the MNIST examples were generated using the *ResNet MMD GAN* architecture with the MMD objective (final FID = 1.74).

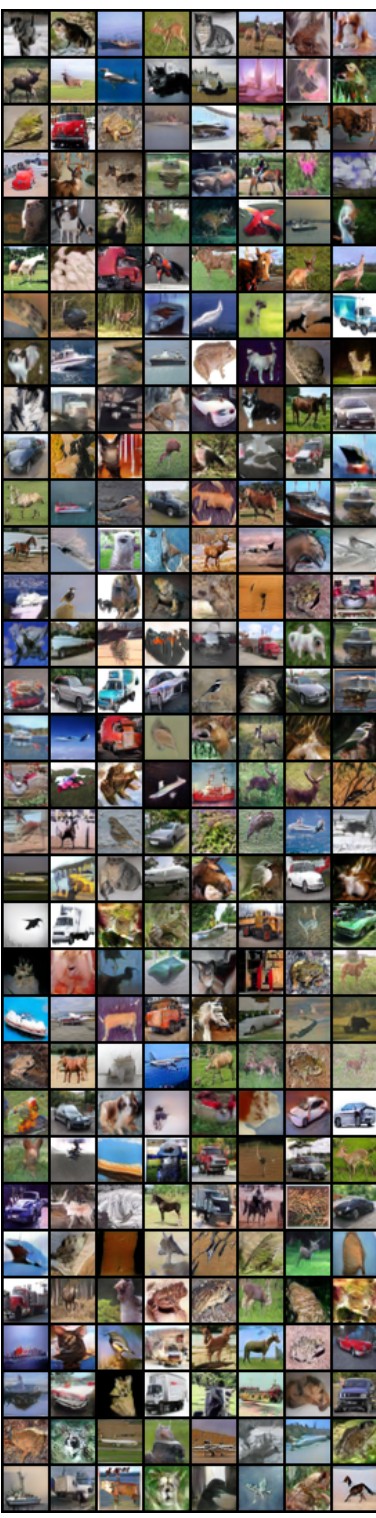

*Figure 13.* Generated samples from WGF with the KL DV formulation with $\tau = 0.2$, trained on CIFAR-10 with *Large-Net* (FID = 8.09).

