# OpenReview forum: "A Unifying View of Variational Generative Wasserstein Flows"
_ICML.cc/2026/Conference — ICML 2026 spotlight_

### Official Review · Reviewer_o5Rn · 2026-02-18

**Soundness:** 3
**Presentation:** 1
**Significance:** 4
**Originality:** 4
**Overall Recommendation:** 5
**Confidence:** 1

**Summary:**

The paper is, in line with its title, a unifying contribution. Its central theme is the use of the JKO scheme for Wasserstein Gradient Flows, which has recently been adopted in various forms within the generative modeling literature.
In Section 3, the authors undertake a unification effort by considering several existing instances of the JKO scheme combined with different f-divergences, and by recasting them within a common theoretical framework. In particular, they discuss connections with VWGF (Fan et al., 2022), SJKO (Choi et al., 2024), as well as with regularized GAN-based approaches.
In Section 4, the scope is extended to divergences admitting a variational formulation, such as Maximum Mean Discrepancy (MMD) and the Wasserstein-1 distance, thereby broadening the applicability of the JKO framework beyond classical KL-based settings.
Experimentally, the paper focuses on image generation. The authors compare two alternative variational formulations of the KL divergence, investigate the use of squared MMD, and analyze the effect of regularization when employing different divergences within the gradient flow framework.

**Compliance With Llm Reviewing Policy:**

Affirmed.

**Final Justification:**

My concerns have been addressed, then I increase my score to accept. For the record, I stress that the submission is not in my area of expertise, as highlighted in the confidence score. I ask the AC to consider this information in order to fairly evaluate the paper.

**Key Questions For Authors:**

1. Unification clarity and scope
Could the authors state their main unification result in a single formal proposition (or theorem) that explicitly specifies the assumptions under which VWGF, SJKO, and regularized GAN-based schemes become equivalent instances of the same JKO framework?

2. Experimental positioning and evaluation metrics
Have the authors considered including a comparison with at least one classical generative modeling baseline (e.g., a standard GAN or diffusion-based model) to better contextualize the empirical performance of GWF-based methods? Additionally, given recent critiques of FID, could the authors report complementary evaluation metrics to strengthen the robustness of their empirical conclusions?

3. Limitations and generalization
What are the main limitations of the proposed unifying framework? For instance, are there classes of divergences, variational formulations, or generative setups for which the JKO-based interpretation breaks down or becomes impractical? A clearer discussion of these boundaries would help delineate the scope of applicability of the proposed theory.

**Limitations:**

No, I think a detailed discussion of the limitations is needed, as highlighted in the comments above.

**Strengths And Weaknesses:**

Before providing a separate evaluation of the four criteria, I would like to stress that my expertise is not closely aligned with the specific topic of the paper. For this reason, my review focuses mainly on general and structural aspects rather than on highly technical details. I have reflected this limitation in the confidence score.


## Soundness ##

Overall, the paper appears to be technically sound. The theoretical objective is clearly stated: to unify previously proposed methods under the JKO framework and to extend the analysis to divergences that admit a variational formulation. On the experimental side, the authors explore several aspects of Gradient Wasserstein Flows (GWF) that are meaningfully connected to the theoretical developments.

My main concern relates to the breadth of the positioning of the work, which sits somewhat between a theoretical and an experimental contribution. From an experimental standpoint, it could strengthen the paper to include a comparison with at least one classical generative modeling approach (even if not in a strictly competitive benchmarking setup). This would help situate the results in a broader generative modeling context and potentially widen the audience.

In addition, the exclusive reliance on FID for image evaluation may be limiting, especially given recent criticisms of this metric (e.g., Jayasumana et al., 2024, Rethinking FID: Towards a Better Evaluation Metric for Image Generation, CVPR). Including at least one complementary evaluation metric would make the empirical validation more robust.

## Presentation ##

In my view, the presentation is currently too oriented toward readers who are already very familiar with the topic. For a broad venue such as ICML, a more pedagogical exposition would likely improve accessibility and impact.

In particular, I suggest reorganizing the theory section as follows:
1. Clearly state which existing methods are being unified.
2. Present the unifying formula(s) explicitly and early.
3. Then derive the corresponding theoretical results in a structured manner.

At present, Section 3 presents several formulas that look related, but I found it difficult to reconstruct the unifying logic step by step. For instance, the sentence:

“To summarize, we have proved in this section the equivalence between several schemes, among which VWGF (Fan et al., 2022) and SJKO (Choi et al., 2024) as well as with the regularized GANs methods applied to F(\mu) = D_f(\mu \|\nu).”

could be reformulated as a formal Proposition, possibly accompanied by a comparative table summarizing assumptions, objective functionals, and resulting update schemes. This would significantly improve clarity.

A similar comment applies to Section 4. A unifying paper, especially on technically involved material, benefits greatly from intuitive summaries, schematic diagrams, or graphical overviews. Otherwise, there is a risk that the central unification message becomes diluted in technical details.

Regarding Section 6, it appears somewhat disconnected from the rest of the manuscript and insufficiently contextualized. I would suggest expanding it and moving it to the appendix, using the freed space to discuss an important missing aspect: the limitations of the proposed framework.

## Significance ##

Given that the topic lies outside my main area of expertise, I cannot provide a strong assessment of significance. At a general level, unifying different approaches under a common framework is usually a valuable contribution. However, for the work to reach its full potential impact, the presentation should be improved and made more pedagogically accessible, as discussed above.

## Originality ##

Similarly, I cannot provide a definitive judgment on originality. At first glance, the authors appear to meaningfully connect several strands of literature. However, given my limited familiarity with the area, I cannot exclude the possibility of missing references or partially overlapping prior work.

---

> ### Author Rebuttal · Authors · 2026-03-30
>
> We thank the reviewers for their feedback. All figures/tables referenced in the responses below, produced in response to reviewer requests for this rebuttal, are available in an additional experiments document accessible via the following anonymous link:https://ibb.co/album/zWBBLS. We answer their questions below.
>
> **1. Unification.**
>
> To clarify this point, we provide the Figure 1 of the additional experiments document. It illustrates how the different formulations arise from the same GWF framework depending on the choice and orientation of the objective functional.
>
> In the revised version, we will add a clearer formal statement of this correspondence together with this explanatory figure and a concise proof when needed. We will also carefully consider the reviewer’s suggestion to reorganize Section 3. The detailed derivation of the equivalence VWGF ≡ f-GAN + RWP is already provided in our response to Q1 of Reviewer gnbM.
>
> **2. Comparison with classical generative approach.**
>
> We perform comparisons with classical GANs in several of the experiments, as it correponds to the *noJKO* setting, see for instance Table 1 and Figure 2 of the additional experiments file.
>
> For comparison with Diffusion, we refer to the paper which introduced SJKO [1] (particular case of GWF) which compared their results with these methods (Table 2 in [1]).
>
> **Evaluation metric.**
>
> Following the reviewer’s suggestion, we conducted an additional experiment reporting complementary evaluation metrics (KID, and Inception Score). The Table 1 of the additional experiments document summarizes the results, and figure 2 of the additional experiments document illustrates the evolution of these metrics across JKO iterations for each divergence. The improved performance observed in these new experiments is due to a small grid search over the *SmallNet* hyperparameters. Importantly, all three metrics lead to consistent conclusions across divergences and JKO step sizes.
>
> **3. Limitations of the method.**
>
> We agree that clarifying the scope and limitations of the framework is important. We will add a dedicated discussion at the end of Section 5 to explicitly discuss these aspects.
> Below we give some key limitations of the proposed framework.
>
> • *Adversarial optimization.*  A key limitation of the current approach is that it relies on adversarial optimization. In recent years, alternative training paradigms such as score matching have reduced the reliance on adversarial objectives, partly due to their improved stability and ease of optimization in practice. Adversarial training may therefore be more challenging to tune and deploy. Nevertheless, recent work shows that carefully designed adversarial models can still achieve strong performance [2].
>
> • *Not SOTA performance.*
> While the framework could potentially be extended in the future to reach state-of-the-art performance, the current goal of the GWF framework is primarily conceptual and methodological rather than to optimize raw benchmark performance. To the best of our knowledge, models directly derived from the current formulation are not yet competitive with the most highly optimized generative models.
>
> • *Need for JKO regularization and tuning of the step size $\tau$.*
> A central parameter in the framework is the JKO step size $\tau$, which controls the strength of the regularization. As illustrated in Figure 2 of the additional experiments document, the choice of $\tau$ must be carefully tuned for each divergence. For instance, for the Wasserstein-1 objective, small values of $\tau$ improve convergence, whereas for sMMD small values of $\tau$ may lead to very slow convergence, and intermediate values (e.g., $\tau=1$) provide the best performance.
>
> This dependence makes the tuning of $\tau$ divergence-specific rather than universal, and the optimal choice may also depend on the architecture and dataset.
>
> • *Cost–benefit trade-off of JKO regularization.*
> While JKO regularization can improve stability or convergence for certain objectives, this benefit is not uniform across divergences. As shown in Table 2 of the additional experiments document, the introduction of JKO leads to an average computational overhead of approximately 8.6\%. In situations where the regularization provides limited performance improvement, the additional computational cost may outweigh its practical benefit.
>
> **About Section 6**
>
> The current Section 6 is preliminary but stays, in our opinion, important in this work. We provide a discussion about the link with first parts and additional result in our response to q4 of reviewer 4FT4.
>
>
> [1] Choi, J., Choi, J., and Kang, M. Scalable Wasserstein Gradient Flow for Generative Modeling through Unbalanced Optimal Transport. ICML, 2024.
>
> [2] Huang, N., Gokaslan, A., Kuleshov, V. and Tompkin, J. The GAN is dead; long live the GAN! A Modern GAN Baseline. NeurIPS, 2024.

---

> > ### Author Rebuttal · Reviewer_o5Rn · 2026-03-31
> >
> > My concerns have been addressed, then I increase my score to accept. For the record, I stress that the submission is not in my area of expertise, as highlighted in the confidence score.

---

### Official Review · Reviewer_4FT4 · 2026-02-20

**Soundness:** 2
**Presentation:** 3
**Significance:** 3
**Originality:** 3
**Overall Recommendation:** 5
**Confidence:** 4

**Summary:**

The paper studies the implementation and simulation of Wasserstein gradient flows of f-divergences, MMD-regularized f-divergences and squared MMDs. To this end, the authors focus on simulating JKO iterations via variational representations of f-divergences as saddle-point problems which are simulated in a GAN-like manner (and establish a relation to GAN-training). The authors consider such gradient flows in the context of generative modeling and implement several examples.

**Compliance With Llm Reviewing Policy:**

Affirmed.

**Final Justification:**

The authors addressed my questions about the numerical results and in response I raised my score to accept.

Actually, I think that this is a very strong paper until section 5. For section 6 I remain skeptical. I thought a bit further about the authors response and I am not convinced about it. Actually, I think that **the equivalence between the GF in Prop 6.2 and the scheme in (24) is not true**. Consider the following $1$-dimensional example: Choose $\mu_0=\mathcal N(0,1)$, the mapping $F_\theta(x)=x+\theta^4-\theta^2 + 0.1 \theta$ and $\mathcal F(\mu)=W_2^2(\mu,\mathcal N(-10,1))$. Then, we have that $J(\theta)=W_2^2(\mathcal N(\theta^4-\theta^2+0.1 \theta, 1),\mathcal N(-10,1))=(\theta^4-\theta^2+0.1\theta + 10)^2$. This function has two local minima (coinciding with the local minima of the polynomial $\theta^4-\theta^2+0.1\theta$), one with positive $\theta$ one with negative $\theta$ and the positive local minimum has a higher function value than the negative local minimum. If we now start with $\theta_0$ in the positive local minimum, the GF from Prop 6.2 will do nothing since $\nabla_\theta \mathcal F(\mu_\theta)=0$ (actually it is not clear what happens since this is precisely one of the cases where $G$ is non-invertible). However, the iteration in (24) will just jump to the global minimum since it does not see that $\theta_{1}$ is far from $\theta_0$, it just sees that $F_{\theta_1}$ is close to $F_{\theta_0}$.

The authors explanations are only correct if $\theta\mapsto F_\theta$ is **globally injective** such that convergence in $\theta$ and in $F_\theta$ are equivalent, which is a strong assumption which is never met in practice. I strongly encourage the authors either to rework or to remove this claim for the final version.

**Key Questions For Authors:**

Questions are stated in the strengths and weaknesses section. As most important I consider number 4.2 (what is the benefit of section 6? Currently I see neither theoretical nor practical insides from this; maybe the authors can clarify) and number 3 (clarifications on the numerical results).

**Limitations:**

If the optimization in the variational representation of the f-divergence / MMD is replaced by optimizing over a neural network, it is no longer clear that solutions of the JKO scheme exist and are stable. Considering the focus of the paper it is understandable not to investigate these issue, but it should be mentioned.

**Strengths And Weaknesses:**

1. Significance and Originality

The paper builds on the literature on the simulation of Wasserstein gradient flows with generative models. None of the sections / considered methods is completely new, but the authors show "on-the-way" a couple of small results connecting and extending the literature summing up to a nice contribution. Even though I don't see this kind of models to beat state-of-the-art generative models in the future, there is certainly sufficient interest in the general topic.

2. Presentation:

Section 6 seems to be a bit disconnected to the rest of the paper, but otherwise the paper is well-written.

3. Numerical Results

3.1 on the classical small test datasets like CIFAR and MNIST the authors should report something like L2-nearest neighbors in order to ensure that the network is generating something else than reproducing the samples from the training set.

3.2 please add something about the computational cost of the implementations. In particular, for comparing the different network architectures this is important.


4. Finally, I am a bit skeptical that section 6 is sound and useful. I generally agree with the general intuition, that the gradient descent on $\theta$ is some kind of preconditioned gradient flow in the image. However:

4.1 the velocity field from (20) is not unique. It is only unique in the regular tangent space of the Wasserstein metric (defined as the closure of all $C_c^\infty$ gradients in $L_2(\mu_{\theta_t})$) which coincides with the minimal norm solution of (20) (actually, any velocity field can be orthogonally decomposed into an element from the tangent space and a zero-divergence field; under enough regularity this is known as Helmholtz decomposition). Now the authors show in Prop 6.1 that the $v_{\theta_t}$ fulfills the continuity equation. But what is missing in the proof is that $v_{\theta_t}$ also belongs to the tangent space (which is the space where the Wasserstein gradients live). The same issue appears in Prop 6.2.

4.2 A naive and direct question on a much less technical level: What are the computations of this section good for? I think I didn't really get the "overall plan" behind the construction. Is the overall idea to run a preconditioned gradient flow on $\theta$?

4.3 "Future works will focus on the theoretical study of the convergence of these flows for neural networks." This is an ambitious goal considering that for many choices of $\mathcal F$ (global) convergence of the corresponding WGF is not really understood; and including a highly non-convex embedding doesn't make it better...

5. Literature:

- A simulation of the MMD^2 flow (like Alg 1) for generative modeling is already done in "Generative Sliced MMD Flows for Discrepancies for Riesz kernels", ICLR 2024 with significantly better results than Figure 4.

---

> ### Author Rebuttal · Authors · 2026-03-30
>
> We thank the reviewers for their feedback. All figures/tables referenced below are available in an additional experiments document via the anonymous link:https://ibb.co/album/zWBBLS. We answer below.
>
> **3.1. Generalization.**
>
> As we can see in Fig. 4 in the additional experiments document, although the nearest neighbors share visual similarities, they are not identical to the generated samples, suggesting that the model does not simply reproduce training examples.
>
> **3.2. Computational cost for different architectures**
>
> See Tables 2–3 in the additional experiments document; they will be included in the revised paper.
>
> **4.1, 4.2 and 4.3. About Section 6:**
>
> The goal of Sec. 6 is to understand the behavior of the JKO schemes studied in this paper, in particular the dynamics induced when the search space is restricted to parametrized distributions. As standard in optimization, we analyze the associated continuous-time dynamics, which are simpler than discrete-time ones.
>
> Prop. 6.1–6.2 make these dynamics explicit: the velocity field corresponds to a preconditioned Wasserstein gradient depending on the parameter dynamics $(\theta_t)\_t$.
> The reviewer is right, velocity fields are not unique ; however, we do not claim uniqueness of the field. Our result only shows that the parametric vector field satisfies the continuity equation. To avoid ambiguity, we will replace statements such as “the velocity field in (20) can be identified with $v_\theta$” by “$v_\theta$ satisfies (20)”.
>
> Moreover, the relevant space is not the full Wasserstein tangent space, but a smaller space of admissible maps (tangent space to the parametric family) given at the end of Prop. 6.2, equipped with the metric defined in l385 (2nd column).
>
> At the end of Sec. 6, we show that parametric JKO updates correspond to a time-discretization of this natural parameter flow, currently presented informally. We will clarify this point by stating it explicitly as a proposition.
>
> Empirically (Appendix I1), this preconditioning improves convergence speed. Direct simulation would require inverting a matrix $G$ of size equal to the number of parameters, which is impractical for large neural networks. A key message of Sec. 6 is that JKO updates provide a scalable way to discretize these dynamics, implicitly incorporating this geometric preconditioning without explicitly inverting $G$.
> In this sense, Sec. 6 provides a theoretical justification for using JKO updates when training parametrized generative models.
>
> A second message is that these continuous-time dynamics differ from Wasserstein Natural Flows in [1], which rely on true Wasserstein distances rather than the $L^2$ upper bound used in eq. (24). Identifying these dynamics is a first step toward analyzing convergence. While global convergence is challenging, many related questions remain active topics in WGF (e.g., convergence to critical points, well-posedness, discretization error).
>
> **5. Link with [2].**
>
> We are aware of this work and thank the reviewer for pointing it out. While related in spirit, it follows a different paradigm from Algorithms 1–2.
>
> Their method can be viewed as a two-stage procedure. First, they simulate an MMD particle flow toward the empirical data distribution by iteratively updating a ***fixed set of particles***. Second, they train a neural network to approximate the resulting transport via regression between initial and transported particles. This is closely related to recent drifting models [3].
>
> In short, they first simulate a ***particle flow***, then introduce neural networks as a separate step to reproduce this transport.
>
> By contrast, our framework directly learns a transport map pushing the latent distribution toward the data distribution through a parametric objective. This setting is more challenging, as the network must learn the transport directly without relying on explicit particle trajectories.
>
> In the current version, Appendix I.2 attributes the limited performance of *Algorithm 1* to the limited discriminative power of a fixed kernel in high dimension. We agree this explanation is incomplete; the different optimization setting (particle flow vs. direct parametric flow) also plays a role. We will clarify this point and explicitly cite the above work in the revision, and we thank the reviewer for the suggestion,.
>
> **JKO via Neural Nets.**
>
> Approximation effects induced by parameterization are discussed in our response to Q3 of Reviewer bPXp.
>
> [1] Chen, Y. and Li, W. Optimal transport natural gradient for statistical manifolds with continuous sample space. Information Geometry, 2020.
>
> [2] Hertrich, J., Wald, C., Altekrüger, F. and Hagemann, P. Generative sliced MMD flows with Riesz kernels. ICLR, 2024.
>
> [3] Deng, M., Li, H., Li, T., Du, Y. and He, K. Generative Modeling via Drifting. preprint, 2026.

---

> > ### Author Rebuttal · Reviewer_4FT4 · 2026-04-01
> >
> > Thank you for your replies. Regarding my key questions, I would consider the points about the numerics as addressed.
> >
> > Also regarding Section 6, I think understand now better the intention. I think the key point which I misunderstood was that not the gradient flow from Prop 6.2 is used to simulate the JKO scheme but vice versa (maybe I was reading too fast). However, I am still a bit confused:
> >
> > - The map $F_\theta$ is in general (in most cases since neural networks are usually overparameterized) not injective. In particular, also the argmin in (24) is not unique (so in any case the $=$ should be a $\in$, but this is not my main point).  If $\theta$ is a minimizer of $J(\theta)$, the iterations in (24) still could select different solutions among the minimizers of $J$, but the gradient flow would just be constant. I think that this issue is not limited to the case that $\theta$ already arrived at a (local) minimizer of $J$, but could also appear during the minimization process.
> >
> > - More generally, there is no reason, why the matrix $G$ should even be invertible. What happens with the flow if $G$ is non-invertible?
> >
> > Can you make the link between the iteration (24) and the gradient flow in Prop 6.2 precise? Is there any kind of convergence from the $\theta_l$ generated by this iteration to the gradient flow? Intuitively, I am quite sure that convergence of the iterates $\theta_l$ towards the gradient flow is false. Maybe one can establish convergence of $\mu_{\theta_l}$ to $\mu_{\theta_t}$ in (weakly or in Wasserstein, where $\theta_t$ is the solution of the GF) or $F_{\theta_l}$ to $F_{\theta_t}$. But even for that I am not sure and in any case such a convergence would require a proof...
> >
> > Since major parts of my questions were addressed I am raising my score, but still remain doubtful about the last section.

---

> > > ### Author Response · Authors · 2026-04-05
> > >
> > > We thank the reviewer for their new questions and respond below.
> > >
> > > **Non-uniqueness of the minimizer.**
> > >
> > > We agree with the reviewer that, in general, the minimizer in (24) may not be unique, in particular for overparametrized neural networks. However, the proximal term naturally selects a solution close to the previous iterate, for a small step-size. Hence in practice, the update remains stable. We will replace the equality sign with the $\in$ symbol in the definition of the $\arg\min$ in the revised version.
> > >
> > >
> > > **Link between (24) and the GF in Prop. 6.2.**
> > >
> > >
> > > We clarify the local connection between the proximal update (24) and the continuous-time dynamics of prop 6.2.
> > >
> > > Under smoothness assumptions, namely Fréchet differentiability of $\theta \mapsto F_\theta$, continuity and L2 integrability of the Jacobian and after some computations, the first-order optimality condition for the parametric JKO (24) yields the expansion
> > >
> > > $$G(\theta_{\ell+1}) \frac{(\theta_{\ell+1}-\theta_\ell)}{\tau} =-\nabla_\theta \mathcal F(\mu_{\theta_{\ell+1}}) + o(1) \quad (\tau \to 0).$$
> > >
> > > This shows that the scheme can be interpreted as a discretization (backward Euler) of the preconditioned gradient flow in Prop. 6.2 when $\tau \to 0$. We have added a formal statement and the detailed proof to our revised version.
> > >
> > > We emphasize that we do not claim convergence of the iterates $\theta_\ell$ toward the continuous-time trajectory $\theta_t$. The result only establishes a local connection between the proximal update and the continuous-time dynamics.
> > >
> > >
> > > **Invertibility of $G(\theta)$.**
> > >
> > > Thank you for this remark. Indeed, in general, $G(\theta)$ might not be invertible. Identifying when it is actually well invertible is beyond the scope of this paper and will be investigated in future works. In Appendix E.2 and E.3, we already identified particular cases where it is well invertible: the Gaussian case with $F_\theta$ affine with non singular diagonal covariance matrix. We will clarify that we work in Section 6 under the assumption that G is invertible. We note that it is a common assumption, see e.g. [1, 2], or that it is possible to use its pseudo-inverse instead [2, 3]. In practice, in our numerical experiments with small networks (Appendix I1), $G(\theta)$ is often singular or poorly conditioned, hence we compute updates using its pseudo-inverse (e.g., via `torch.linalg.pinv()`).
> > >
> > >
> > >
> > >
> > > [1] Dumont, T., Lacombe, T., & Vialard, F-X. (2026). Learning Monge maps by lifting and constraining Wasserstein gradient flows. arXiv preprint arXiv:2603.25182.
> > >
> > > [2] Zuo, X., Zhao, J., Liu, S., Osher, S., & Li, W. (2025). Numerical analysis on neural network projected schemes for approximating one dimensional Wasserstein gradient flows. Journal of Computational Physics, 114501.
> > >
> > > [3] Jin, Y., Liu, S., Wu, H., Ye, X., & Zhou, H. (2025). Parameterized Wasserstein gradient flow. Journal of Computational Physics, 524, 113660.

---

### Official Review · Reviewer_bPXp · 2026-03-03

**Soundness:** 4
**Presentation:** 4
**Significance:** 3
**Originality:** 3
**Overall Recommendation:** 6
**Confidence:** 3

**Summary:**

The paper presents a unified framework for generative modeling based on Wasserstein gradient flows, using the Jordan–Kinderlehrer–Otto (JKO) scheme as the main building block. The key idea is to view training as a sequence of proximal steps in probability space, where each update balances: (i) decreasing a divergence/objective toward the target distribution, and (ii) staying close to the previous distribution in $W_2$ (Wasserstein-2) distance.

A central contribution is a primal (map-based) formulation of the JKO step (via Brenier’s theorem), where the update is written as an optimization over a transport map $T$. For $f$-divergence objectives, the paper then uses a variational representation to rewrite the JKO subproblem as a sample-based min–max problem, which naturally leads to an adversarial implementation: $T$ acts like a generator/transport map and $h$ acts like a critic/discriminator. This avoids explicit density estimation and makes the method practical with neural networks.

The paper also shows how this perspective connects to and unifies several existing approaches, including formulations related to unbalanced optimal transport (UOT) and alternative variational objectives. In particular, it compares different variational forms of KL divergence (including the Donsker–Varadhan representation) and explains how they fit into the same JKO-based framework.

Beyond $f$-divergences, the framework is extended to IPMs such as MMD and Wasserstein-1, showing that many GAN-like objectives can also be interpreted through this generative Wasserstein-flow lens. Empirically, the paper studies the effect of JKO regularization (especially the step size $\tau$) and finds that it can improve training stability/performance, with stronger gains in some objectives (notably MMD/$W_1$) than others.

**Compliance With Llm Reviewing Policy:**

Affirmed.

**Final Justification:**

My final recommendation is Strong Accept. I found this to be a very strong submission that scores highly across soundness, clarity, significance, and originality. The paper presents a principled and well-executed framework for generative modeling based on Wasserstein gradient flows and JKO updates, and it offers a valuable unifying perspective connecting several important existing formulations.

From a technical standpoint, the work appears sound and the main claims are well supported by the theoretical and empirical results. The presentation is also strong: the paper is clearly written, well structured, and succeeds in making a fairly technical framework accessible. I also view the contribution as significant, since this type of unifying perspective can shape how related methods are understood and developed in future work. In terms of originality, although some ingredients are known, the way they are combined and articulated here is meaningful and insightful.

The rebuttal addressed my main questions satisfactorily and reinforced my positive assessment. In particular, it clarified several practical and conceptual points without exposing any substantive weakness in the work. Taking the paper, the rebuttal, and the broader discussion into account, I am updating my recommendation to Strong Accept.

**Key Questions For Authors:**

1. For the adversarial JKO formulations (especially for (f)-divergences / KL variants), could the authors comment on optimization stability in practice (e.g., critic updates, regularization, failure modes, sensitivity to architecture choices), and whether certain variational forms are consistently easier to optimize than others?

2. Computational overhead vs. benefit of JKO regularization. Could the authors provide a clearer discussion (or a small benchmark) of the additional computational cost introduced by JKO regularization compared to corresponding non-JKO baselines, and in which regimes (datasets/objectives) the improvement justifies this cost?

3.  Several derivations rely on assumptions such as absolute continuity / existence of Brenier maps and function-class conditions for the variational formulations. Could the authors clarify which parts of the framework remain exact under practical neural parameterizations, and which should be interpreted as approximations (e.g., due to restricted map/critic classes, optimization non-convexity, finite samples)?

**Limitations:**

The paper appears to adequately discuss the main practical/theoretical limitations of the proposed framework. I do not have major concerns regarding unaddressed negative societal impact beyond the standard risks associated with generative modelling methods.

**Strengths And Weaknesses:**

Overall, I find this to be a **very strong paper**. In my view, it is technically sound, clearly written, and addresses a relevant problem with a compelling and well-executed perspective. I did not identify any major flaws in the current version.

**Soundness.** The submission appears technically sound. The methodology is appropriate for the problem setting, and the paper presents a coherent combination of theoretical development and empirical validation. The main claims seem well supported by the analysis and experiments provided, and the paper is careful in connecting the mathematical formulation to the implemented training procedures. I did not notice obvious inconsistencies, unsupported claims, or methodological issues. From my reading, the work is rigorous and competently executed.

**Presentation.** The paper is clearly written and well structured. The narrative is easy to follow, especially given the technical nature of the topic. The progression from background concepts to the proposed framework and then to the practical formulations is handled well. I found the exposition strong overall, and the paper does a good job of making the core ideas understandable while preserving mathematical precision. In particular, the connection between the theoretical framework and the resulting optimization problems is well explained.

**Significance.** I consider the paper relevant and potentially impactful. It addresses an important topic in generative modeling and provides a principled framework that may help unify and clarify multiple existing approaches. Even beyond immediate empirical gains, I think the conceptual contribution is valuable: this kind of unifying perspective can influence how future work is formulated and compared. The paper may be especially useful to researchers working at the intersection of optimal transport, variational formulations, and generative modeling.

**Originality.** The originality is good. Even if some ingredients build on existing ideas (e.g., JKO / OT / variational formulations), the paper offers a meaningful and well-articulated synthesis that leads to useful new formulations and insights. In that sense, the contribution is original not only in specific technical derivations, but also in the way it organizes and connects prior concepts into a unified framework. This is a legitimate and valuable form of originality.

**Weaknesses / limitations.** I did not find major flaws. My main caveat is that I am not fully up to date with the entire recent literature and state of the art in this area, so I cannot make a definitive judgment on whether every aspect of the novelty claim is maximal relative to all concurrent works. However, based on my current knowledge and reading, the contribution appears strong, well motivated, and technically solid.

**Summary.** In general, I think this is a very good paper. It is technically sound, clearly presented, relevant, and sufficiently original, and I did not identify substantive weaknesses in the current version.

---

> ### Author Rebuttal · Authors · 2026-03-30
>
> We thank the reviewer for their positive comments. Additional experiments are available via the anonymous link:https://ibb.co/album/zWBBLS. We answer below.
>
> **1. JKO formulation: Practical insights.**
> - *KL.* DV can be sensitive due to the log-sum-exp structure, potentially yielding high-variance gradients if poorly regularized. We did not observe major convergence differences between KL and reverse KL. The usual mode-seeking behavior of reverse KL appears mitigated, likely because the variational forms depend only on expectations and are less sensitive to support mismatch.
> - *Inner steps.* The number of generator/critic updates per JKO step matters. A moderate number is sufficient: too many increase cost, too few degrade the subproblem solution (Fig. 3 of additional exp file). Asymmetric schedules (e.g., 2 critic steps per generator step) can be used (as in [1,2]), but we did not observe consistent gains across divergences.
> - *Gradient penalty.* As in GANs, insufficient critic regularization leads to unstable training across divergences. The required strength and form depend on the divergence, implementations are provided in Appendix G.
> - *Architecture / parametrization.* Performance generally increases with capacity, but not strictly monotonically (a tuned *SmallNet* may outperform Unet). For MMD variants, critic embedding dimension is critical: too small → underfitting; too large → overly complex representations.
> - *Regularization effect.* The JKO step size $\tau$ controls the stability–speed trade-off and strongly depends on the divergence.
>
> **2. Computational overhead of JKO.**
> We benchmarked JKO computational cost on CIFAR-10 (Table 2 in additional exp file). Mean overhead is +8.9% (range: +2.5%–14.8%), indicating a modest additional cost.
>
> Table 1 of additional experiments file, shows that JKO benefit heavily depends on $\tau$ and the divergence. Small $\tau$ slows convergence, while moderate values typically improve stability at similar cost. JKO is particularly beneficial for objectives requiring stronger reg. (e.g., W1 or some MMD variants), whereas divergences such as KL often perform well with large $\tau$ or even without JKO. A clear insight from Figure 1 (in the paper) is precisely this variation in effectiveness across divergences, highlighting that the benefit of JKO is objective-dependent rather than uniform.
>
> Importantly, JKO overhead is independent of the base architecture. We also report cost comparisons across all networks used in the paper (Table 3 of the additional experiments file).
>
> **3. Exact vs. approximate aspects.**
> At the theoretical level, formulations in Sec. 3–4 are exact, as no parametrization is introduced. Objectives are defined for $T \in L^2(\mu_l)$ and $h \in C_b(\mathbb{R}^d)$ or $h \in \mathcal H_k$.
>
> **Exact.** The JKO scheme and its reformulations are exact under standard assumptions (e.g., abs. continuity ensuring OT maps via Brenier). In particular, equivalence between primal, dual, and adversarial JKO formulations (e.g., VWGF and S-JKO) holds exactly in the infinite-dimensional setting (Prop. 3.1). The interpretation of parametric dynamics as projected Wasserstein gradient flows (Sec. 6) is also exact at the velocity-field level.
>
> **Approximations from neural parametrization.**
> - *Restricted generator class.* Transport maps are parameterized by NNs and thus only approximate the admissible set (e.g., gradients of convex functions). As noted in Sec. 3.1, the quadratic penalty $||T-I||^2_{L^2(\mu)}$ becomes an upper bound on the W2 distance when the class is restricted. Min–max theorems cannot be applied directly in Prop. 3.1 under restriction; however, VWGF and SJKO still coincide since they rely on the same updates.
> - *Restricted critic class.* Variational forms (e.g., Eq. (5), (8), (11)) involve suprema over function spaces (e.g., $C_b(\mathbb{R}^d)$) approximated by NNs, introducing bias (as in GANs).
> - *Finite-sample estimation.* Expectations are replaced by empirical averages, yielding stochastic approximation error.
> - *Geometric approximation.* In Sec. 6, we show that parametric updates correspond to a preconditioned WGF. However, as discussed there, the induced metric $G(\theta)$ does not in general coincide with the true Wasserstein metric on the family of distributions (because $ ||T_\theta-I||^2_{L^2(\mu)}$ is an upper bound on the $W_2$ distance, and this is why these flows differ from the Natural WGF introduced in [3], that would rely on the true $W^2_2(\mu_\theta, \mu_\theta'))$. Hence, the resulting dynamics should be interpreted as approximations of true Wasserstein gradient flows restricted on $(T_\theta\sharp\mu)_{\theta \in \Theta}$.
>
> **Overall.** The framework is exact at the infinite-dimensional level, but approximate in practice due to parametrization, optimization, and sampling, as most generative models. This distinction will be clarified in the final version.
>
> [1] Li et al., Neurips 2017  [2] Arjovsky et al., 2017 [3] Chen et al., 2020

---

> > ### Author Rebuttal · Reviewer_bPXp · 2026-04-02
> >
> > Thank you for the clear and detailed rebuttal. The authors have adequately addressed the main questions I raised in my original review. In particular, the response provides useful practical clarification on optimization stability for the adversarial JKO formulations, including the role of critic/generator update schedules, critic regularization, architecture choices, and the effect of the JKO step size.
> >
> > The additional discussion of computational overhead is also helpful. The reported benchmark suggests that the extra cost introduced by JKO regularization is modest, while the benefits depend in a meaningful way on the objective and the choice of step size. This makes the trade-off much clearer.
> >
> > I also appreciate the clarification regarding which parts of the framework are exact at the infinite-dimensional level and which parts become approximate in practice due to neural parameterization, restricted function classes, optimization, and finite-sample estimation. This distinction is important, and the rebuttal explains it well.
> >
> > Overall, the rebuttal resolves my questions and further strengthens my positive assessment of the paper.

---

### Official Review · Reviewer_gnbM · 2026-03-12

**Soundness:** 4
**Presentation:** 3
**Significance:** 3
**Originality:** 4
**Overall Recommendation:** 5
**Confidence:** 2

**Summary:**

The authors propose a unified framework for generative modelling through the lens of Wasserstein Gradient Flows (WGFs). They show that the variational Wasserstein Gradient Flow (VWGF; Fan et al., 2022) and Scalable JKO (Choi et al., 2024) are optimising the same objective. RWP-GAN (Lin et al., 2021) for KL divergence corresponds to VWGF with reverse KL divergence and the proximal scheme proposed by Baptista et al. (2025) is a JKO step. These links are novel and previously unknown. They extend the framework beyond f-divergences to integral probability metrics (IPMs) and the squared maximum mean discrepancy (MMD), deriving new JKO-regularised versions of the original losses and fleshing out the link to MMD-GANs. They also study how gradient descent on the neural network (used to parameterise the push-forward map) parameters can be viewed as a projected Wasserstein gradient descent. Experiments on image datasets such as MNIST and CIFAR-10 are performed to assess how JKO regularisation across various divergences affects generative performance, quantified using the Fréchet Inception Distance (FID).

**Compliance With Llm Reviewing Policy:**

Affirmed.

**Final Justification:**

This is a theoretical paper. Propositions 3.1 and 3.2 are correct and non-trivial, the KL polarity observation is clean, and Section 6 provides a geometric interpretation of parametric gradient descent. The main weaknesses I raised were the missing KL polarity proof, the unexplained DV performance gain, and the uncontextualised CIFAR-10 numbers and each of which was addressed in the rebuttal. The KL polarity proof sketch is credible, and the clarification that the experiments were designed for controlled comparison rather than competitive performance is appropriate. I recommend accept.

**Key Questions For Authors:**

1. Can the authors provide a complete proof of the claim that RWP-GAN (Lin et al., 2021) for KL divergence corresponds to VWGF with reverse KL divergence (and vice versa)?
2. The DV formulation beats the standard variational KL across all three architectures in Figure 1, but this has been attributed to the tighter lower bound. Can the authors clarify the reason for this performance gain?
3. FID for the CIFAR-10 experiments are far from state-of-the-art models. With larger models and some hyperparameter tuning, would it be possible to achieve better results?

**Limitations:**

Societal impact: The work is primarily theoretical in nature and does not directly involve human subjects or sensitive data. The authors can mention that as with other advances in generative modeling, improper use of learned models in downstream decision-making systems could lead to unintended consequences. It would be fair to expect that the authors do not anticipate significant negative societal impacts beyond those common to the broader use of statistical modeling techniques.

**Strengths And Weaknesses:**

__Strengths:__
1. Propositions 3.1 and 3.2 are interesting results and genuinely novel.
2. Experiments on a range of divergences is thorough and demonstrates the effect of JKO regularization clearly.
3. The correspondence between RWP-GAN (Lin et al., 2021) for KL divergence and VWGF with reverse KL divergence is interesting but requires a proof.
4. Viewing gradient descent on the neural network  parameters being viewed as a projected Wasserstein gradient descent is also intriguing.

__Weaknesses:__
1. The claim RWP-GAN (Lin et al., 2021) for KL divergence corresponds to VWGF with reverse KL divergence (and vice versa) is not proved in the appendix.
2. The experiments about the Donsker-Varadhan formulation of the KL divergence demonstrate better results than GWF. The reason for this gain is not explained adequately.
3. The CIFAR-10 FID numbers are far from state-of-the-art. The paper must state explicitly that the experimental setup is not designed for competitive generation quality.


__Minor Issues:__
1. $h^{c_{\tau}}$ is not defined before being used in Eq. (8).
2. $C_{b}(\mathbb{R}^{d})$ is not introduced before being used in Eq. (5).
3. On line 259, "we discuss Integral" instead of "we discuss of Integral".
4. The first two sentences of the introduction are awkward and don't read very well. Consider reframing them to read more naturally.
5. On line 156, Corollary is misspelt.

---

> ### Author Rebuttal · Authors · 2026-03-29
>
> We thank the reviewer for their positive comments. All figures/tables referenced in the responses below, produced in response to reviewers requests for this rebuttal, are available in an additional experiments document accessible via the following anonymous link:https://ibb.co/album/zWBBLS. We answer their questions below.
>
> **1. Proof that RWP-GAN $\equiv$ VWGF with reverse KL.**
>
> We first show that VWGF and RWP-$f$-GAN coincide when applied to the same objective functional. We then use the standard relationship between opposite orientations of an $f$-divergence which implies the desired result. We will add this proof to the revised version. Note that a comprehensive figure illustrating the equivalence derived from Sections 3 and 4 and the overall GWF framework is provided as Figure 1 in the additional experiments document.
>
> Proof.
>
> The standard VWGF derivation in Section 3 is written for $\tilde F(\mu)=D_f(\mu||\nu)$, which leads, after the reparameterization trick, to Eq(12).
>
> We apply exactly the same derivation to the functional $F(\mu)=D_f(\nu||\mu)$. Using the variational representation of the $f$-divergence in this orientation gives
>
> $$ D_f(\nu||\mu) = \sup_h \{ \mathbb E_\nu[h(y)] - \mathbb E_{\mu}[f^*(h(x))] \}. $$
>
> The corresponding VWGF step reads $$\inf_{\tilde T} \frac{1}{2\tau} \mathbb{E}\_{\mu\_l} [ ||\tilde T - Id||_2^2 ] + \sup_h \{ \mathbb E\_\nu[h(y)] - \mathbb E\_{\mu\_l} [ f^*(h(\tilde T(x))) ] \}.$$
>
> Using the reparameterization trick described in Section 3.2 and given in equation (12), yields
>
> $$ \inf_T \sup_h \{ \mathbb E\_\nu[h(y)] - \mathbb E_{\mu_0} [ f^*(h(T(x))) ] + \frac{1}{2\tau} \mathbb{E}\_{\mu_0} [ || T - T_\ell||_2^2 ] \},$$
>
> which is exactly the RWP-$f$-GAN objective.
>
> Therefore,
>
> $$ \text{VWGF applied to } D_f(\nu\|\mu) \equiv \text{RWP-}f\text{-GAN applied to } D_f(\nu\|\mu). $$
>
> We now relate this formulation to the orientation used in the paper.
>
> Defining the adjoint generator $\tilde f(r):= r f(1/r)$, we have the standard identity
>
> $$ D_f(\nu||\mu) = D_{\tilde f}(\mu||\nu). $$
>
> Consequently,
>
> $$ \text{RWP-}f\text{-GAN on } D_f(\nu||\mu) \equiv \text{VWGF on } D_{\tilde f}(\mu||\nu), $$
>
> which corresponds exactly to the VWGF formulation used in the paper.
>
> In particular, for $f(r)=r\log r$, the left-hand functional corresponds to the KL divergence, while the right-hand functional corresponds to the reverse KL divergence.
>
> **2. Why DV beats standard KL.**
>
> The DV formulation is a tighter lower bound in the sense that the objective is pointwise greater or equal to the classical variational bound (see DV vs. Variational KL paragraph in Appendix C.2), and thus is closer to the true KL divergence that we actually aim to minimize. In that sense, we hypothesize that the better results obtained with the DV formulation come from the better approximation of the objective.
>
> **3. FID not SOTA.**
>
> As correctly noted, the main experiments reported in Table 1 and Figure 1 (in the paper) were conducted using relatively small architectures (referred to as *Small-Net* in the paper), with approximately 1M parameters for the generator (see Appendix H for details). This design choice was motivated by computational considerations and by the need to run a large number of controlled comparisons across divergences and JKO step sizes $\tau$.
>
> More importantly, these experiments were not designed to achieve the best possible performance, but rather to isolate and study the effect of JKO regularization and to compare it against a baseline without regularization under a unified experimental setup. Using the same architecture across all divergences allows us to attribute performance differences to the optimization scheme rather than to model capacity. This clarification will be added in the revised version of the paper.
>
> Nevertheless, we performed additional hyperparameter tuning for this architecture. We obtained improved FID scores on Cifar10 for all divergences using the same architecture (median FID across divergences at τ = 1 decreased from 28.16 in the paper to 17.85 after tuning). We observe a similar behaviour to the experiment depicted in Figure 1 (in the paper), but with better performance (see Table1 and Figure2 of the additional experiments file).
>
> In addition, more expressive architectures can indeed yield significantly better results. For instance, while not SOTA, using the *Large-Net* architecture employed in S-JKO, we obtain substantially improved FID scores, achieving 9.35 ± 0.89 for KL-DV and 11.47 ± 1.10 for KL.
>
> **Minor issues and Societal impact.** We thank the reviewer for pointing out the wording and editing issues, it will be corrected in the revised version.
>
> Concerning the societal impact, we thank the reviewer for this remark. In the revised version, we will add a brief clarification in the Impact Statement section addressing potential societal impacts, consistent with those commonly associated with generative modeling methods for images/videos.

---

> > ### Author Rebuttal · Reviewer_gnbM · 2026-04-02
> >
> > Thank you for your detailed rebuttal. I consider my concerns fully resolved. The authors directly addressed the main issues I raised: they supplied the missing proof of the claim that RWP-GAN with KL divergence corresponds to VWGF with reverse KL divergence, clarified the performance gain in the DV setting, and explicitly stated that the CIFAR-10 experiments were not meant to be state-of-the-art but rather to isolate the effect of JKO regularisation. They also provided additional experimental results that show that better results can be achieved with larger networks and hyperparameter tuning.

---

### Decision · Program_Chairs · 2026-04-30

**Decision:**

Accept (spotlight)

**Comment:**

The authors present a unified theoretical framework—Generative Wasserstein Flows—that formulates various generative modeling approaches as continuous-time Wasserstein gradient flows. They demonstrate that many existing methods optimizing $f$-divergences or Integral Probability Metrics (such as Maximum Mean Discrepancy) can be elegantly cast as parametric JKO schemes. All the reviewers agree that this is a strong paper, its contributions are novel, and would make worthy addition to the field. I thus recommend acceptance.